# A global analysis of extreme coastal water levels with implications for potential coastal overtopping

Rafael Almar [1✉], Roshanka Ranasinghe[2,3,4], Erwin W. J. Bergsma [1,5], Harold Diaz[1], Angelique Melet [6], Fabrice Papa [1,7], Michalis Vousdoukas [8], Panagiotis Athanasiou [3,4], Olusegun Dada[9], Luis Pedro Almeida[10,11] & Elodie Kestenare [1]

Climate change and anthropogenic pressures are widely expected to exacerbate coastal hazards such as episodic coastal flooding. This study presents global-scale potential coastal overtopping estimates, which account for not only the effects of sea level rise and storm surge, but also for wave runup at exposed open coasts. Here we find that the globally aggregated annual overtopping hours have increased by almost 50% over the last two decades. A first-pass future assessment indicates that globally aggregated annual over-topping hours will accelerate faster than the global mean sea-level rise itself, with a clearly discernible increase occurring around mid-century regardless of climate scenario. Under RCP 8.5, the globally aggregated annual overtopping hours by the end of the 21st-century is projected to be up to 50 times larger compared to present-day. As sea level continues to rise, more regions around the world are projected to become exposed to coastal overtopping.

[1] LEGOS (CNRS/IRD/CNES/Toulouse University), Toulouse, France. [2] Department of Coastal and Urban Risk & Resilience, IHE Delft Institute for Water Education, Delft, The Netherlands. [3] Harbour, Coastal and Offshore Engineering, Deltares, Delft, The Netherlands. [4] Water Engineering and Management, Faculty of Engineering Technology, University of Twente, Enschede, The Netherlands. [5] EOLab, French Space Agency (CNES - Centre National d'Etudes Spatiales), Toulouse, France. [6] Mercator-Ocean, Toulouse, France. [7] Universidade de Brasília (UnB), IRD, Instituto de Geociências, Brasília, Brazil. [8] European Commission, Joint Research Centre (JRC), Ispra, Italy. [9] Federal University of Technology, Akure, Nigeria. [10] Universidade Federal do Rio Grande do Sul, Rio Grande, Brazil. [11] +ATLANTIC, Edifício LACS Estrada da Malveira da Serra, Cascais, Portugal. ✉email: rafael.almar@ird.fr

Over the twenty-first century, sea level rise (SLR) is projected to at least double the frequency of coastal flooding at most locations around the world[1–4] potentially affecting millions of people living in low-lying coastal zones, unless effective flood mitigation strategies are implemented in the years ahead[5–9]. Regions with limited water-level variability at the coast (i.e., short-tailed flood-level distributions), mainly located in the Tropics, are likely to be the most affected[1]. An increase in flood occurrence in low-lying, vulnerable coastal zones could force significant population migration and socio-economic damage[3,10–13]. One process that could contribute to coastal flooding is overtopping, which occurs when the extreme coastal water level (ECWL, as defined by Gregory et al.[14]) exceeds the maximum coastal elevation (e.g., dunes, dykes, cliffs[15]). However, the occurrence of overtopping does not necessarily imply that the entire low elevation coastal zone is flooded, rather, this phenomenon drives localized coastal flooding, immediately adjacent to areas of overtopping. The flooding that may occur due to overtopping is likely to be both temporally and spatially variable due to the combined effects of temporal and alongshore gradients in breaking wave heights and alongshore variations in coastal elevation maxima. In addition, overtopping can result in protection failure[16], which can result in broader, more catastrophic flooding[17].

ECWL results from the combination of several different coastal processes (Fig. 1 and Eq. (1)): the regional sea level anomaly (here referred to as SLA) due to the steric effect, ocean circulation, and transfer of mass from the continents (ice sheets, glaciers, land water) to the ocean, storm surge (DAC) due to atmospheric pressure and winds, astronomical tide (T), and wave effects here referred to collectively as runup (R), which includes a time-averaged component (setup) and an oscillatory component (swash) (see Melet et al.[18]).

$$ECWL = SLA + DAC + T + R \tag{1}$$

Despite the important role that ocean waves play in determining water level at the coast[4,18,19] via wave setup and wave runup, their contribution is still largely disregarded in most studies, notably due to the lack of global information on detailed coastal topography, which is required to compute these wave contributions to ECWL accurately. Topographic and foreshore slope data, excepting local datasets acquired during site-specific studies, are often coarse, outdated, or simply non-existent in large parts of the world, leading to inaccurate estimates of potential coastal flooding and their associated risks to coastal communities and assets. Owing to this, global studies (e.g., [1,2,12,18,20,21]) that do account for the contribution of waves to extreme sea levels are still based on highly simplified coastal topography/bathymetry assumptions (e.g., constant slope worldwide). While many studies have acknowledged that local topography can greatly influence wave runup, and consequently flood exposure and the associated risk[9,13,22,23], no concerted efforts have been taken yet to address this shortcoming on a global scale. In this study, we address this need by combining a new state-of-the-art global digital surface elevation model (ALOS World 3D from JAXA at 30 m spatial resolution, hereafter referred to as AW3D30[24,25]) with ECWLs derived from a combination of satellite altimetry, tide and surge models, and wave reanalyses, taking into account the key contribution of wave runup at open coasts. Using the occurrence of ECWLs above the maximum coastal elevation as a proxy for coastal overtopping, here we quantify, for the first time, the global scale increase of potential coastal overtopping in recent decades, and present first-pass, globally aggregated projections of future coastal overtopping in response to projected global mean sea level rise (GMSLR) over the twenty-first century.

## Results and discussion

**Global distribution of maximum coastal elevation and sub-aerial coastal slope.** Of the global coastline spanning approximately 1.5 million kilometers, only about one-third is exposed to waves, with direct wave action being less relevant on sheltered coasts, including bays, estuaries, and rugged coasts[11,26]. The topology of open coasts is highly variable, comprising open sandy coasts, barrier islands, cliffs, river deltas, and engineered coasts[27]. Two key coastal topographical parameters that are relevant for coastal overtopping are the foreshore slope, which influences wave runup and thus ECWL, and the maximum sub-aerial coastal elevation, which sets the threshold that is to be exceeded by ECWL for overtopping to occur. The global distribution of the maximum sub-aerial coastal protection elevations (within 1 km landward of the shoreline) derived from the AW3D30 data base (see "Methods" section for a description of the transect extraction

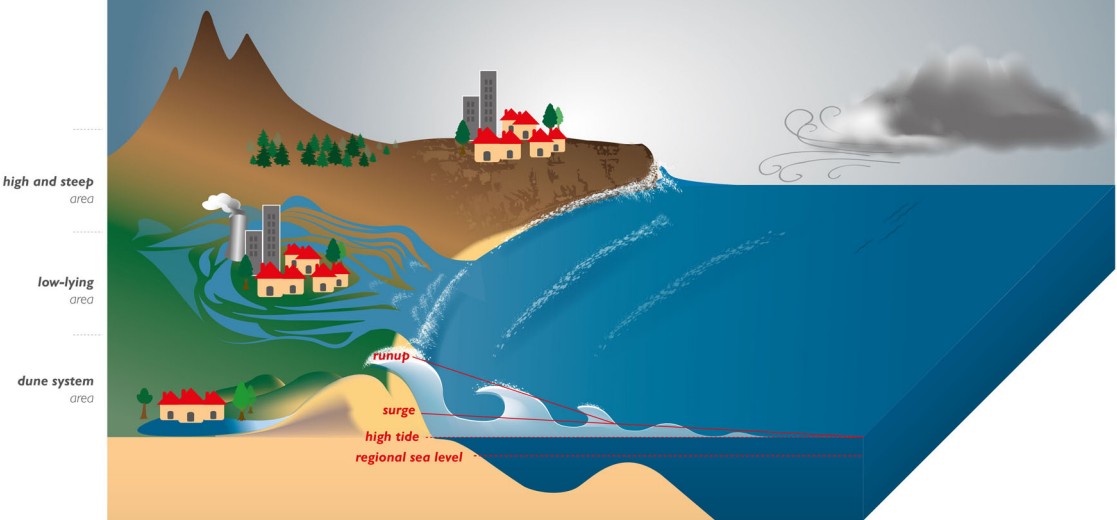

**Fig. 1 Schematic of the processes governing coastal overtopping and the different levels of potential flooding depending on coastal topography.** The extreme coastal water level (ECWL) results from the combination of regional sea level anomaly (SLA) due to the steric effect, ocean circulation and transfer of mass from the continents (ice sheets, glaciers, land water) to the ocean, astronomical tide (T), storm surge due to atmospheric pressure and winds (DAC), and wave runup (R), decomposed into a time-averaged component (setup) and an oscillatory component (swash).

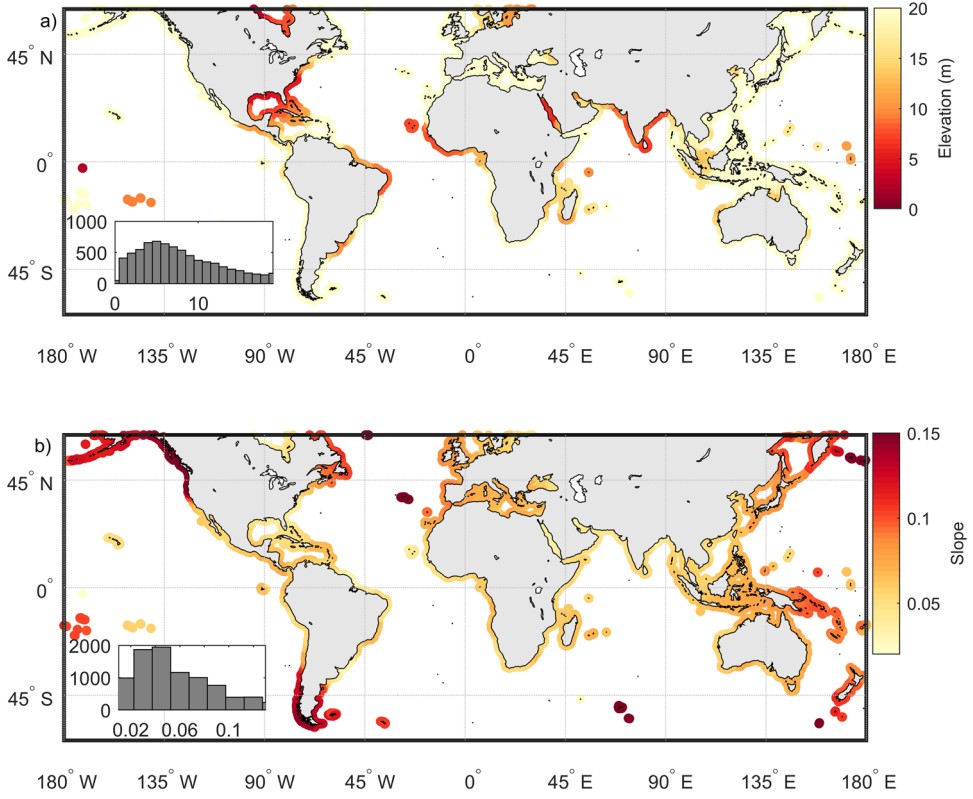

**Fig. 2 Global coastal topography along the world's coastline. a** Maximum coastal elevation and **b** sub-aerial coastal slope. Insets show the distribution of elevations (mean = 7 m) and slopes (median = 0.04). For readability, values have been regionally smoothed in this figure, such that regional patterns are clearly distinguishable.

method adopted) is shown in Fig. 2a. These maximum coastal elevations also take into account coastal dunes and coastal structures if resolved by the 30 m resolution of AW3D30 data base. The maximum sub-aerial coastal elevation appears to generally increase with latitude (Fig. 2a) and has a global average of 7 m. The sub-aerial coastal slope (Fig. 2b) relevant for wave runup calculations is computed from the shoreline to the maximum sub-aerial coastal elevation as derived in Fig. 2a (see "Methods" section). The global median value of the coastal slope thus derived is 0.04. Regional patterns are visible, such as the along-coast gradient in coastal slope along the west coast of North America, from relatively low (0.04) in the tropics to rather steep (0.15) in high latitudes with rockier coastlines. Similar features are observed in the southern hemisphere. Africa, the continent with the largest length of sandy coasts[28], generally has gentle coastal slopes.

**Overtopping events over recent decades**. ECWL over the 23 years between 1993 and 2015 were computed at 14,140 coastal profiles situated along the open coasts of the world using Eq. (1). Regional SLA was derived at each computational profile from satellite altimetry sea level time series using the SSALTO/DUACS multi-mission data[29]. Storm surge values (DAC) for the study period were taken from a global application of the MOG2D-G model[30], forced by surface winds and atmospheric pressure from the ERA-interim reanalysis[31] while astronomical tides ($T$) were extracted from the global tide model FES (Finite Element Solution[30]). Wave runup was computed using two forms (for steep and mild slopes, as appropriate for the profile under consideration—see "Methods" section) of the commonly used Stockdon et al.[32] parametrization, using wave conditions from the ERA-interim global wave reanalysis. All these individual

components feeding into Eq. (1) were re-sampled at an hourly resolution to enable computing ECWL at an hourly resolution such that co-occurrence of high values in tides, storm surge, and waves will be captured in the analysis.

From the ECWL time series derived using Eq. (1) and the maximum coastal elevations, the potential for overtopping was computed at each transect at a resolution of 0.5° alongshore. A detailed illustration and validation of the methodology adopted for this computation are provided in Fig. S5 for selected historical overtopping events (e.g., Katrina in the USA, Xynthia in Europe/ France). Figure 3a shows the time-averaged (averaged over the 1993–2015 period) annual number of overtopping hours ($N_{a,l}$) around the world. Regional hotspots of overtopping can be seen in northern Europe, southern Mediterranean, western Canada, far-eastern Russia, eastern Africa and Madagascar, and parts of southeast Asia and northern Australia. The limitations associated with the application of our approach at deltaic coasts are addressed in the "Limitations and way forward" section.

The annual number of overtopping hours ($N_{a,g}$) exhibits a positive (i.e., increasing) trend (computed using the complete hourly time series of ECWL, discretized into years) in most parts of the world over the period 1993–2015 (Fig. 3b). The highest rates of increase are observed in the Gulf of Mexico, Eastern Europe (Baltic Sea), Southern Mediterranean, Eastern Africa and Madagascar, far-eastern Russia, and parts of Southeast Asia and Northern Australia. This might be because most of these regions generally have small variability in ECWL (variance of the time series), and hence, even small increases in regional sea level can have a large impact on overtopping[33]. A few areas appear to have experienced a small trend over 1993–2015, mainly in the mid to high latitudes: e.g., the west coast of North America, Northern Europe, most of South America, and South Asia.

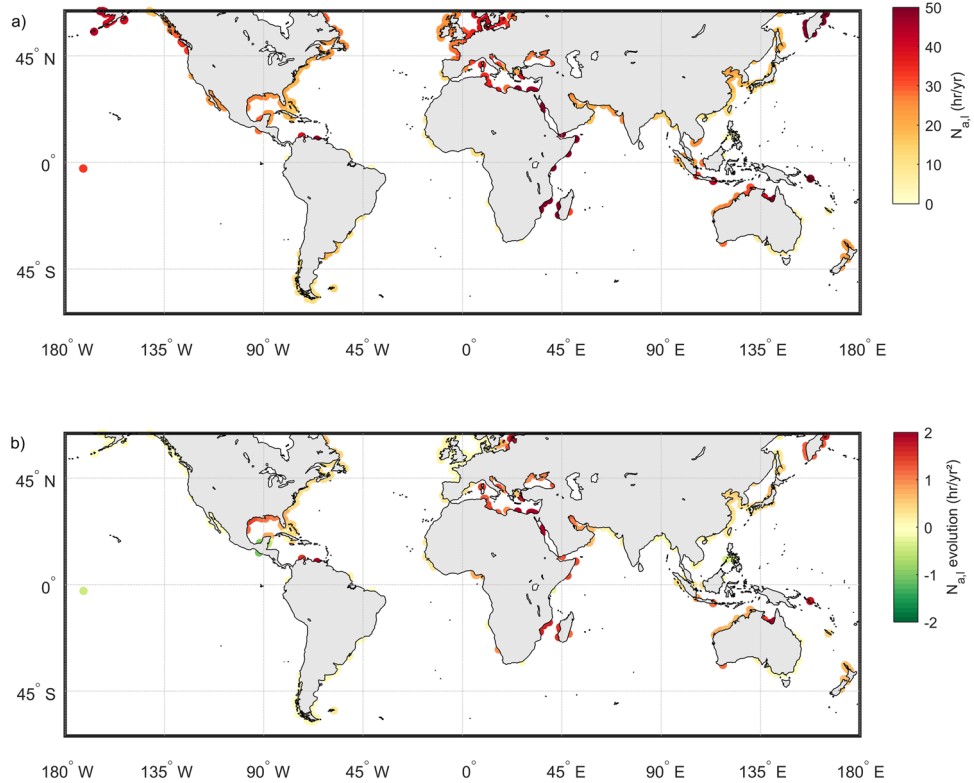

**Fig. 3 Global distribution of coastal overtopping over the period 1993–2015. a** Time-averaged annual number of overtopping hours (h/year) and **b** the 23-year trend in the annual number of overtopping hours ($N_{a,l}$) (computed using the complete hourly time series of ECWL, discretized into years) (see "Methods" for details of the computational approach adopted). Overtopping is assumed to occur when ECWL exceeds the maximum coastal elevation (derived from AW2D30). ECWL = SLA + DAC + T + R is computed by combining hourly data of all contributing components over the 1993–2015 period. For clarity, locations for which close to zero overtopping was computed are not shown in this figure (in contrast with Fig. 2 where all computational points are shown).

In a globally aggregated sense, overtopping events are mostly due to a combination of wave runup and tides over the 1993–2015 period (Fig. 4a). When wave runup and tides are not accounted for (orange bars), the globally aggregated annual number of overtopping hours is much less than when all components contributing to ECWL are considered (gray bars). When all components of ECWL are accounted for, an increasing (significant at 95% level, using the Mann–Kendall test) trend is found for $N_{a,g}$ over the 1993–2015 period (Fig. 4a), which has resulted in approximately a factor 1.5 increase in $N_{a,g}$ from 1993 to 2015. However, this increasing trend cannot be explained by the combined contributions of tides and runup alone (Fig. 4a, blue bars). Over a short period such as the 23 years analyzed here, local trends in individual ECWL components (except for SLA at some locations), and consequent overtopping, are mostly indistinguishable from trends induced by internal climate variability[18,34]. However, since $N_{a,g}$ is a globally aggregated quantity, it is likely that the signature of internal climate variability is damped by the spatial aggregation compared to the globally coherent signature of SLR. Therefore, the observed increasing trend in $N_{a,g}$ over the 1993–2015 period can only be confidently attributed to the increasing trend in SLA.

**Overtopping and projected SLR.** How will projected SLR influence coastal overtopping characteristics over the twenty-first century? To answer this question, here we considered GMSLR trajectories projected for the twenty-first century under different climate change scenarios (i.e., representative concentration pathway (RCP) 8.5—high emission, low mitigation; RCP 2.6—low emissions, high mitigation; and RCP 4.5—middle of the

road[35,36]) to compute future ECWL time series, keeping the other contributions (R, T, and DAC) unchanged from the 1993–2015 period. Figure 4b shows that, in a globally aggregated sense, if wave and tide contributions are not considered in ECWL computations (orange bars), $N_{a,g}$ by 2100 would be underestimated by over 80%. Figure 4b also shows that, when the wave and tide contributions are included in the computation (gray bars), a discernible increase in $N_{a,g}$ is projected to occur around mid-century regardless of climate scenario, as indicated by the upward inflection around mid-century. In contrast, if wave and tide contributions to overtopping were ignored, a noticeable increase in overtopping hours is only expected by around 2080 and only under RCP 8.5. Figure 4b shows that, relative to its present-day value, $N_{a,g}$ could be as much as 50 times larger by the end of the twenty-first century under RCP 8.5. Since the occurrence of overtopping is based on the exceedance of a topographic threshold (considered to be static here), a non-linear relationship exists between the future increase in ECWL, mainly due to the rate of GMSLR and the rate of change in $N_{a,g}$ (Fig. 4.b, inset). This is because the threshold elevation for overtopping is exceeded more often with accelerated SLR due to greater water depths and more wave energy reaching the coast, leading to a faster increase in the rate of change in $N_{a,g}$, compared to the rate of change of sea level itself. Not surprisingly, therefore Fig. S1 shows that the number of regions around the world that are exposed to overtopping increases non-linearly with increasing global mean sea level.

**Limitations and way forward.** Being a global-scale assessment, inevitably there will be several limitations if attempts are made to

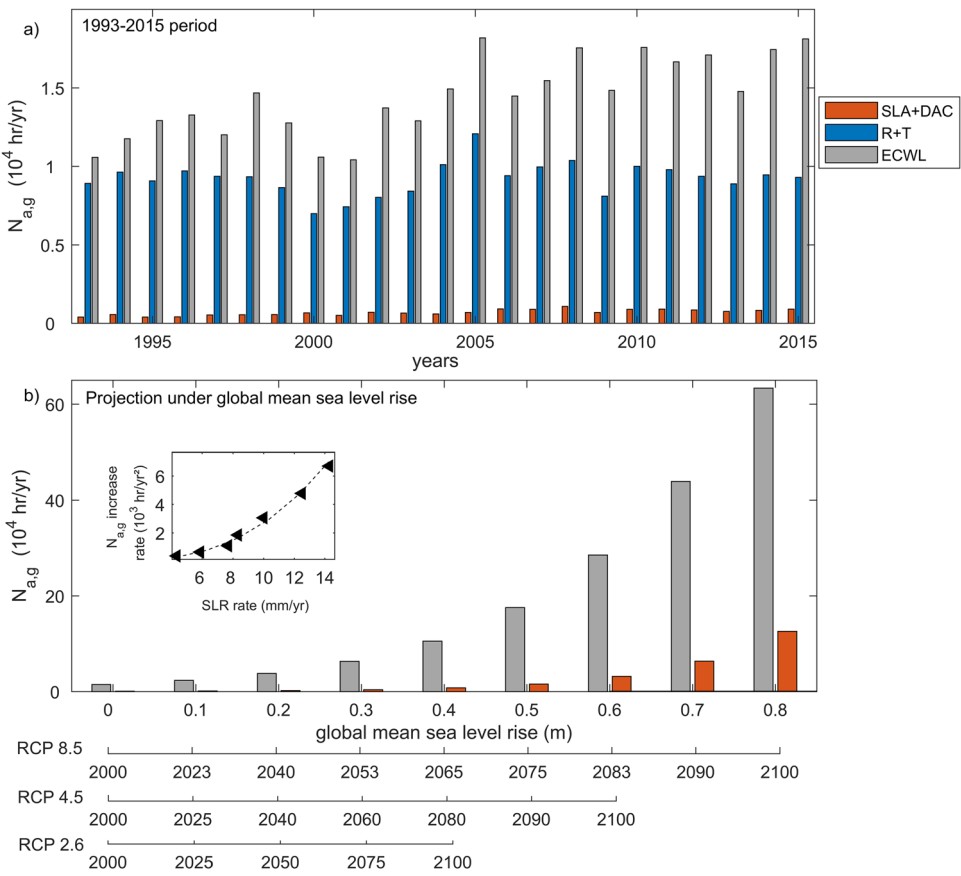

**Fig. 4 Globally aggregated annual number of present and future overtopping hours. a** The globally aggregated annual number of overtopping hours ($N_{a,g}$) (gray bars), contribution to overtopping from waves and tide only ($R + T$, blue bars), and the contribution to overtopping from regional sea level rise and storm surge only (e.g., SLA + DAC, orange bars). **b** Future projections of $N_{a,g}$ when contribution to overtopping from waves and tide ($R + T$) are included (gray bars) or excluded (orange bars) considering projected median global mean sea level rise. The secondary $x$-axes at the bottom indicate the years at which time the various median global mean sea level rise values from the main $x$-axis will be reached under RCP 8.5, RCP 4.5, and RCP 2.6 projections presented in IPCC, 2019[3]. Inset in **b** compares different rates of changes of global mean sea level (in mm/year) with the computed rate of change of $N_{a,g}$. Triangles represent computed $N_{a,g}$ values and the dashed line is an exponential regression ($R^2 = 0.8$) fitted to the triangles, indicating an exponential factor 2.7 between the rates of change global mean sea level and $N_{a,g}$.

interpret our results at the local scale. One of the main limitations would be due to the different impacts waves will have on different types of coasts (e.g., deltas and sheltered coasts vs open coasts). Recent studies have shown that waves might have a complex influence on flooding at tidal inlets and estuaries, and particularly at large deltas, in combination with local hydrology and other sea level contributions derived from met-ocean forcing[37–40]. Local precipitation or river discharge can lead to compound flood events when they occur concurrently with storm surge events and/or large wave runup events[41–45]. These additional factors could not be taken into account in our analysis due to the lack of suitable datasets at a global scale. Furthermore, when interpreting the consequences of overtopping on coastal flooding, it should be noted that the occurrence of overtopping does not necessarily imply that the entire low elevation coastal zone is flooded. Rather, overtopping drives localized coastal flooding, immediately adjacent to areas of overtopping, which would likely be both temporally and spatially variable due to the combined effects of temporal and alongshore gradients in breaking wave heights, and alongshore variations in coastal elevation maxima.

Although satellite digital elevation models DEMs) are increasingly used in global/regional coastal flooding studies[11,12,22,46–50], the quantitative accuracy of such assessments would necessarily be a function of the noise and accuracy associated with the DEM used; a limitation also applicable to our study. However, with more and more advanced technologies used in successive missions, the accuracy of satellite DEMs is continually improving, enabling more reliable impact assessments based on satellite DEMs. While a global validation of AW3D30 is beyond the scope of the present study, the regional DEM/light detection and ranging (LIDAR) comparisons shown in Fig. 5 for two different sites in France and The Netherlands show that, among the 6 satellite DEMs tested here (CoastalDEM[51], MERIT, SRTM[52], ASTER[53], TandemX[54], and AW3D30), AW3D30 has the lowest error (average error ~1.5–2 m) with respect to maximum coastal elevation estimations (i.e., the critical parameter in terms of overtopping). It should, however, be noted that the results of global scale studies cannot be expected to capture intricate local scale details. Nevertheless, first-pass global scale studies, such as that presented here, will enable the determination of current trends and the identification of regional hotspots, which may then be investigated further via higher-resolution, local studies at vulnerable locations.

Global-scale coastal overtopping and flooding studies currently face a double observational bottleneck. On one hand, it is currently impossible to observe sea levels right at the coast all along the global coastline and in particular wave contributions to ECWLs. On the other hand, accurate measurements of global coastal morphological evolution and subsidence trends are still to be done despite promising local to regional emerging satellite

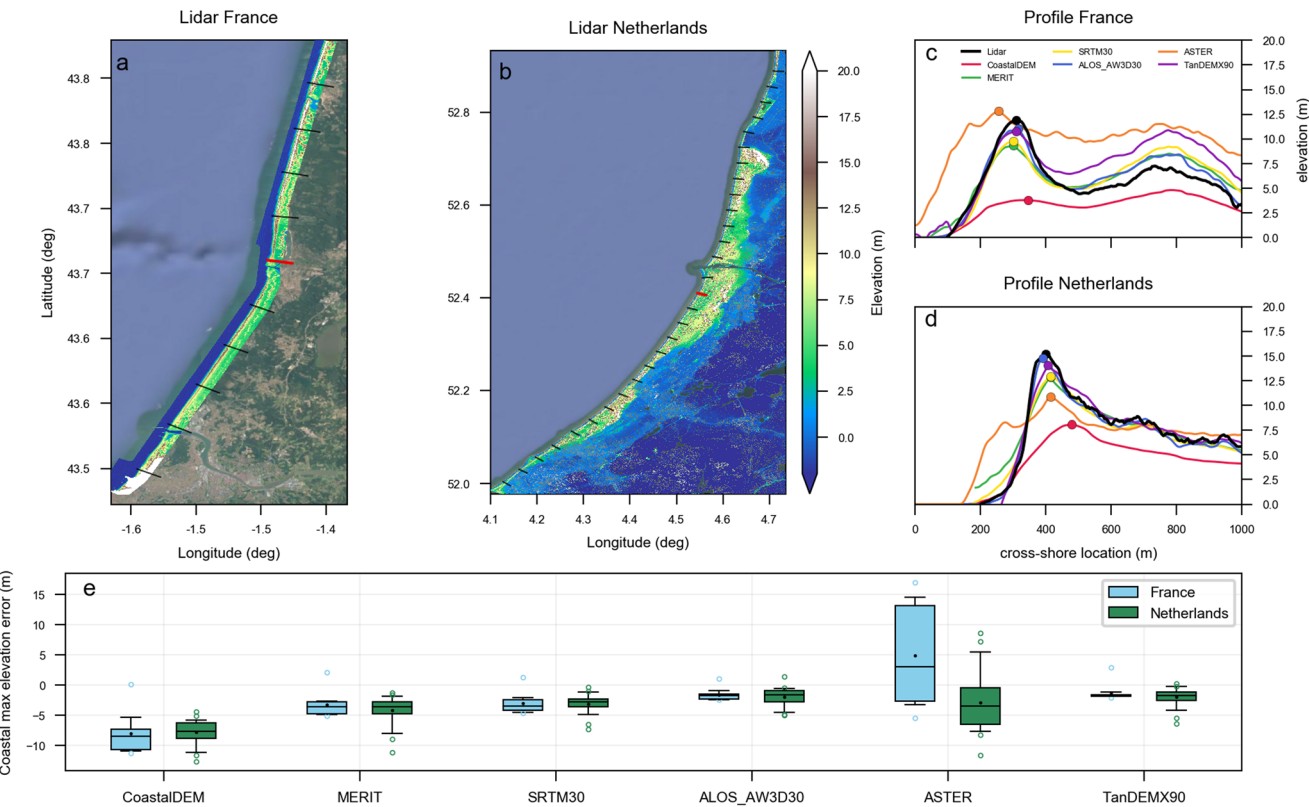

**Fig. 5 Error quantification of coastal maximum elevations derived from six global DEMs (CoastDEM, MERIT, SRTM30, ASTER, TandemX90, and AW3D30), relative to LIDAR data for two regions in France and the Netherlands. a, b**, respectively, show the maps of the local elevation data and the transects (5 km spacing) that were used to calculate the differences in France (10 transects) and the Netherlands (29 transects). **c, d** show the elevation profiles derived for example profiles (red lines in the maps) in France and the Netherlands. Identified coastal maxima for each case are shown by a circle marker. **e** presents boxplots of the coastal maximum elevation differences separately for France and the Netherlands for each global DEM considered. Boxes indicate the 25th–75th percentile range, with a horizontal line and dot showing the median and the mean, respectively. Whiskers indicate the 5th–95th range and circles points that are out of this range.

techniques[55–57]. In the near future, an advanced approach such as that presented by Vos et al.[58] to reconstruct foreshore slope from satellite-derived shoreline tracking and tide level could be applied globally. The global scale of the analysis presented here thus necessitates some simplifications in estimating ECWL, particularly in calculating wave runup using Stockdon et al.'s[32] empirical formulae. Stockdon et al.'s[32] wave runup parametrization was developed for and is applicable for open wave-exposed sandy beaches. For these sandy coasts and beaches, as a rule of thumb, the wave setup is 20% of offshore wave height[32,59,60]. These formulae are, however, commonly used for different environments, such as gravel beaches[61]. At rocky coasts with rocky platforms, wave runup is important but reduced by bottom friction over the rocky bottom[62]. At muddy coasts like the Amazon River to Guyana in South America, high suspended sediment concentrations tend to dampen wave action (and hence overtopping)[63]. The indiscriminate application of Stockdon et al.'s[32] formulations at these latter two types of coasts may have therefore resulted in overestimations in wave runup therein and hence overtopping in our analysis.

Concerning storm surge estimates used in this study, the relatively coarse resolution of the barotropic model (MOG-2D) used and the known inability of ERA-interim to capture extreme wind events mean that our analysis would not account for tropical cyclones[64] in our estimates of extreme storm surge events.

Regarding globally aggregated projections of future coastal overtopping computed in this study, non-linear interactions

between SLR and other contributing components (tides, waves, storm surge) have not been accounted for. Furthermore, climate change-driven variations in storm surge and waves have not been accounted for in this study.

The nearshore topography was considered constant in time here with no morphodynamic evolution, which means that possible SLR-driven changes in the coastal slope and maximum coastal elevation are neglected in our analysis. Even if detailed present-day bathymetry were available, past and future bathymetry still remain unknown. As a result, this and other recent global studies use a fixed coastal bathymetry in time, over periods spanning between 50 and a 100 years. However, coastal systems are among the most dynamic natural environments on Earth[65,66], continually evolving at various spatiotemporal scales with, e.g., a single large storm being able to reshape regional bathymetry that could significantly affect ECWLs in subsequent years. Thus, considering a passive coastal bathymetry over a 100-year period assumes that computed coastal flooding is only a function of changes in coastal water levels[67,68]. It should also be noted that local vertical land movement (e.g., land subsidence) can in places (e.g., Jakarta, New Orleans, Ho Chi Minh City) result in relative SLR rates that are far greater than the GMSLR[69,70] considered in all future projections herein. Consideration of these regional and local contributions to relative SLR will affect coastal flooding projections for certain locations, in particular at coastal cities and low-lying deltas[9,71–74].

Finally, coastlines have been modified in various ways by human activities, particularly in urbanized areas in which, for example, ports have been constructed, land has been reclaimed from the ocean[28], seawalls built to combat coastline recession, cliffs stabilized, and groins placed in an attempt to retain a beach fringe and maintain dunes[68]. For example, in the US alone, 14% of the national coastline is estimated to be hardened with engineering structures, and this percentage is expected to increase to 33% by 2100[75]. Such human interventions (e.g., seawalls, dikes) to the natural system generally results in steepening of coastal slopes[76], resulting in smaller wave dissipation zones compared to natural coasts, which is not accounted for in this study.

In summary, this study has, for the first time, quantitatively assessed the potential for coastal overtopping at a global scale, both for recent decades and over the twenty-first century, by combining high-resolution coastal topography from recently developed global satellite-based products with state-of-the-art computations of ECWL (including wave contributions). Our results show an increasing trend in overtopping, resulting in almost a factor 1.5 increase in globally aggregated annual overtopping hours from 1993 to 2015. While overtopping events are mainly due to the combined effect of wave runup astronomical tides, these processes alone do not explain the observed increasing trend in the globally aggregated annual overtopping hours. Rather, it is the combination of regional sea level, storm surge, wave runup, and tide that is responsible for the observed increasing trend in overtopping, with the increasing trend in regional sea level being the main driver. A first-pass assessment of overtopping potential over the twenty-first century, undertaken in a globally aggregated sense under RCP 2.6, RCP 4.5, and RCP 8.5, shows that, relative to its present-day estimate, the globally aggregated annual overtopping hours will increase by as much as 50 times by the end of the twenty-first century under RCP 8.5. These projections also show that the globally aggregated annual overtopping hours will increase at a rate faster than that of the GMSLR itself, following an exponential relationship (with an exponential factor of 2.7) between rates of overtopping and SLR. The acceleration in coastal overtopping is expected to continue throughout the twenty-first century and will be discernible by mid-century under any climate scenario. Projections indicate that more and more regions around the world will become exposed to coastal overtopping with increasing mean sea level, especially in the Tropics, Northwestern USA, Scandinavia, and Far-Eastern Russia.

## Methods

**AW3D30 Global Digital Surface Model.** The new and freely available ALOS Global Digital Surface Model (ALOS World 3D—30 m, JAXA[24,25]), known as AW3D30, was used in this study. This database is used here with its maximum freely available resolution of 1 arc-second (i.e., approximately 30 m, while commercial AW3D PRISM resolution is 5 m). AW3D30 was acquired over the 2006–2011 period using optical stereo-based photogrammetry and is created as a digital surface model converted from the WGS84/GRS80 ellipsoid height based on the ITRF97 coordinate system, using the EGM96 geoid model. Marine ECWL and land topography datasets used here are referenced to the same datum. Our analysis is restricted to the coverage of AW3D30, from 60 degrees north to 60 degrees south. High latitudes associated with no-data or low-quality area are not considered in this analysis. Although AW3D30 targets <5 m absolute accuracy, Tadono et al.[24], and also Fig. 5 (concentrating on two sites in France and The Netherlands), show the accuracy is in fact higher than that for gentle slopes, which is mostly the case in the low-elevation coastal zone (i.e., the focus study area in our study).

A sensitivity analysis of our overtopping estimates was conducted by using two other bathymetric/topographic datasets and is presented in Fig. S4. The first independent dataset is a merged product of GEBCO and MERIT[77]. It was used in our sensitivity analysis to compute estimates of maximum sub-aerial coastal elevations (Fig. S3) and two different coastal slopes (Fig. S2): the coastal slope from the shoreline to the maximum sub-aerial coastal elevation, as with AW3D30, and the foreshore slopes (computed from the depth-of-closure to the shore, see Athanasiou et al.[77]) that are required as input for the wave runup formulae. The second independent dataset is the FLOod PROtection Standards FLOPROS[78]

dataset. It was used as a third estimate of maximum sub-aerial coastal elevations[22] and explicitly accounts for artificial coastal protection structures.

**Coastal topography extraction.** Maximum sub-aerial coastal elevation and coastal slopes were extracted from the above-mentioned MERIT, FLOPROS, and AW3D30 datasets along the global coastline. Here the Global Self-consistent, Hierarchical, High-resolution Geography Database (GSHHS[79]) coastline "h" highest resolution (~kilometric) was used. The coastal shoreline and topography are highly variable alongshore. In order to obtain reasonably robust estimates, cross-shore aerial topography profiles were extracted every 0.05°. From these, a robust regional profile is constructed every 0.5° alongshore by averaging ten 0.05° spaced transects. This means that our analysis and conclusions are representative of main regional topographical features (e.g., typical low-lying beach-dune, high cliff coastline) but not of local features (e.g., estuaries). Furthermore, Islands with a circumference <0.5° were excluded from the analysis, as we deemed it sufficient at a global scale and representative of the regional values seen in the literature. This resulted in a total of 14,140 profiles along the open coasts of the world, for which the analysis was performed.

The maximum coastal elevation and coastal slope at each profile were calculated using an automated detection method. In this method, the first step is the identification of the local sea-land orientation of each profile, based on the average topography values on the two sides of the shoreline: the higher side is taken to be land and the lower side to be sea. Second, the highest coastal point of each transect (e.g., dune, cliff top, crest of structure) was taken as the local landward maximum that was closest to the shoreline, within 1 km landward of the shoreline (see Fig. 5). The slope used in the wave contribution calculations was then estimated as the average slope between the shoreline and maximum coastal elevation, following the approach presented by Diaz et al.[80]. Figure 5 shows the performance of six different satellite DEMs (CoastDEM, MERIT, SRTM, ASTER, TandemX, and AW3D30) relative to airborne LIDAR data at two low-lying coastal regions: (a) the open coast of the Netherlands, which covers the largest part of the North and South Holland provinces (acquired for the entire country over the 2014–2019 period) and (b) the South West coast of France (acquired in October 2017). Despite the fact that the AW3D30 DEM is based on composite imagery acquired over the period 2006–2011, different to the LIDAR dates, it is still the DEM with lowest error (average error ~1.5–2 m) among the six DEMs considered, in terms of maximum coastal elevation estimations—the critical parameter where coastal overtopping is concerned.

**Components of sea level at the coast.** Altimetry-based sea-level time series anomalies (SLA in Eq. (1), with reference the ellipsoid—WGS84/GRS80) are extracted at the closest points to the coast from the gridded daily maps produced by the SSALTO/DUACS multi-mission[29] and distributed by the Copernicus Marine Environment Monitoring Service[81]. Atmospheric variables (surface winds, sea level pressure) and wave data (significant wave height $H_s$ and peak wave period $T_p$) were extracted from the ERA-interim reanalysis[31], developed by the European Centre for Medium-Range Weather Forecasts model (ECMWF), at 0.5° × 0.5° and 6 hourly temporal resolution between 1993 and 2015. The ERA-interim reanalysis uses a coupled ocean wind-wave and atmospheric model, which has been extensively validated[31,82,83]. Storm surge (DAC in Eq. (1)) time series were extracted for the same period from the MOG2D-D barotropic model forced by ERA-interim surface winds and atmospheric pressure with 6 hourly outputs. Astronomical tidal elevations (T in Eq. (1)) for the 1993–2015 period were obtained at the closest points to the coast from the global tide model FES[30] at an hourly resolution. Wave runup (R in Eq. (1)) was computed from the commonly used parametrization by Stockdon et al.[32], where R is given as a function of deep-water significant wave height $H_s$, wave length ($L_o$), and sub-aerial coastal slope ($\beta$). Here Stockdon et al.'s[32] parametrization was used in two forms depending on the ratio between the coastal slope and incident waves as described by the Iribarren number $\xi = \tan(\beta)/(H_s/L_o)$[84]:

- Equation (2) at coasts with $\xi < 0.3$:

$$R = 0.043\sqrt{H_s L_o} \tag{2}$$

- Equation (3) at coasts with $\xi > 0.3$:

$$R = 1.1\left(0.35\beta\sqrt{H_s L_o} + 0.5\left[H_s L_p\left(0.5625\beta^2 + 0.004\right)\right]^{1/2}\right) \tag{3}$$

R can be predicted using different methodologies, such as direct numerical modeling with process-based local coastal models, meta-models, and empirical formulations (e.g., Dodet et al.[59]). In Melet et al.'s[21] discussion on the limitations of using Stockdon et al.'s[32] parametrization at global scale, it is highlighted that process-based coastal models also need local nearshore profiles as inputs and cannot yet simulate R with nearshore morphological updating over long timescales and along the entire global coastline. R is therefore commonly predicted via empirical formulations that relate it to a set of simple environmental parameters (see review by Dodet et al.[59]). As this study aims at providing a first-order estimate, R is here computed using empirical formulae. For instance, Diaz-Sanchez et al.[85] mention that the scatter between empirically predicted and observed R can be due to local processes that are not represented by the formulations' predictors.

The automated computation procedures used in this study ensures that Eq. (2) is used on natural beaches with milder slopes, while Eq. (3) is used at steep profiles, such as, for instance, when coastal defense structures are present.

All the components feeding into Eq. (1) are ultimately interpolated to an hourly resolution to account for compound nature of ECWL.

**Method to compute overtopping**. Using the above described hourly datasets, the different contributions to ECWL were calculated hourly over the 1993–2015 period. Potential overtopping is defined to occur when the ECWL thus computed exceeds the maximum coastal elevation. To temporally aggregate the event-level information, the number of hours of potential overtopping occurrences is counted at each computational point for every year in the 1993–2015 period. A sensitivity analysis of the overtopping projections to the choice of the topography dataset (i.e., AW3D30, MERIT, FLOPROS) was conducted and the results are shown in Fig. S4. Figure S4 shows that using FLOPROS or MERIT in the computations leads to higher overtopping rates, which is a direct result of the lower estimates of maximum sub-aerial coastal elevation compared in these two datasets, compared to AW3D30 (Fig. S3).

## Data availability

The SSALTO/DUACS altimeter products were produced and distributed by the Copernicus Marine Environment Monitoring Service (http://marine.copernicus.eu/). Dynamical atmospheric corrections were produced by the Collecte Localisation Satellites Space Oceanography Division using the MOG2D model from Laboratoire d'Etudes en Géophysique et Océanographie Spatiales (LEGOS) and distributed by AVISO (Archiving, Validation and Interpretation of Satellite Oceanographic data), with support from Centre National d'Etudes Spatiales (CNES) (http://www.aviso.altimetry.fr/). FES2014 tidal data are produced by LEGOS. Tide gauge data were downloaded from the University of Hawaii Sea Level Center (https://uhslc.soest.hawaii.edu/data). ERA-Interim data were produced by the European Centre for Medium-Range Weather Forecasts (https://www.ecmwf.int/en/forecasts/datasets/reanalysis-datasets/era-interim). LIDAR data in Fig. 5 were obtained from the Observatoire de la Côte Aquitaine (OCA). Coastal topographical and protection products, AW3D30, MERIT, SRTM30, ASTER, TanDEM-X90 and FLOPROS, are also freely available. CoastalDEM is distributed by Climate Central under a non-commercial license. Relevant data are available on request from the authors.

## Code availability

The code that supported the findings of this study is available from the corresponding author upon reasonable request.

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

## Acknowledgements

R.R. is partly supported by the AXA Research fund. P.A. is supported by the EU Horizon 2020 Programme for Research and Innovation under grant no. 776613 (EUCP: European Climate Prediction system).

## Author contributions

R.A. conceived the study together with R.R. E.W.J.B., P.A., O.D., and L.P.A. participated in the early-stage developments and ideas. E.K. extracted waves dataset. A.M. and F.P. contributed to the manuscript at different stages. P.A. produced the Fig. 5 and AW3D30 comparison with DEMs. H.D. produced the extreme coastal sea level dataset and computed overtopping statistics. M.V. added the FLOPROS dataset to the study, with all authors discussing results and implications and commenting on the manuscript and responses to reviewer comments.

## Competing interests

The authors declare no competing interests.
