## [Peer Review File · Nature Communications]

REVIEWER COMMENTS

Reviewer #1 (Remarks to the Author):

General Comments:

Almar et al. presents a global assessment of future overtopping vulnerability using calculated water level variations at the coast derived from a combination of satellite altimetry, tide and surge models and wave reanalyses. Beach topography is derived from AW3D30, a global DEM. The content of the paper is potentially of great interest to Nature Communications readership, however several fundamental weaknesses limit my enthusiasm for publication. This paper contains numerous misspellings and grammatical errors, canonical wave overtopping studies are absent, datums are not described and beach topography is derived from a product with root mean square elevation errors of ~5 m. The authors make no attempt to consider the uncertainty in predictions which could fundamentally alter the manuscript's conclusions.

From a general perspective the work is framed against global studies. The authors do not investigate wave overtopping modeling and prediction literature (e.g., LeRoy et al., 2015, Gallien et al., 2016, Eurotop, 2018). Overtopping decreases with freeboard, accurate beach elevations are required to calculate freeboard and therefore overtopping. AW3D30 vertical elevation errors are approximately 5 m. The lack of consideration for overtopping uncertainty given the large topographic errors are concerning and could impact the conclusions of the paper.

The authors use $TWL = SLA + DAC + AT + R$ where SLA represents sea level changes, DAC storm surge, AT tide and R wave runup. However the validation is inconsistent and highlights the critical nature of correct back beach elevations. In the manuscript the Xynthia event (Figure 1a) which shows tide and surge and waves (thick black line) seem inconsistent. If $H_s \sim 7m$, $T_p \sim 7-10 s$, for Xynthia at La Pallice (Bertin et al., 2013) the Stockdon $R2\%$ would be 2-3 m on a 1:10 beach. I'd expect to see a 2-3 m offset from the water level in Figure 1. In the case of Xynthia, if the water level was ~4 m (tide + surge) and the waves added 2-3m then the total water level would be ~6-7 m. The error in the DEM used is ~5 m VRMS. Essentially the same order of TWL. A rigorous treatment of the impact of dem uncertainty is critical to developing accurate conclusions.

Figure 1, adapted from Almar et al., in review. (a) Xynthia event, (b) Katrina event.

Fig. 2. Predicted tide (blue line), observed water level at the La Pallice tide gauge (black circles) and modeled water level from Bertin et al. (2012a) storm surge modeling system (red line), in meter NGF, during the Xynthia storm.

Adapted from Nat. Hazards Earth Syst. Sci., 13, 1595–1612, 2013
<https://doi.org/10.5194/nhess-13-1595-2013> under a CC BY 3.0 license (<https://creativecommons.org/licenses/by/3.0/>)

Figure 2, adapted from Breilh et al., 2013.

The authors do not discuss datums (e.g., NGF, OD, NAVD88) and instead use the term ‘elevation’. In wave runup and overtopping where the land and water are linked the datum is critical. Datum errors may fundamentally alter results. The authors need to include explanations of the datums used.

Figure quality is poor by Nature standards. Particularly figure 4. Figure 3 coloring is difficult to read. See specific comment below.

This study does not explicitly consider overtopping. Stockdon runup is used as proxy for overtopping, that is if $R2\% > \text{beach elevation}$ then it is considered overtopping. Although this is a reasonable proxy, strictly speaking it is not overtopping and as such the title and assertions are somewhat misleading. This is minor compared to the above comments.

Specific items:

L43: What is EWL? Should this be TWL?

L63 – misspelling reanalyses

L75-75 – units? Beach grain size is critical to beach slope, infiltration processes and wave runup. If the authors present grain size units are critical to determine if it is sand, gravel, rock....

L90 – misspelling subaeral

L147 – figure 4 is low quality and inconsistent with typical Nature figures

L164 – misspelling pojections

L170 – “We demonstrate that overtopping events are in fact mainly due to the combined effect of large wave runup events and high astronomical tides.” This has been well demonstrated in literature previously.

L183 – wen thet?

L184 – Gallien et al., 2019 should be added as are reference

L189 – reference order is incorrect

L191 – misspelling beaddresses

L201 – and this is a fundamental shortcoming

L266 – misspelling ration

L275 – misspelling autmuted

Figure 3, the color scheme makes it very difficult to understand. It almost appears that overtopping is decreasing for much of the coast. A color scheme that has a clear zero color (the blue/green is undistinguishable from the blue) would be helpful. This also applies to the supplementary figures.

Specific Nature Review Format:

What are the major claims of the paper? The authors investigate overtopping changes with sea level rise and suggest overtopping will increase 50 fold.

Are the claims novel? yes

Will the paper be of interest to others in the field? yes

Will the paper influence thinking in the field? potentially

Are the claims convincing? If not, what further evidence is needed? Conclusions are unconvincing. The impacts of a large elevation uncertainty (~5 m vrms) are ignored, which could alter conclusions.

Are there other experiments that would strengthen the paper further? How much would they improve it, and how difficult are they likely to be? Contextualizing the impacts of elevation/freeboard uncertainty would significantly improve the manuscript. However, it is likely the authors may find they are unable to form conclusions given that water levels and relative topographic vertical uncertainty and similar in magnitude.

Are the claims appropriately discussed in the context of previous literature? Wave overtopping literature is ignored. This may be why authors missed the critical nature of beach elevation (and consequently freeboard) and the significant, order one impact of vertical uncertainty.

If the manuscript is unacceptable in its present form, does the study seem sufficiently promising that the authors should be encouraged to consider a resubmission in the future? The manuscript is not acceptable in its present form.

Is the manuscript clearly written? If not, how could it be made more accessible? Extensive editing is required.

Could the manuscript be shortened to aid communication of the most important findings? The length is appropriate.

Have the authors done themselves justice without overselling their claims?

Have they been fair in their treatment of previous literature? The literature framing is primarily global in nature (which in and of itself is not an issue). However, overtopping and flooding from wave impacts is fundamentally a local process. This mismatch is unexplained.

Have they provided sufficient methodological detail that the experiments could be reproduced? probably

Is the statistical analysis of the data sound? No comment

Should the authors be asked to provide further data or methodological information to help others replicate their work? (Such data might include source code for modelling studies, detailed protocols or mathematical derivations). Datums and how land/water elevations were linked must be explained.

Are there any special ethical concerns arising from the use of animals or human subjects? Not applicable.

Reviewer #2 (Remarks to the Author):

This paper presents an analysis of the potential for coastal overtopping at the global scale, using a series of simple assumptions about coastal topography and hydrodynamic conditions. The paper is well written and easy to follow although in a number of areas more detail should be offered so the reader isn't simply referred to other references or the methods section. This is a very challenging study to pull off, and the authors have done an admirable job in attempting to do this. However, to pull this off the authors are forced to make a number of simplifying assumptions to look at this issue at the global scale. In doing so there are a number of caveats that need to be discussed in more detail to be able to evaluate the quality and applicability of the conclusions. For example, how is the joint probability between tides and wave driven water levels handled in the analysis? Are coastal defense structures adequately resolved by a 30 m DEM? What about the uncertainty of all these different components, at least in a semi-quantitative sense? What future work could make such a study more robust? e.g., resolving nearshore wave conditions, protected embayments, foreshore slope, future wave climatology, etc.. So while the study is certainly a topic of interest for the readership of NCC, the conclusions need to be presented more cautiously in the context of the limitations resulting from the methods applied at such a massive scale.

Specific comments

Line 30-there seems to be something missing here, such as one and a half times a rate or? 50 times larger, I think you mean overtopping at a rate 50 times greater than the historic past?

Line 53-and also the lack of nearshore wave data or model projections

Line 77-this component of water level increase due to wave set up goes back to Guza and Thornton in JGR (1981) "Wave set-up on a natural beach".

Line 90- more is needed on what the maximum coastal elevation is, not just referring to the methods section here. A simple explanation here should suffice.

Line 93-also, more of an explanation on coastal slope should be added here. For example, the gradient along the west coast of North America is incorrect, in fact coastal slopes decrease from California through the highly dissipative beaches of the Pacific Northwest (see Ruggiero references for beach slope data in Oregon and Washington states) e.g., Ruggiero, P., Kaminsky, G. M., Gelfenbaum, G. & Voigt, B. Seasonal to interannual morphodynamics along a high-energy dissipative littoral cell. *J. Coast. Res.* 21, 553–578 (2005).

Line 103-I understand that many of the details are referred to the methods section, but a brief synopsis of how these overtopping features were extracted would help give the reader confidence for the analysis of the overtopping events

Line 113-how are all these different water level components combined? The timing of the different water level components is certainly a complex factor to consider

Figure 3-there should be more detail on this figure caption. I believe this is based on the 1993 to 2015 ERA interim reanalysis? But this should be clearly stated so figure can stand alone

Line 135-please describe in more detail here the potential drivers of these observed changes, such as regional sea level rise variability as has been well documented in the western tropical Pacific, changes in the wave climate, changes in storminess, etc..

Line 139-by integrated I think you mean a summation of all the profile overtopping hours? This should be made more clear, because it could also be interpreted as an average but the overtopping hours are clearly far too high for that.

Line 145- again are you talking about the rate of global overtopping, i.e. the number of hours per year?

Line 168-171- at this point there has been no description of coastal topography resolution,

nor any indication that high astronomical tides are a major driver of the projected increase in overtopping

Line 183- typos here

Line 186-a bigger question is how well are estuaries resolved in your approach? Are waves allowed to penetrate into large estuaries where most of the mega cities reside, such as London, Bombay, Sydney, New York, etc. this could lead to significant overestimates of overtopping if embayments are not resolved. This is one of the big challenges for looking at global flooding risk due to sea level rise and waves.

Line 212- you should also refer to Manoo Shirzaei's work in Science Advances 2018; Shirzaei, M., Bürgmann, R. (2018), Global climate change and local land subsidence exacerbate inundation risk to the San Francisco Bay Area. Science Advances, 4, <https://doi.org/10.1126/sciadv.aap9234>

Line 212- other caveats to clearly layout and discuss include the resolution of the elevation data, which is not sufficient to resolve many overtopping points, the issue of resolving waves in estuaries and the nearshore, and the coincident timing of high tides and wave events, among others.

Line 226-how is this foreshore slope calculated?

Line 244 – but the slope used in run up equations is typically along the upper shore face around or between MSL and MHW? So if you're using the maximum coastal elevation in that region, which is not necessarily relevant to the run up surface, you could be drastically overestimating the overtopping.

Line to 273-just because Stockdon has been applied to beaches with greater slopes, does not mean that it is valid. Steeper slopes are typically referred to other engineering equations such as TAW:

van der Meer, J. (2002). Technical report wave run-up and wave overtopping at dikes: Delft, Netherlands, Technical Advisory Committee on Flood Defence, 42 p.

that are designed for overtopping of structures, not for wave run up on sandy beaches

Supplementary material

While the analysis of the different global data sets is interesting, a more accurate understanding of the uncertainty of the overtopping projections could be obtained by a higher resolution locally derived survey such as airborne LiDAR or GPS surveys in a given location

Figure S6-the validation is a nice piece, but it seems the validation is based on a projection simply exceeding a given elevation over a vast area so perhaps this is highly simplified? It is hard to discern from these plots the skill of the overtopping analysis that you have conducted. Perhaps just more explanation is required to explain these plots, which seem to be highly pertinent to being able to defend the methods in this manuscript

Reviewer #3 (Remarks to the Author):

Review of Almar et al "How waves are accelerating global coastal overtopping"

Overall this is an interesting and innovative paper which advances the state-of-the-art of the global analysis of wave processes and especially overtopping at the coast. So I commend the authors on pulling it all this information and analysis together. It also demonstrates the complexity of the problem, something which is quite familiar to experts such as coastal scientists and

engineers, but is not well understood beyond this small community due to the multiple and sometimes interacting factors that are involved.

While the analysis has great merits, there are also numerous details that raise questions and require development. Having read the material several times more questions and inconsistencies are apparent and these need to be addressed so the paper is understandable. Below I make suggestions on how the manuscript can be improved firstly via some general comments and then more specific comments referring to specific locations. In some cases the specific comments build on the general comments.

I would recommend publication if the authors can respond adequately to the points raised below.

General comments

Title: "How waves are accelerating global coastal overtopping" – this tends towards a journalist title -- the paper certainly shows evidence of an increase but not an acceleration of overtopping over 23 years of hindcast conditions (Figure 4). Further, the increase in overtopping is driven by mean sea level rise influencing wave transformation to the coast -- so again this makes the title misleading to me – a more neutral, as well as better and more accurate title would be "A global analysis of trends in coastal overtopping". I note acceleration of overtopping is apparent in the simulation of the future but this reflects an accelerating RCP8.5 emission trajectory – sea-level rise under an RCP1.9 or RCP2.6 emissions pathway would more likely show a steady rising, but non-accelerating trend in overtopping (see Oppenheimer et al., 2019 – SROOC Special Report by IPCC).

There is a lack of practical experience of flooding coming through in the manuscript and there seems to be an opinion that overtopping will impact all coastal flooding – for example the mention of deltas. Yes it will impact the seaward edge of deltas and may create new pathways, but the interior flooding in deltas will mainly be driven by river flow, surge, tide and sea level. In narrow flood plains or on low-lying small islands, overtopping the main source of floods. The paper does a bad job of communicating this important context to the research in a nuanced manner and this needs improvement.

The literature is very focussed on post-2010 papers with only two citations from the 20th Century. This partly reflects the recent explosion in literature in this area, but the basic ideas being implemented here have been around and applied more locally for several decades – there is no link to that literature which would give qualitative expectations of the results found. For example, there has been a large amount of analysis on joint probability methods for wave overtopping and water levels to understand defence performance in more developed areas like Europe. Building on this, in places where sea-level rise is part of planning such as in much of north-west Europe, design of new defences often considers these effects i.e. adaptation to increased overtopping is already happening. The novelty of the analysis is the global aspect of the analysis. However, this sub-global experience – which in many ways this global analysis builds upon – should be acknowledged.

The estimate of maximum coastal elevation is not especially clear and yet the outcome of this analysis is absolutely critical to the results that emerge -- how far inland is maximum elevation evaluated? In the examples a few hundred metres, but what are the spatial limits on the algorithm that is applied? In Figure 2 the maximum elevation seem high in many locations such as in NW Europe where there is a large longshore variability in coastal elevation which is simply not expressed? Based on the numbers that I see here I would expect overtopping to be excluded in all cases. And yet in Figure 5, there is overtopping in these areas which surprises me. I would welcome the author's comments on this observation and query.

On reading the paper several times, an additional figure defining a schematic profile and the different types of maximum coastal elevation would make the paper clearer – I suggest to develop such an additional figure.

Figures 3 and 5 -- why is the overtopping data shown along the coast discontinuous compared to earlier Figure 2 where component data is continuous along the coast? As far as I can see, this

change is not mentioned in the text and undermines confidence in the results.

The Discussion needs to focus on insights on overtopping – more general comments on coastal flooding are not appropriate with this dataset.

There is a lot of consideration of beaches. What about highly muddy coasts like the Amazon River to Guyana in South America – on these muddy coasts with high suspended sediment concentrations waves suspend material, but at high concentrations this tends to dampen wave action and hence overtopping as far as I understand. I would welcome the author's comments on this observation and query.

Specific comments

Line 20-22. "This study, for the first time, presents global scale coastal overtopping estimates, which account for not only the effects of sea level rise, storm surge and wave setup as traditionally done, but also that of wave runup and existing coastal protection measures." I think the wave runup is improved much more than coastal protection. No discrimination of the importance and significance of the improvement.

Line 27 "under "business-as-usual" scenario RCP 8.5" – is this a business-as-usual scenario? (see latter comment and reference) – maybe more honestly described as a "high climate change" scenario?

Line 27-28 "the globally integrated number of annual overtopping hours will increase at a rate faster than that of the global mean sea level rise itself." Explain the physics why – linked to greater water depths and more wave energy reaching the coast – an indirect effect of sea-level rise.

Line 30-31 "and will reach values more than 50 times larger by the end of the 21st century." – this sounds like the RCP8.5 scenario is inevitable – this conclusion is conditional on a specific emissions pathway and this should be stated here in the Abstract

Line 34-35 "Coastal flooding is threatening human societies (Hinkel et al., 2014; Vousdoukas et al., 2018) and infrastructures (Koks et al., 2019) and sea level rise is expected to exacerbate the situation in the decades to come." – a bit awkward as an opening – rephrase.

Line 39-41. "In particular, the low-lying coasts of Africa and Asia are thought to be the most vulnerable areas worldwide, at which an increase in flood occurrence could force population migration (Nicholls and Cazenave, 2010)." – forced migration could occur anywhere so better to say – "force significant population migration".

Line 52-53 "mainly due to the lack of global information on detailed coastal topography, knowledge of which is required to compute the wave contributions accurately." Historically lack of computer power has been a big constraint. Earlier assessments often considered waves but very simply to avoid lots of computation.

Line 57 "highly simplified coastal topography/bathymetry" – is it better to say "highly simplified coastal topography/bathymetry assumptions"?

Line 58-60 "While many studies have acknowledged that local topography and foreshore slope can influence flood exposure and risk greatly (Vousdoukas et al., 2018; Luijendijk et al., 2018; Hauer et al., 2020; Minderhoud, 2019; Kulp et al., 2019), no concerted efforts have been taken to address this shortcoming to date." I find this statement too vague – this paper attempts to address a specific aspect of this deficiency concerning waves and overtopping, rather than say improving flood plain topography and inundation. This text should be made more specific to the aims and objectives of this paper.

Line 64-65 "global scale estimates of the acceleration of overtopping in recent decades and under one high-end sea level rise scenario." Change 'acceleration' to 'increase' – and state the timescale

of the scenario analysis.

Line 72 Change "World" to "Global".

Line 73 "The length of the global coastline exceeds 1.6 million kilometers (Burke et al., 2001)" – is this an appropriate length to quote as the estimate includes estuary costs. My understanding is that the analysis in the paper is focussed on the open coast where wave action is important, so is the length of open coast used in the analysis the length that should be cited – I expect this is much shorter than the number from Burke et al (2001). Please review and revise.

Line 74-76 -- 31% sandy beaches translates into more than 500,000 km of sandy beaches worldwide based on the global coastal length of more than 1.6 million km reported in the paper. Is this credible and consistent with the length assumed in Luijendijk et al. (2018) who focussed mainly on the open coast – this links to the point above and they might be addressed together.

Line 74-76 "Among these coastlines, sandy beaches (fine to coarse sand) represent ~31% of ice-free world coasts (Luijendijk et al., 2018, Voudoukas et al., 2020). In general, sandy beach slopes range from 0.01 (for finer sediment) and 0.2 (for gravel beaches) (Poate et al., 2016)." This is confusing as gravel is by definition not sand – are you talking about all beaches regardless of grain size or just sand beaches? -- please revise and clarify.

Line 79-81 "Coastal morphology has been modified in various ways by human activities, particularly in urbanized areas in which, for example ports have been constructed, seawalls built to combat coastline recession, cliffs stabilized, and groins placed in an attempt to retain a beach fringe and maintain dunes (Serafin et al., 2019)." There has also been significant coastal reclamation or land claim – a major process. Add as a process with a source.

Line 84-85 "urbanization and certain land use practices (agriculture, deforestation of mangroves; see Luijendijk et al., 2018; Mentaschi et al., 2018)," – these are not really appropriate references for this point. Please revise with appropriate references.

Line 89-90 "The coastal elevations shown in Figure 2(a) are the maximum subaerial coastal elevations (including dunes and coastal structures)." – the definition of coastal elevation is unclear. Improve Figure 5 to define what this means?

Line 97 "Japan show high variability of slope/elevation within small distances" – I do not see this in either panel – is Figure 2 at the right scale to show this global information?

Line 103-104 "Overtopping occurs when the coastal water level exceeds that of the crest level of natural (e.g. dunes) or artificial (e.g. dykes) coastal defenses." This is an important definition as it is core to the paper and should be stated as the language and definitions vary across the literature. "Overtopping occurs when the coastal water level exceeds that of the crest level of natural (e.g. dunes) or artificial (e.g. dykes) coastal defences and landward flow can occur."

Line 103-104 – how well are coastal defences represented in the dataset – this must be a large uncertainty in the data. Is it even useful to include coastal defences in the analysis with the available data?

Line 107 "In all, Eq. 1 was applied at 14,140 coastal profiles around the world." So one profile covers roughly every 100 km based on the length stated in Burke et al (2001)? The sampling strategy for selecting these profiles needs to be explained and linked to data/methods.

Line 115 "small-scale coastal defences" – so what are we talking about here? Elsewhere a virtue of including defences is made – now defences cannot be resolved. What is small scale. More generally there is a lack of clarity of this aspect of the research.

Line 117-119 "Among these, the low-lying sedimentary plains such as deltas (e.g. Bengal, Nile and Mississippi Deltas for instance, see Nicholls et al., 2007 and Besset et al., 2019) emerge as the areas in the world that are most threatened by episodic coastal flooding." Firstly how important is

overtopping to flooding in these huge and wide delta plains? This research sheds little light on flooding in deltas and this statement demonstrates that the authors have not thought through the implications of their results sufficiently. Further I do not even see a high incidence of overtopping in the Bengal delta. Check and rephrase.

Line 137-138 "Figure 3. Global map of coastal overtopping (number of hours per year): a) occurrence and b) 23 year trend of occurrences (see methods for details on computation approach adopted)." – is a 23 year trend meaningful – when conventionally 30 years of data is required to define a climate variable?

Line 145-146 "Global overtopping has increased almost by 1.5 from 1993." Not especially clear to a reader – try "Global overtopping has increased almost 1.5 times from 1993 to 2016."

Line 148 – Figure 4 -- overtopping without waves – this is sometimes called overflow as it reflects time averaged water levels.

Line 148 – Figure 4 – the scale on the x axis from 2000 to 2100 is wrong – correct.

Line 148 – Figure 4 – why does the (b) panel not include waves only as panel (a)?

Line 158 ("business-as-usual") – correct based on earlier comments

Line 159-160 "Figure 4.b shows that, in a globally aggregated sense, if wave runup were not to be considered in computations, the total annual overtopping hours by 2100 would be underestimated by over 40%." This is a great global quantification – but presented like a completely new result. Qualitatively it is not a surprise to coastal scientists and engineers and it would be better to present it this way – we knew the term was missing and we knew it had a positive sign so the qualitative effect is self evident. You have quantified it, which is an important contribution – but this is not fundamentally new. See for example <https://doi.org/10.1016/j.oceaneng.2016.01.026> and earlier work which inspired it such as Townend (1994). The Townend paper in particular illustrates the relevant previous analysis which this paper fails to cite -- Townend IH (1994) Variation in design conditions in response to sea-level rise. Proc Inst Civil Eng 106(3):205–213.

Line 170-171 "We demonstrate that overtopping events are in fact mainly due to the combined effect of large wave runup events and high astronomical tides." Is this new?

Line 171-173 "However, these contributing processes by themselves do not induce a significant trend in the globally integrated number of annual overtopping hours, rather, it is the combination of regional sea level, wave runup and tide that results in an increase of this quantity." – it is the combination but I would say sea level is the driver – so it is the combination and the fact that one component (SLR) has a strong trend.

Line 173-175 "Thus, our results reaffirm the previously reported (Prime et al., 2016; Serafin et al., 2017) finding that sea-level rise will have a greater impact on 21st century coastal flooding than future changes in wave climate)." – your results are only relevant to flooding due to overtopping and there are other flood mechanisms linked to sea-level rise.

Lines 175-176 "The interaction of sea level and topography increases overtopping events at a rate faster than sea level rise itself with a found exponential factor of 2.7 with SLR." On soft coasts topography would respond to sea-level rise – is this effect considered as it would offset increase in run-up?

Lines 177-178 "Under the RCP 8.5 sea level rise trajectory, the projected acceleration in coastal overtopping should be starting about now and will be clearly discernible by about 2050." This is a good and interesting result. However, as some authors have argued that the RCP8.5 emissions scenario is becoming less likely and lower emissions are much more likely (Hausfather and Peters, 2020, Emissions – the 'business as usual' story is misleading, Nature, 577, 618-620.) Some commentary on this aspect and the possible changes with lower emission scenarios is required.

Line 179-186 – this paragraph is not very clear – what are the relevant insights or limitations from the paper.

Line 197 “accurate modelling” – what does ‘accurate’ mean – it is meaningless without some standard as if you look at modelling in more and more detail you will always find inaccuracies. Review and improve to indicate your standards.

Line 227-228 “To account for artificial coastal protection, the FLOod PROtection Standards FLOPROS (Scussolini et al., 2016) dataset was used as a third estimate of maximum subaerial coastal elevations (Vousdoukas et al., 2018).” How good is this dataset?

Line 229-250 “Coastal topography extraction” The method is not clear in terms of issues such as how far inland elevation is measured and other key factors. Please review and check that others could follow and reproduce your results. Also what is the estimated accuracy of the elevation.

Line 229-250 “Coastal topography extraction” The elevation is assumed to be constant with time – a quite reasonable assumption but this needs to be stated explicitly.

Figure S6. These validations are hard to understand – where are they. For example, Katrina affected much of the coast of Louisiana and Mississippi so where is this – surges and waves varied substantially in space and there is hardly surge in this validation? Expand this part.

References: There are errors or inconsistencies in the references which need a complete check – what is Kulp et al (2019) and Minderhoud (2019) in the text which are not in the references – although there are other similar references. Luijendijk et al., 2018 is missing in the references. Beetham, E. P. & Kench, P. reference incomplete. These are just examples – other errors were noted.

English: A native speaker should review the revised document before it is resubmitted to smooth the language where needed.

We would like to thank the editor and the three reviewers for reading carefully our manuscript and for providing us constructive comments. We have given full attention to all individual comments and revised the manuscript accordingly. Reviewers' feedback and suggestions have resulted in an improved manuscript, which we hope is now suitable for publication. Our detailed responses to each comment are provided below. For clarity, we use the following color code:

reviewer comments

our response

our modifications in the manuscript

GENERAL RESPONSE TO REVIEWER COMMENTS

The main messages conveyed by our paper are: (1) there are several hotspots of coastal overtopping around the world, (2) there has been an increasing trend in globally aggregated annual overtopping hours, resulting in an increase of 50% over the last two decades, and (3) annual overtopping hours will increase at rate faster than that of global mean sea level rise, reaching values that are as much as 50 times larger by the end of the 21st century under RCP 8.5. We have added more discussion highlighting that our global approach, at a 0.5° alongshore resolution, the resolution of the global satellite-based product adopted to derive coastal topography, and the resolution of the global scale ocean forcing used, precludes any conclusions at the local scale. This is a best effort, first-pass global study focusing on regional to global scales at open coasts exposed to coastal flooding induced by overtopping due to extreme coastal water levels (combination of sea level, storm surge, tide and wave runup). Despite the intrinsic errors of the datasets, the epistemic uncertainty, not being directly applicable at muddy coasts, nor inlet and estuaries, our findings on open coasts exposed to waves demonstrate, for the first time at global scale, how overtopping will increase exponentially with sea level rise.

Comments that were common among the Reviewers responses to these comments are provided below first:

- Global-scale analysis vs local scale analysis: amplitude of uncertainties and need for global-scale analysis.

Several reviewer comments refer to local scale processes. We completely acknowledge the need for local scale detailed studies, which we hope will be motivated by our first-pass global scale study. We believe that studies at two scales should be treated separately, but inform and add value to each other. Here we chose to focus specifically on the global scale (it is a first), which imposes some inevitable simplifications in the way Extreme Coastal Water Level (ECWL) and topography can be treated due to scale in focus. We have now provided an exhaustive description of the limitations associated with our study in the manuscript, which we hope helps provide more context to our main messages.

- Uncertainty in topography and sensitivity of overtopping to the topography product adopted.

An exhaustive limitations section has been added to the manuscript. In this new section we clarify the uncertainty associated with the satellite-based products and simplified formulae used, and processes accounted for or not accounted for. In this new section, we also clarify the general types of coasts for which our methodology applies and might not apply. More details have been added in the Data and Methods section

on the workflow followed to derive topography parameters (i.e. coastal slope and maximum sub-aerial coastal elevation) on computational transects. We believed this clarification was necessary, based on the Reviewer comments. A comparison of available satellite-based products is presented in the Supplementary material (S3) which quantitatively illustrates the sensitivity our computed overtopping to the topography product. Finally, in response to several requests by the Reviewers, we have now included a more comprehensive validation of our approach to compute overtopping by comparing our estimates with field measurements obtained during five different historical coastal flooding events from around the world (Supplementary material S4).

- Quality of the manuscript and figures.

We have taken great effort to improve the quality of the text and the figures aligned well with Nature Communication guidelines and quality standards. We hope that the editor and reviewers find the general narrative, argumentation and substantiation of the conclusions to be much improved compared to our initial submission.

- Grammar and spelling mistakes

As part of the comprehensive updating of the manuscript, we have also corrected all grammar and spelling mistakes we found, including those that were pointed out by the reviewers. As the text has been very substantially modified, some phrase where Reviewers had earlier pointed out mistakes, may now be removed from the manuscript. Therefore, we will not be responding individually to the grammar/spelling related reviewer comments.

Below we provide our responses to comments that are specific to each set of Reviewer comments.

REVIEWER COMMENTS

Reviewer #1 :

Almar et al. presents a global assessment of future overtopping vulnerability using calculated water level variations at the coast derived from a combination of satellite altimetry, tide and surge models and wave reanalyses. Beach topography is derived from AW3D30, a global DEM. The content of the paper is potentially of great interest to Nature Communications readership, however several fundamental weaknesses limit my enthusiasm for publication. This paper contains numerous misspellings and grammatical errors, canonical wave overtopping studies are absent, datums are not described and beach topography is derived from a product with root mean square elevation errors of ~5 m. The authors make no attempt to consider the uncertainty in predictions which could fundamentally alter the manuscript's conclusions.

We thank the reviewer for these constructive comments, and we greatly appreciate that he/she finds the paper potentially of great interest for Nature Communications readership.

As described in detail below, we have improved the manuscript following this constructive comments. Throughout the paper, we have corrected all grammatical/spelling errors (essentially we have re-written the paper). We have also added key missing references on overtopping studies (see in responses below). All datums are now fully described and we have now added a comparison of available satellite-based products in the Supplementary material (S3) which quantitatively illustrates the sensitivity our computed overtopping to the topography product.

An exhaustive section on ‘Limitations’ has been added to the manuscript. In this new section we clarify the uncertainty associated with the satellite-based products and simplified formulae used, processes accounted for and not accounted for etc. In this new section, we also clarify the general types of coasts for which our methodology applies and might not apply. While we have now formulated our conclusions more cautiously, we hope that this new Limitations section will further help provide more context to our main messages.

From a general perspective the work is framed against global studies. The authors do not investigate wave overtopping modeling and prediction literature (e.g., LeRoy et al., 2015, Gallien et al., 2016, Eurotop, 2018). Overtopping decreases with freeboard, accurate beach elevations are required to calculate freeboard and therefore overtopping. AW3D30 vertical elevation errors are approximately 5 m. The lack of consideration for overtopping uncertainty given the large topographic errors are concerning and could impact the conclusions of the paper.

We completely agree with the Reviewer's comment that accurate topography is essential for accurate estimations of overtopping, and this is true at both local and global scales. We are fully cognizant of this, especially given the fact that a large part of the co-authors of the present study has a long track record of studies on nearshore hydrodynamics at the local scale. We apologize for failing to cite the seminal works highlighted by the reviewer, and have corrected this now (LeRoy et al., 2015; Gallien et al., 2018; Townend (1994); Eurotop, (2018) have all been cited now in Section Limitations and way forward).

Conducting a global scale analysis as done in the present paper by necessity requires: 1) simplifying assumptions on hydrodynamic processes, 2) approximations on topography estimates and topographical evolution in time, and 3) usage of currently existing and available global datasets, which are often satellite-based, and necessarily less accurate than local in situ measurements. It is in recognition of these constraints that we adopted AW3D30, the latest and state-of-the-art global elevation model currently available, as the topography data set of choice for this study. While a global scale coastal validation of our overtopping estimates derived using AW3D30 is impossible (due to lack of a global data base of measured overtopping), we have attempted to nevertheless maximize the confidence in our estimates by doing the following:

- Comparison of AW3D30 and MERIT derived maximum coastal elevations and coastal slopes needed for our overtopping computations. MERIT is currently considered as a reference topography product, and

is one of the only freely available worldwide topography product. In Figures S4 and S5 (Supplementary material S3), we compared maximum coastal elevations and coastal slopes derived from MERIT and AW3D30 at global scale, using the same methodology of data extraction in both DEMs. Even though the two products show qualitatively consistent regional patterns of maximum coastal elevation and coastal slope in Figures S4, S5, the differences are substantial, with, in general, MERIT tending to underestimate maximum coastal elevations compared to AW3D30. This was however already noted in Diaz et al's. (2019) local validation of AW3D30 and can be attributed to the fact that MERIT generally cannot resolve fine features and tends to smooth them. The difference plot of maximum coastal elevation shown in Figure R1 below indicates that the difference between the two products is less than 5 m at ~ 80 % (and less than 2.5 m at ~ 52%) of our computational points worldwide. A few regions, however, show large differences (e.g. Pacific coast of North America and the South East Asian coast).

- All our overtopping results were recomputed using the independent datasets MERIT and FLOPROS to assess the sensitivity of our results to the topography product used (Supplementary material S3). While substantial differences can be seen, the general conclusions on the spatial and temporal patterns of overtopping are robust. Moreover, the overtopping projections obtained when using AW3D30 for both coastal slopes and maximum coastal elevations lies roughly in the middle of the range of projections obtained with different parameter/data set combinations, providing confidence to undertake our analysis with AW3D30 derived parameters.
- Provided a more detailed description (in the Data and Methods section of the paper) of how this study has used the methods used by Diaz et al (2019) to extract coastal topographical information from AW3D30. Diaz et al (2019) presented a detailed local validation of AW3D30 for a coastal stretch in southwestern France using LiDAR data. This validation has shown that AW3D30 has good skills at retrieving coastal topographies (RMSE around 1 m regionally). A further validation of AW3D30 is provided by Zhang et al., (2019) for the Iberian Peninsula. As the AW3D30 product is quite recent, there are still only a few studies dedicated to validating this product specifically for the coastal zone.

Here we would also like to highlight that, although AW3D30 targets lower than 5 m absolute accuracy, Tadono et al., (2016) shows the accuracy is in fact higher than this for gentle slopes which is mostly the case in the low-elevation coastal zone (i.e. the focus study area in our study). This was also confirmed by Zhang et al., (2019).

Figure R1: The upper panel shows the global map of differences in maximum coastal elevation between two existing available global satellite-based products MERIT and AW3D30 and the lower panel shows the distribution of the difference. Note that the differences shown here are not errors but reflect only the divergence between the two estimates.

The authors use $TWL = SLA + DAC + AT + R$ where SLA represents sea level changes, DAC storm surge, AT tide and R wave runup. However, the validation is inconsistent and highlights the critical nature of correct back beach elevations. In the manuscript the Xynthia event (Figure 1a) which shows tide and surge and waves (thick black line) seem inconsistent. If $H_s \sim 7m$, $T_p \sim 7-10 s$, for Xynthia at La Pallice (Bertin et al., 2013) the Stockdon $R2\%$ would be 2-3 m on a 1:10 beach. I'd expect to see a 2-3 m offset from the water level in Figure 1. In the case of Xynthia, if the water level was $\sim 4 m$ (tide + surge) and the waves added 2-3m then the total water level would be $\sim 6-7 m$. The error in the DEM used is $\sim 5 m$ VRMS. Essentially the same order of TWL. A rigorous treatment of the impact of dem uncertainty is critical to developing accurate conclusions.

We certainly agree with the Reviewer that DEM uncertainty is crucial for the type of analysis we conduct in our study. We acknowledge that, for global DEMs, local noise and uncertainties are generally important and therefore datasets such as AW3D30 need to be used cautiously for detailed, local scale flooding studies, although this is commonly done for locations where local topo surveys do not exist. To address this issue, we

present our analysis and conclusions as being representative of the regional scale. To this end, each regional 0.5° alongshore transect is made more robust by averaging ten 0.05° transects. Therefore, what we use in our analysis are the main regional topographical features (e.g. typical low-lying beach-dune, high cliff coastline) and not the local features together with their complexity (e.g. small inlets), which are smoothed out through the above mentioned averaging procedure. This is now highlighted in the text

Supplementary material section 3 (Lines 95 – 97):

“Our methodology with a 0.5° alongshore resolution (i.e. the resolution of the global satellite-based product adopted to derive coastal topography and the resolution of the global scale ocean forcing used in this analysis) precludes any conclusions at the local scale and hence our analysis should be considered as the first-pass, best-effort assessment. “

And main manuscript (Lines 95 – 97):

“The coastal shoreline and topography are highly variable alongshore. In order to obtain reasonably robust estimates, cross-shore aerial topography profiles were extracted every 0.05 degrees alongshore (see **Figure M1**). From these, a regional profile every 0.5° alongshore was made robust by averaging ten 0.05° transects. This means that our analysis and conclusions are representative of main regional topographical features (e.g. typical low-lying beach-dune, high cliff coastline) but not of local features (e.g. estuaries). Furthermore, islands with a circumference less than 0.5° were excluded from the analysis, as we deemed it sufficient at a global scale and representative of the regional values seen in the literature. This resulted in a total of 14,140 profiles for which the analysis was performed.”

The evaluation of our results against historical cases as described in Fig. S5 (former S7) aims to illustrate how the full methodology adopted in the paper is able to capture observed overtopping and flooding at locations around the world. Due to the above described regional treatment of our computational approach (as is reasonable in a global scale study), the results we have presented in Fig. S5 are not directly comparable with detailed local scale studies such as Bertin et al. (2011). In this specific case of storm Xynthia (Bertin et al., 2013), it is known as one of the most damaging coastal flooding events to have occurred in France over the last few decades. A maximum water level up to 4m (above mean sea level) was observed during the peak of this storm. This is close to the maximum coastal elevation obtained for this region from the two satellite-derived topography global products used in this study - AW3D30 and MERIT – with consistent regional median around 4 m (MERIT coastal maxima threshold have been added to AW3D30 ones in Figure S5 to give to the reader a level of uncertainty). The Xynthia impact area is complex with islands (Il de Re), inlets, sheltered and more open stretches of coasts (i.e. La Faute sur Mer, North of La Rochelle). Due to the regional resolution of our study, here we do not claim to be describing this level complexity, as the detailed representation of such complex local features are outside of the scope of our global scale analysis. Furthermore, to our knowledge, a dyke

failure occurred at La Faute Sur Mer. Such processes cannot be dealt with at global scale as that would require data on coastal defenses and their state at the time of the extreme event.

Reprinted from Ocean Modelling, 42, Bertin et al, Importance of wave age and resonance in stormsurges: The case Xynthia, Bay of Biscay, 16–30, Copyright 2012, with permission from Elsevier

Figure R2: For comparison with global wave product used in our study. Timeseries of wave height (Hs) and peak period (Tp) observed at the close Oleron buoy. Xynthia induced flooding occurred on 2010/02/28. From Figure 6 of Bertin et al., (2012).

Figure R3: a) Timeseries of deep water wave data (ERAinterim, ECMWF) used in our study for the West Europe Atlantic case in Fig S5.a. Wave are taken at the closet node. b) components of the Extreme Coastal Water Level (ECWL), runup (R), storm surge (DAC), regional sea level (SLA), and astronomical tide (T).

As mentioned by the Reviewer and reported by Bertin et al., 2012, waves peaked around $H_s \sim 7\text{m}$, $T_p \sim 7\text{-}10\text{ s}$, for Xynthia at the close Oléron buoy (see Figure R2). In Figure R3.a we show the wave values used in our computations, which reflects 6-hourly regional deep water values of global ERAInterim product. As mentioned by Bertin et al., (2012), waves during Xynthia were not the largest of the storm sequence and for this reason cannot explain alone any reported flooding. Our wave data at the closest node well represents the temporal pattern with the storm sequence, while underestimating H_s , which peaks at 4.8 m during Xynthia with T_p 10.1 s. Our related computed runup was 1.2m. In Figure R3.b, ECWL components are shown, as a complement to Fig S5. The maximum of the timeseries corresponds well with the moment when flooding was observed and reported (Feb 28, 2010). As we detail in the Supplementary material Section 3, it is mainly due to a compound effect of increasing tide, runup, and very large storm surge well captured by the barotropic model (MOG-2D), forced by of ERA-interim.

Also, AW3D30 derived coastal slopes in the Xynthia impact region used in our analysis are about 0.02, in contrast to the beach slope of 0.1 that the reviewer mentions. In a near future, an advanced approach such as that presented by Vos et al. (2020) could be applied globally to obtain more accurate coastal slope estimates from satellite imagery. Vos et al. (2020) estimated beach face slopes from tidal data and satellite derived

shoreline positional changes and reported a median coastal slope of 0.062 / 0.068 for the SE coast of Australia and for California respectively. Again, as mentioned above, our regional scale approach does not aim or claim to be as accurate as a detailed local study could be at that scale.

Nevertheless, we have double checked our calculations in all cases shown in Fig. S5 and can confirm that there are no errors. Please also note that we have now added an exhaustive “Limitations and way forward” section to the paper, which discusses this and other limitations associated with our assessment in a complete transparent way, such that our results and conclusions can be considered within the context of these limitations.

The authors do not discuss datums (e.g., NGF, OD, NAVD88) and instead use the term ‘elevation’. In wave runup and overtopping where the land and water are linked the datum is critical. Datum errors may fundamentally alter results. The authors need to include explanations of the datums used.

We thank the reviewer for pointing this out. This is a particularly important issue when combining sea level with topography. Our marine datum is that of altimetry data. Altimetry gives the surface height above the reference ellipsoid. This height comprises parts of local geoid anomalies (from the reference ellipsoid) and the mean dynamic topography (MDT, due to the mean ocean circulation). SLA are anomalies from the mean sea surface (including geoid and MDT). The mean sea surface is also referenced to the reference ellipsoid - WGS84/GRS80. The AW3D30 product is created as a digital surface model converted from the GRS80 ellipsoid height based on the ITRF97 coordinate system, using the EGM96 geoid model.

Therefore, we ensure that land and water have the same vertical reference. This is now clarified in the manuscript (Lines 303-305):

“Altimetry based sea-level timeseries anomalies (SLA in Eq. 1, with reference the ellipsoid - WGS84/GRS80) were extracted at the closest points to the coast from the gridded daily maps produced by the SSALTO/DUACS multi-mission (Pujol et al., 2016) and distributed by the Copernicus Marine Environment Monitoring Service (Le Traon et al., 2019).”

And for AW3D30 (Lines 258-263):

“The recently released ALOS Global Digital Surface Model (ALOS World 3D - 30m, JAXA - Tadono et al., 2016; Zhang et al., 2019), known as AW3D30, was used in this study. This database is used here with its maximum freely available resolution of 1 arc-second (i.e. approximately 30 m, while commercial AW3D PRISM resolution is 5 m). AW3D30 was acquired over the 2006-2011 period using optical stereo-based photogrammetry, and is created as a digital surface model converted from the WGS84/GRS80 ellipsoid height based on the ITRF97 coordinate system, using the EGM96 geoid model. Marine ECWL and land topography datasets are referenced to the same datum.”

Figure quality is poor by Nature standards. Particularly figure 4. Figure 3 coloring is difficult to read. See specific comment below.

The quality and the content of all the figures (in the paper and in the supplementary material) have been substantially improved. All figures now follow NComm guidelines. We hope they now satisfy the Reviewer and readers' expectations.

This study does not explicitly consider overtopping. Stockdon runup is used as proxy for overtopping, that is if $R2\%$ is $>$ beach elevation then it is considered overtopping. Although this is a reasonable proxy, strictly speaking it is not overtopping and as such the title and assertions are somewhat misleading. This is minor compared to the above comments.

We thank the Reviewer for pointing out this important distinction. In response the title of the manuscript has now been modified to "A global analysis of trends in potential coastal overtopping". Using the compounded total water level, including the runup $R2\%$ estimate, here referred to as Extreme coastal water level, following Gregory et al., (2019), is indeed a reasonable and very commonly used proxy for coastal overtopping.

Specific comments:

Note: As the manuscript has been substantially modified, some of the specific comments pertaining to grammar and spelling may not be applicable anymore and therefore we do not list/respond to such comments below. See also our overall response on Grammar and spelling mistakes at the end of our General response to Reviewer comments section above.

L43: What is EWL? Should this be TWL?

Yes, it has been corrected. We thank the reviewer for noting this typo. We actually decided to adopt the terminology Extreme Coastal Water Level (ECWL), following recent literature (Gregory et al., (2019))

Gregory, J.M., Griffies, S.M., Hughes, C.W. et al. Concepts and Terminology for Sea Level: Mean, Variability and Change, Both Local and Global. *Surv Geophys* 40, 1251–1289 (2019). <https://doi.org/10.1007/s10712-019-09525-z>

L75-75 – units? Beach grain size is critical to beach slope, infiltration processes and wave runup. If the authors present grain size units are critical to determine if it is sand, gravel, rock....

Coastal slope is here used as a ratio therefore has no units. We agree that grain size is crucial to estimate beach slope, but such information is unfortunately not available for the entire global coastline. More extended discussion on the limitation of the methodology used to compute R is provided by Melet et al. (2020).

L184 – Gallien et al., 2019 should be added as are reference

Done, thanks, Gallien et al., 2018 was added (Lines 205-208).

“Recent studies have shown that waves might have a complex influence on flooding at tidal inlets and estuaries, and particularly at large deltas, in combination with local hydrology and other sea level contributions derived from met-ocean forcing (Tazkia et al., 2017; Gallien et al., 2018; Lashley et al., 2019; Le Roy et al., 2015).”

And in the reference list:

Gallien T.W., Kalligeris N., Delisle M.P.C., Tang B.X., Lucey J.T.D., Winters M.A. (2018), Coastal Flood Modeling Challenges in Defended Urban Backshores. *Geosciences* 8, 450, <https://doi.org/10.3390/geosciences8120450>

Figure 3, the color scheme makes it very difficult to understand. It almost appears that overtopping is decreasing for much of the coast. A color scheme that has a clear zero color (the blue/green is undistinguishable from the blue) would be helpful. This also applies to the supplementary figures.

The colormap of the corresponding figures has been modified accordingly, with a close-to-white color for the zero value. More generally, the quality of all figures (in main manuscript and supplementary material) has been substantial improved to meet NCOMM standards.

Reviewer #2 (Remarks to the Author):

This paper presents an analysis of the potential for coastal overtopping at the global scale, using a series of simple assumptions about coastal topography and hydrodynamic conditions. The paper is well written and easy to follow although in a number of areas more detail should be offered so the reader isn't simply referred to other references or the methods section. This is a very challenging study to pull off, and the authors have done an admirable job in attempting to do this. However, to pull this off the authors are forced to make a number of simplifying assumptions to look at this issue at the global scale. In doing so there are a number of caveats that need to be discussed in more detail to be able to evaluate the quality and applicability of the conclusions. For example, how is the joint probability between tides and wave driven water levels handled in the analysis?

We thank the Reviewer 2 for her/his comments and for acknowledging the overall quality of this challenging study.

We do agree that global scale analyses involving wave runup and overtopping (such as ours) necessarily have to have some simplifying limitations. Otherwise an assessment of this scale will simply not be possible. We apologize for not highlighting the limitations of our approach in the previous version. In response to this very valid comment (and similar comments from the other Reviewers) we now transparently address the main limitations of our approach, in a dedicated new section (see below and also the general responses to reviewers at the top of this document).

Regarding the specific comment on the co-occurrence of tides and waves, our historical ocean data are resampled at an hourly resolution which allows the deterministic resolution of the combination of tides with other met-ocean forcing components (such as wave runup, storm surge). For the future period, we make the reasonable assumption that the statistical distributions will not change significantly. Even if this assumption were not to hold for the future, given Vousdoukas et al., (2018) finding that the effect of changes in meteorological extremes on extreme sea levels are minor compared to that of sea level rise, the results will not be significantly different.

Are coastal defense structures adequately resolved by a 30 m DEM? What about the uncertainty of all these different components, at least in a semi-quantitative sense? What future work could make such a study more robust? e.g., resolving nearshore wave conditions, protected embayments, foreshore slope, future wave climatology, etc... So while the study is certainly a topic of interest for the readership of NCC, the conclusions need to be presented more cautiously in the context of the limitations resulting from the methods applied at such a massive scale.

We completely agree with the reviewer and apologize for not highlighting the limitations of our approach in the previous version. In response, we have now added a devoted section on "Limitations" in the manuscript

where we address most of the issues raised by the Reviewers on caveats (see below). We also modified the conclusions, which are now presented more cautiously.

Regarding the Reviewer's specific comment on uncertainty associated with topography data base used, AW3D30 was built with a 5 m RMSE target. However, it is shown by Takaku et al. (2016) that the accuracy increases when the terrain slope decreases and RMSE values as low as 2.49 m are reported for slopes that are less than 10°. Therefore, and given that the focus of our study is limited to generally low-lying coastal topographies, we expect the accuracy level of the AW3D30 to be sufficient to capture at least the main coastal defenses. On the other hand, steeper slopes which are generally linked to high elevation topography (e.g. cliffs), are not so crucial in our analyses as the chances of overtopping at such locations remain low.

Moreover, while AW3D30 has a 30 m spatial resolution, it is derived from a 8 m commercial higher resolution product. As a result, it shows good skills at describing relatively small-scale features such as artificial and natural dunes (see Diaz et al., 2019) when compared with SRTM and WorldDEM (TerraSar/Tandem-X). More importantly, AW3D30 is also less floating in the vertical compared to other products. Coastal slopes computed from relative elevations are therefore less affected than by errors on the absolute error. It should also be noted that our global scale study does not aim to resolving smaller scale barrier breaches or harbor entrances, but rather our focus is on regionally averaged characteristic profiles. The same applies to ocean forcing characteristics: we do not aim here at resolving local, detailed coastal processes and wave transformation such as refraction in small bays.

In our work, most of the uncertainty on overtopping is associated with uncertainties in the topography (coastal slope and maximum coastal elevation) and the consequences on wave runup (i.e. slope). Melet et al., (2020) discussed in detail the uncertainty introduced to global scale assessments of wave runup from beach slope variability and usage of simplifying assumptions. We believe that our study is a takes a significant step forward towards enabling a better anticipation of coastal flooding, including the effect of wave runup, at global scale to inform coastal protection efforts around the world.

Takaku, J., Tadono, T., Tsutsui, K., Ichikawa, M., Validation of "AW3D" GLOBAL DSM Generated from ALOS PRISM III, (2016), pp. 25-31, 10.5194/isprsannals-III-4-25-2016

The new "Limitations" section is as follows (Lines 202 – 254):

"Limitations and way forward

Being a global scale assessment, inevitably there will be several limitations when interpreting our results at local scale. One of the main limitations is due to different impacts waves will have on different types of coasts (e.g deltas and sheltered coasts vs open coasts). Recent studies have shown that waves might have a complex influence on flooding at tidal inlets and estuaries, and particularly at large deltas, in combination with local hydrology and other sea level contributions derived from met-ocean forcing (Tazkia et al., 2017; Gallien et al., 2018; Lashley et al., 2019; Le Roy et al., 2015). Local precipitation or river discharge can lead to compound flood events when they occur concurrently to storm surge events and/or large wave runup events (Brammer,

2014; Ward et al., 2018; Mofstakhari et al. 2017; Bevacqua et al. 2019; Paprotny et al., 2020). These additional factors could not be taken into account in our analysis due to the lack of suitable datasets at the global scale.

Global scale coastal flooding studies currently face a double observational bottleneck. On one hand, it is currently impossible to observe sea levels right at the coast, in particular wave contributions to extreme coastal water levels. On the other hand, accurate measurements of global coastal morphological evolution and subsidence trends is still to be done despite promising local to regional emerging satellite techniques (Benveniste et al., 2019; Melet et al., 2020b). In a near future, an advanced approach such as presented by Vos et al. (2020) to reconstruct foreshore slope from satellite-derived shoreline tracking and tide level, could be applied globally. The global scale of the analysis presented here necessitates some simplifications in estimating ECWL, particularly in calculating wave runup using Stockdon et al.'s (2006) empirical formulae Stockdon et al.'s (2006) wave runup parametrization was developed for and is applicable for open wave-exposed sandy beaches. For these sandy coasts and beaches, as a rule of thumb, the wave setup is 20% of offshore wave height (Guza and Thornton, 1982). It is however commonly used for different environments such as gravel beaches (Poate et al., 2016). At rocky coasts with rocky platforms, wave runup is important but reduced by bottom friction over the rocky bottom (Dodet et al., 2018). At muddy coasts like the Amazon River to Guyana in South America, high suspended sediment concentrations tend to dampen wave action (and hence overtopping) (Winterwerp et al., 2007). The indiscriminate application of Stockdon et al.'s (2006) formulations at these latter two types of coasts may have therefore resulted in over-estimations in wave runup and hence overtopping in our analysis.

With respect to storm surges estimates used in this study, the relatively coarse resolution of the barotropic model (MOG-2D) used and the known inability of ERA-interim to capture extreme wind events, mean that our analysis would not account for hurricanes in our estimates of extreme storm surge events.

Regarding future globally aggregated projections of future coastal overtopping computed in this study, it must be noted that non-linear interactions between sea level rise and other contributing components (tides, waves, storm surge) have not been not accounted for. Furthermore, climate change driven variations in storm surge and waves have were not accounted for in this study.

The nearshore topography was considered constant in time here with no morphodynamic evolution, which means that possible sea level rise driven changes in the coastal slope and maximum coastal elevation are neglected herein. Even if detailed present-day bathymetry were available, past and future bathymetry would still remain unknown. As a result, this and other recent global studies, use a fixed coastal bathymetry over time periods spanning 50 – 100 years. However, coastal systems are among the most dynamic environments on Earth, continually evolving at various spatio-temporal scales, with, for e.g. a single large storm being able to reshape regional bathymetry which could significantly affect extreme coastal water levels in subsequent years. Thus, the consideration of passive coastal bathymetry over a 100-year period necessarily assumes that computed coastal flooding is only a function of changes in coastal water levels (Le Cozannet et al., 2019; Serafin et al., 2017; 2019). It should also be noted that local vertical land movement (e.g. land subsidence) can

in places (e.g. Jakarta, New Orleans, Ho Chi Minh City) result in relative sea level rise rates that are far greater than the global mean sea level rise (Nicholls et al., 2014; Shirzaei and Bürgmann, 2018) considered in all future projections herein. Consideration of these regional and local contributions to relative sea level rise will affect coastal flooding projections for certain locations, in particular at coastal cities and low-lying deltas (Hallegatte et al., 2013; Erkens et al., 2015; Brown and Nicholls, 2015; Kulp and Strauss, 2019; Becker et al., 2020).

Finally, coastlines have been modified in various ways by human activities, particularly in urbanized areas in which, for example ports have been constructed, land has been reclaimed from the ocean (Luijendijk et al., 2018), seawalls built to combat coastline recession, cliffs stabilized, and groins placed in an attempt to retain a beach fringe and maintain dunes (Serafin et al., 2019). For example, in the US alone, 14% of national coastline is estimated to be hardened with engineering structures, and this percentage is expected to increase to 33% by 2100 (see Tavares et al., 2020). Such human interventions to the natural system generally results in steepening of coastal slopes (e.g. seawalls, dikes), resulting in smaller wave dissipation zones compared to natural coasts, which is not accounted for in this study. “

Specific comments

Note: As the manuscript has been substantially modified, some of the specific comments pertaining to grammar and spelling may not be applicable anymore and therefore we do not list/respond to such comments below. See also our overall response on Grammar and spelling mistakes at the end of our General response to Reviewer comments section above.

Line 77-this component of water level increase due to wave set up goes back to Guza and Thornton in JGR (1981) “Wave set-up on an natural beach”.

Yes, we agree and this seminal reference has been added.

Line 90- more is needed on what the maximum coastal elevation is, not just referring to the methods section here. A simple explanation here should suffice.

Thank you for this constructive comment. In response we have now significantly expanded this paragraph as shown below (Lines 78 – 87):

“Depending on the level of detail considered, the total length of global coastlines ranges from 0.5 million kilometers if only open coasts are considered to 1.5 million kilometers, when including bays, estuaries and rugged coasts (Hinkel et al., 2013; Burke et al., 2001). The topology of open coasts is highly variable, comprising open sandy coasts, barrier islands, cliffs, river deltas, and engineered coasts (Schwartz, 2003). Two key coastal topographical parameters that are relevant for coastal overtopping are the foreshore slope, which influences wave runup and thus ECWL, and the maximum sub-aerial coastal elevation, which sets the threshold that is to be exceeded by ECWL for overtopping to occur. The global distribution of the maximum sub-aerial coastal protection elevations (within 1 km landward of the shoreline) derived from the AW3D30 data base (see Data

and Methods and Figure M1 for an illustration of the transect extraction) is shown in Figure 2.a. These maximum coastal elevations also take into account coastal dunes and coastal structures if resolved by the 30 m resolution of AW3D30 data base.”

Line 93-also, more of an explanation on coastal slope should be added here. For example, the gradient along the west coast of North America is incorrect, in fact coastal slopes decrease from California through the highly dissipative beaches of the Pacific Northwest (see Ruggiero references for beach slope data in Oregon and Washington states) e.g.,

Ruggiero, P., Kaminsky, G. M., Gelfenbaum, G. & Voigt, B. Seasonal to interannual morphodynamics along a high-energy dissipative littoral cell. J. Coast. Res. 21, 553–578 (2005).

We thank the reviewer for this comment and have now clarified this point in the text accordingly. From our results and also several recent other regional validations of satellite-based DEM, it appears unrealistic to use such global satellite-based DEM to retrieve intertidal beach slopes. New methodologies for retrieving satellite derived intertidal beach slope estimates (e.g. Vos et al., 2020) could be used for such efforts in a near future. This mainly fails with MERIT, but also WorldDEM and ALOS/AW3D30 (see an inter-comparison around Spanish coasts in Zhang et al., 2019), and even with sub-metric very high resolution on-demand satellite data (Almeida et al., 2019). This limitation is not only due to hydrodynamic (high tide and swash covering the intertidal zone) issues and satellite resolution, but due to the lack of texture issues related to solving photogrammetric optical stereo reconstruction. Going back to the US pacific coast, while beach slope actually increases from north to south (linked to wave energy and grain size), the coastal slope as defined in our study (which excludes the intertidal beach) decreases. This is due to the coastal landscape which changes from sounds and rocky coasts in the north to flatter sandy beaches in the south.

We have now these issues throughout the manuscript and more details have been added to Figure M1 as shown below:

Figure M1. Regional AW3D30 topography in southwestern France compared to airborne LIDAR measurements that serves as a reference topography (from Diaz et al., 2019). Black and red segments represent cross shore profile examples at fine and coarse resolution, respectively. The insert shows LIDAR data and different cross-shore profiles (s1, s2 and s3 transects). Purple and blue lines in the insets represent the LIDAR and AW3D30 cross-shore transects, respectively. Markers are points used to compute the coastal slope (average of slopes between successive points from the shoreline to the maximum subaerial coastal elevation, indicated by a solid triangle).

A first brief explanation is given (Lines 88-90):

“The coastal slope (Figure 2.b) relevant for wave runup calculations is computed from the shoreline to the maximum sub-aerial coastal elevation as derived in Figure 2.a (see Data and Methods and Figure M1).”

New text reads in Data and method section (Lines 288-294):

“The maximum coastal elevation and coastal slope at each profile were calculated using an automated detection method. In this method, the first step is the identification of the local sea-land orientation of each profile, based on the average topography values on the two sides of the shoreline: the higher side is taken to be land and the lower side to be sea. Second, the highest coastal point of each transect (e.g. dune, cliff top, crest of structure) was taken as the local landward maximum that was closest to shoreline, within 1 km landward of the shoreline (see Figure M1). The slope used in the wave contribution calculations was then estimated as the average slope between the shoreline and maximum coastal elevation, following the method presented by Diaz et al. (2019) – see insert in Figure M1.”

Line 103-I understand that many of the details are referred to the methods section, but a brief synopsis of how these overtopping features were extracted would help give the reader confidence for the analysis of the overtopping events

Thank you for this constructive comment. We have now added brief a synopsis of the key methodological features in the main text as below (Lines 78- 90):

“Depending on the level of detail considered, the total length of global coastlines ranges from 0.5 million kilometers if only open coasts are considered to 1.5 million kilometers, when including bays, estuaries and rugged coasts (Hinkel et al., 2013; Burke et al., 2001). The topology of open coasts is highly variable, comprising open sandy coasts, barrier islands, cliffs, river deltas, and engineered coasts (Schwartz, 2003). Two key coastal topographical parameters that are relevant for coastal overtopping are the foreshore slope, which influences wave runup and thus ECWL, and the maximum sub-aerial coastal elevation, which sets the threshold that is to be exceeded by ECWL for overtopping to occur. The global distribution of the maximum sub-aerial coastal protection elevations (within 1 km landward of the shoreline) derived from the AW3D30 data base (see Data and Methods and **Figure M1** for an illustration of the transect extraction) is shown in **Figure 2.a**. These maximum coastal elevations also take into account coastal dunes and coastal structures if resolved by the 30 m resolution of AW3D30 data base. The maximum sub-aerial coastal elevation appears to generally increase with latitude (**Figure 2.a**) and has a global average of 7 m. The coastal slope (**Figure 2.b**) relevant for wave runup calculations is computed from the shoreline to the maximum sub-aerial coastal elevation as derived in **Figure 2.a** (see Data and Methods and **Figure M1**). “

Line 113-how are all these different water level components combined? The timing of the different water level components is certainly a complex factor to consider

We apologize for not describing this in the previous version. This has now been clarified as below (Lines 333-334)

“All the above described components of ECWL feeding into Eq. 1 were calculated over the 1993-2015 period and ultimately interpolated to an hourly resolution to account for compound nature of ECWL.”

Figure 3—there should be more detail on this figure caption. I believe this is based on the 1993 to 2015 ERA interim reanalysis? But this should be clearly stated so figure can stand alone

In response to this constructive comment, we have now added more detail to the caption of Figure 3 as follows:

Figure 3. Global distribution of coastal overtopping over the period 1993 -2015: a) time-averaged annual number of overtopping hours (hrs/yr) and b) The 23 year trend in the annual number of overtopping hours ($N_{a,i}$) (computed using the complete hourly time series of ECWL, discretized into years) (see Methods for details on the computation approach adopted). Overtopping is assumed to occur when ECWL exceeds the maximum coastal elevation (derived from AW2D30). ECWL = $SLA + DAC + T + R$ is computed by combining hourly data of all contributing components over the 1993-2015 period. For clarity, locations for which zero overtopping was

computed are not shown in this figure (in contrast with Figure 2 where all computational transects are shown).”

Line 135-please describe in more detail here the potential drivers of these observed changes, such as regional sea level rise variability as has been well documented in the western tropical Pacific, changes in the wave climate, changes in storminess, etc..

We thank the reviewer for his comment. In response we have now added concise clarifications (Lines 120 – 131) on regional features of components of ECWL that work in unison (to result in higher ECWL in some regions) or in opposition (to result in lower ECWL in other regions) with each other. In particular, Melet et al. (2018) show at over long interannual timescales that the negative runup trend along the West coast of America mostly cancels out positive trends in SLA and surge (DAC), and this is also the case, albeit to a lesser extent, along the West coast of Europe. Although, the alternating coastal conditions around the Pacific, with shifts in wave activity and water level anomalies between the Northeastern and Northwestern Pacific, has recently been attributed to ENSO (Barnard et al., 2015, 2017; Mentaschi et al., 2017), here we prefer to not to comment on this possible mechanism due to relatively short historical analysis period of our study (23 years), which is not long enough to investigate such interannual mechanisms with decadal features (e.g. Pacific Decadal Oscillation).

Barnard, P., Hoover, D., Hubbard, D. et al. Extreme oceanographic forcing and coastal response due to the 2015–2016 El Niño. *Nat. Commun.*, 8, 14365. <https://doi.org/10.1038/ncomms14365>, (2017).

Barnard, P., Short, A., Harley, M. et al. Coastal vulnerability across the Pacific dominated by El Niño/Southern Oscillation. *Nature Geosci.*, 8, 801–807, <https://doi.org/10.1038/ngeo2539>, (2015).

Mentaschi, L., M. I. Vousdoukas, E. Voukouvalas, A. Dosio, and L. Feyen, Global changes of extreme coastal wave energy fluxes triggered by intensified teleconnection patterns. *Geophys. Res. Lett.*, 44, 2416–2426, doi:10.1002/2016GL072488, (2017).

The related text has now been modified as follows (Lines 120 – 131):

“The annual number of overtopping hours ($N_{a,g}$) exhibit a positive (i.e. increasing) trend (computed using the complete hourly time series of ECWL, discretized into years) in most parts of the world over the period 1993-2015 (**Figure 3.b**). The highest rates of increase are observed in the Gulf of Mexico, northern Europe (Baltic Sea), eastern Mediterranean region, east coast of Africa, southeast Asia, and northwestern Australia. This might be because these regions generally have small variability in extreme coastal water level (variance of the time series), and hence, even small increases in regional sea level can have a large impact on overtopping (Rueda et al., 2017). A few areas appear to have experienced a small trend over 1993 - 2015, mainly in the mid to high latitudes: the west coast of North America, northern Europe, and the southeast coast of South America.

Melet et al., (2018) showed that in different parts of the world, some component of sea level at the coast work in unison (to result in higher total sea level at the coast in some regions) or in opposition (to result in lower total sea level in other regions) with each other. In particular, the negative runup trend along the West coast of America mostly cancels out other increasing components and this is also the case, albeit to a lesser extent, along the West coast of Europe. “

Line 139-by integrated I think you mean a summation of all the profile overtopping hours? This should be made more clear, because it could also be interpreted as an average but the overtopping hours are clearly far too high for that.

In response to this comment we have now introduced Figure 4 as follows (Lines 132 – 137):

“In a globally aggregated sense, overtopping events are mostly due to a combination of wave runup and tides over the 1993-2015 period (Figure 4.a). When wave runup and tides are not accounted for (orange bars), the globally aggregated annual number of overtopping hours is much less than when all components contributing to ECWL are considered (grey bars). When all components of ECWL are accounted for, an increasing (significant at 95% level, using the Mann-Kendall test) trend is found for $N_{a,g}$ over the 1993 – 2015 period (Figure 4.a). This positive trend has resulted in approximately an increase by 50% in $N_{a,g}$ from 1993 to 2015. “

Furthermore, we have now made sure that there is no misunderstanding about the two main overtopping quantities we use as diagnostics by referring to them consistently and differentially as the annual number of overtopping hours (at each computational point) (Line 120) and the globally aggregated annual number of overtopping hours (Line 132) throughout the manuscript and the supplementary material. Moreover, we have introduced different symbols to refer to these two quantities as $N_{a,i}$ and $N_{a,g}$, and use these consistently and as appropriate throughout the manuscript and the supplementary material.

Line 145- again are you talking about the rate of global overtopping, i.e. the number of hours per year?

Please see response above. Yes this is the globally aggregated annual number of overtopping hours ($N_{a,g}$).

Line 168-171- at this point there has been no description of coastal topography resolution, nor any indication that high astronomical tides are a major driver of the projected increase in overtopping

Thank you for pointing out this shortcoming on our part, for which we apologize. We have now addressed this comment by adding the following text up front (Lines 63 – 66):

“Here we address this long-felt need by combining a new state-of-the-art global digital surface elevation model (ALOS World 3D from JAXA at 30m spatial resolution, named hereafter AW3D30 – Tadono et al., 2016; Zhang et al., 2019) with ECWL derived from a combination of satellite altimetry, tide and surge models and wave reanalyses, taking into account the key contribution of wave runup at open coasts.”

And in the results (Lines 132-133):

“In a globally aggregated sense, overtopping events (Na,g) are mostly due to a combination of wave runup and tides over the 1993-2015 period (Figure 4.a).”

Line 186-a bigger question is how well are estuaries resolved in your approach? Are waves allowed to penetrate into large estuaries where most of the mega cities reside, such as London, Bombay, Sydney, New York, etc. this could lead to significant overestimates of overtopping if embayments are not resolved. This is one of the big challenges for looking at global flooding risk due to sea level rise and waves.

Resolving estuaries is a challenge in any global scale assessment. A few recent studies in the literature investigated this particular and complex topic, sometimes considering wave setup (Kirezci et al., 2020), but this is out of the scope of our analysis. Our study focuses on open coasts where wave runup has a substantial contribution to ECWL and consequent coastal overtopping. This is now stated more explicitly in the revised manuscript at Lines (278 – 285):

“Maximum subaerial coastal elevation and coastal slopes were extracted from the above-mentioned MERIT, FLOPROS and AW3D30 datasets along the global coastline. Here, the Global Self-consistent, Hierarchical, High-resolution Geography Database (GSHHS - Wessel and Smith, 1996) coastline at the "h" highest resolution (~kilometric) was used. The coastal shoreline and topography are highly variable alongshore. In order to obtain reasonably robust estimates, cross-shore aerial topography profiles were extracted every 0.05 degrees alongshore (see **Figure M1**). From these, a regional profile every 0.5° alongshore was made robust by averaging ten 0.05° transects. This means that our analysis and conclusions are representative of main regional topographical features (e.g. typical low-lying beach-dune, high cliff coastline) but not of local features (e.g. estuaries).”

Line 212- you should also refer to Manoo Shirzaei's work in Science Advances 2018;

Shirzaei, M., Bürgmann, R. (2018), Global climate change and local land subsidence exacerbate inundation risk to the San Francisco Bay Area. Science Advances, 4, <https://doi.org/10.1126/sciadv.aap9234>

We thank the reviewer for point out this reference, which has now been cited . The associated text now reads as follows (Lines 234 - 247):

“The nearshore topography was considered constant in time here with no morphodynamic evolution, which means that possible sea level rise driven changes in the coastal slope and maximum coastal elevation are neglected herein. Even if detailed present-day bathymetry were available, past and future bathymetry would still remain unknown. As a result, this and other recent global studies, use a fixed coastal bathymetry over time periods spanning 50 – 100 years. However, coastal systems are among the most dynamic environments on Earth, continually evolving at various spatio-temporal scales, with, for e.g. a single large storm being able to reshape regional bathymetry which could significantly affect extreme coastal water levels in subsequent years. Thus, the consideration of passive coastal bathymetry over a 100-year period necessarily

assumes that computed coastal flooding is only a function of changes in coastal water levels (Le Cozannet et al., 2019; Serafin et al., 2017; 2019). It should also be noted that local vertical land movement (e.g. land subsidence) can in places (e.g. Jakarta, New Orleans, Ho Chi Minh City) result in relative sea level rise rates that are far greater than the global mean sea level rise (Nicholls et al., 2014; Shirzaei and Bürgmann, 2018) considered in all future projections herein. Consideration of these regional and local contributions to relative sea level rise will affect coastal flooding projections for certain locations, in particular at coastal cities and low-lying deltas (Hallegatte et al., 2013; Erkens et al., 2015; Brown and Nicholls, 2015; Kulp and Strauss, 2019; Becker et al., 2020).”

Line 212- other caveats to clearly layout and discuss include the resolution of the elevation data, which is not sufficient to resolve many overtopping points, the issue of resolving waves in estuaries and the nearshore, and the coincident timing of high tides and wave events, among others.

As also indicated above, in response this comment (and similar comments by other Reviewers), we have now added a devoted section on “Limitations” in the manuscript (and reproduced above) where we address most of the issues raised by the Reviewers on caveats. Where possible, we have also clarified many of these points throughout the manuscript:

- On the resolution of the elevation data (Lines 83 – 87): “The global distribution of the maximum sub-aerial coastal protection elevations (within 1 km landward of the shoreline) derived from the AW3D30 data base (see Data and Methods and **Figure M1** for an illustration of the transect extraction) is shown in **Figure 2.a**. These maximum coastal elevations also take into account coastal dunes and coastal structures if resolved by the 30 m resolution of AW3D30 data base.”
Furthermore, more details on the method to extract coastal slope and maximum coastal elevations from the DEM are given in the new Figure M1 and a comprehensive sensitive test on the influence of the topography dataset on our overtopping estimates is given in Supplementary material S2.
- On the issue of resolving waves in estuaries and the nearshore - As stated previously, this is beyond the scope of this study. We present our analysis and conclusions as being representative of the regional scale. To this end, each regional 0.5° alongshore transect is made more robust by averaging ten 0.05° transects. Therefore, what we use in our analysis are the main regional topographical features (e.g. typical low-lying beach-dune, high cliff coastline) and not local features (e.g. estuaries), which are smoothed out through the above mentioned averaging procedure. This is now highlighted in the text (Lines 279 – 285):

“Here, the Global Self-consistent, Hierarchical, High-resolution Geography Database (GSHHS - Wessel and Smith, 1996) coastline at the “h” highest resolution (~kilometric) was used. The coastal shoreline and topography are highly variable alongshore. In order to obtain reasonably robust estimates, cross-shore aerial topography profiles were extracted every 0.05 degrees alongshore (see Figure M1). From these, a regional profile every 0.5° alongshore was made robust by averaging ten 0.05° transects. This means that our analysis

and conclusions are representative of main regional topographical features (e.g. typical low-lying beach-dune, high cliff coastline) but not of local features (e.g. estuaries).”

- On the issue of co-occurrence of tides and waves - As stated previously, our historical ocean data are resampled at an hourly resolution which allows the deterministic resolution of the combination of tides with other met-ocean forcing components (such as wave runup, storm surge). For the future period, we make the reasonable assumption that the statistical distributions will not change significantly. Even if this assumption were not to hold for the future, given Vousdoukas et al., (2018) finding that the effect of changes in meteorological extremes on extreme sea levels are minor compared to that of sea level rise, the results will not be significantly different.

Line 226-how is this foreshore slope calculated?

The foreshore slope information required for the comparison of topographical datasets was obtained from Athanasiou et al. (2019). Here, the foreshore slope is computed from the depth of closure to the shore using GEBCO data (assumed to be an approximation of the shoreface slope). While this dataset is a good global one for the time being, it is known to underestimate the slope in coastal shallow waters.

This has now been clarified in the Supplementary material (Lines 56– 57):

“In Athanasiou et al. (2019), the nearshore slopes are computed from the depth of closure to mean sea level position.”

Line 244 – but the slope used in run up equations is typically along the upper shore face around or between MSL and MHW? So if you’re using the maximum coastal elevation in that region, which is not necessarily relevant to the run up surface, you could be drastically overestimating the overtopping.

We agree that the runup calculation depends on which section of the topography profile is used in the formulae (e.g. Stockdon et al. 2006, Blenkinsopp et al. 2015). However, there is currently no global dataset available on intertidal profile. A sensitivity study has been recently conducted by Melet et al. (2020a) on the influence of using locally reported beach slopes in runup estimates together with the temporal variations of this slope with wave conditions. Results of that study shows that improvements can be gained and we believe this refinement is a way forward. However, such a product is not available at global scale and global studies therefore have to necessary rely on satellite estimates. In a near future, an advanced approach such as that presented by Vos et al. (2020) could be applied globally to obtain more accurate coastal slope estimates from satellite imagery.

To address this point and report on the uncertainties in potential overtopping related to the topography, we have conducted a sensitivity analysis of the globally aggregated annual number of overtopping hours ($N_{a,g}$) on

the coastal topography dataset. This is reported in the Supplementary Material S2. We used different estimates of the coastal slope (from the MERIT and AW3D30 datasets, see Figure S3) and of the maximum coastal elevation (see Fig S2), including the FLOPROS dataset for coastal protections. This comparison shows substantial differences in the estimates of both coastal slopes and maximum subaerial elevations (Figs S2, S3). Subaerial coastal slopes can be different from intertidal slopes, but the latter is not always steeper: in the tropics for example, the intertidal section can be steep with flat low-lying coastal areas. Yet, despite these differences, our sensitivity analysis shows a robust increase of global potential overtopping with sea level rise.

Line to 273-just because Stockdon has been applied to beaches with greater slopes, does not mean that it is valid. Steeper slopes are typically referred to other engineering equations such as TAW: van der Meer, J. (2002). Technical report wave run-up and wave overtopping at dikes: Delft, Netherlands, Technical Advisory Committee on Flood Defence, 42 p. that are designed for overtopping of structures, not for wave run up on sandy beaches Small-scales features such as local dikes are not well resolved by our method which focuses more on regional scale topographical features (as explained above). However, at global scale, the principle behind Stockdon offers reasonable regional estimates, particularly at sandy low-lying coasts, which are the most exposed environments (as compared to cliffs, rocky or engineered and protected environments). As mentioned previously, we have now added a comprehensive section on “Limitations” where the global applicability of Stockdon’s parametrization for wave runup in different environments is discussed as follows (Lines 217 – 226).

“The global scale of the analysis presented here necessitates some simplifications in estimating ECWL, particularly in calculating wave runup using Stockdon et al.’s (2006) empirical formulae Stockdon et al.’s (2006) wave runup parametrization was developed for and is applicable for open wave-exposed sandy beaches. For these sandy coasts and beaches, as a rule of thumb, the wave setup is 20% of offshore wave height (Guza and Thornton, 1982). It is however commonly used for different environments such as gravel beaches (Poate et al., 2016). At rocky coasts with rocky platforms, wave runup is important but reduced by bottom friction over the rocky bottom (Dodet et al., 2018). At muddy coasts like the Amazon River to Guyana in South America, high suspended sediment concentrations tend to dampen wave action (and hence overtopping) (Winterwerp et al., 2007). The indiscriminate application of Stockdon et al.’s (2006) formulations at these latter two types of coasts may have therefore resulted in over-estimations in wave runup and hence overtopping in our analysis.”

Related text was added also in Data and method section (Lines 322-332)

“R can be predicted using different methodologies, such as direct numerical modeling with process-based local coastal models, meta-models, and empirical formulations (e.g. Dodet et al., 2019). In the Melet et al. (2020) discussion on the limitations of using Stockdon et al.’s (2006) parametrization at global scale, it is pinpointed that process-based coastal models also need local nearshore profiles as inputs, and cannot yet simulate R with nearshore morphological updating over long timescales and along the global coastline. R is therefore commonly predicted via empirical formulations that relate them to a set of simple environmental parameters

(see review by Dodet et al. (2019)). As this study aims at providing a first-order estimate, R is computed using empirical formulae. For instance, Diaz-Sanchez et al. (2014) mention that the scatter between empirically-predicted and observed R can be due to local processes that are not represented by the formulations' predictors based on deep water offshore waves. The automated computation procedures used in this study ensures that Eq. 2 is used on coasts with milder slopes, while Eq. 3 is used at steep profiles, such as for instance, when coastal defense structures are present.”

Supplementary material

While the analysis of the different global data sets is interesting, a more accurate understanding of the uncertainty of the overtopping projections could be obtained by a higher resolution locally derived survey such as airborne LiDAR or GPS surveys in a given location

This comment is comparable to Comment 3 of Reviewer 1. A validation of global satellite-based dataset was conducted in SW France (beach dune environment), but also recently in the Iberian Peninsula (Zhang et al., 2019). This includes extensive LIDAR surveys. In our paper, we present profiles derived from LIDAR survey and AW3D30 (in the section on *Coastal topography extraction in Data and Methods*, also reproduced above) and refer to Diaz et al., (2019) for more details on the inter-comparison.

This is also further addressed in Supplementary section 2 (Lines 40-62):

“The way in which the choice of the topography dataset may influence the overtopping results was investigated is described here. The choice of the topography dataset may affect overtopping calculations in two ways: 1) the limitation in the intertidal area (foreshore) which has an influence on the computation of coastal slope, and 2) the resolution of coastal features (coastal highs) in addition to issues in the absolute elevation (floating DEM relative to sea level). The capabilities of AW3D30 to represent coastal topography has been investigated in detail locally by Diaz et al. (2019) at Capbreton, SW France and compared to other satellite-derived topography data sets. Diaz et al.,s (2019) results show that AW3D30 has good skills to reproduce the topography of the coastal zone, with an overall good estimate of absolute elevation (a major improvement compared to previous products such as MERIT) because of the correction with ICESat-1. AW3D30 was found to be particularly capable of estimating coastal elevation maxima such as dune tops. The foreshore area is generally lacking from the dataset, which is an artefact of the optical methodology of stereoscopy employed by AW3D30 on the ALOS mission and as low texture is found in the foreshore zone due to hydrodynamics and rather uniform optical characteristics which precludes the method from finding homolog points in pairs of images. The global validation of AW3D30 is beyond the scope of this manuscript, but recent studies have performed local/regional validations of AW3D30 using ground truth and LIDAR (Zhang et al., 2019). An assessment of the uncertainty associated to the product is however necessary to gain confidence in our overtopping calculations. To this end, here the results obtained using AW3D30 are compared with different independent datasets. The MERIT topo-bathymetry dataset (Athanasidou et al., 2019) is used to obtain a second estimate of coastal slopes that are used to compute wave runoff. In Athanasidou et al. (2019), the

nearshore slopes are computed from the depth of closure to mean sea level position. In addition to MERIT, to account for artificial coastal protection, the FLOod PROtection Standards FLOPROS (Scussolini et al., 2016) dataset was used to obtain a third estimate of maximum coastal elevations and overtopping (Vousdoukas et al., 2018). FLOPROS was generated by using flood exceedance levels from national protection policies, excluding overtopping. This is different from satellite radar-based and optical-based MERIT and AW3D30 products respectively, with the result that FLOPROS derived coastal protection levels are significantly lower than MERIT and AW3D30 (Figure S2)."

Figure S6-the validation is a nice piece, but it seems the validation is based on a projection simply exceeding a given elevation over a vast area so perhaps this is highly simplified? It is hard to discern from these plots the skill of the overtopping analysis that you have conducted. Perhaps just more explanation is required to explain these plots, which seem to be highly pertinent to being able to defend the methods in this manuscript

In response to this constructive comment, we have now provided a more detailed description of the validation in Supplementary material S3 (Lines 95-118).

"Our methodology with a 0.5° alongshore resolution (i.e. the resolution of the global satellite-based product adopted to derive coastal topography and the resolution of the global scale ocean forcing used in this analysis) precludes any conclusions at the local scale and hence our analysis should be considered as the first-pass, best-effort assessment.

Here, the methodology adopted to compute overtopping at the global scale is tested for four documented major coastal flooding events (Figure S5) along the Atlantic coast of Europe (Figure S5.a, Xynthia storm in France, Bertin et al., 2012), Gulf of Mexico (Figure S5.b, Hurricane Katrina hurricane in USA, Fritz et al., 2007), Mediterranean South-East coast (Figure S5.c, Nile delta in Egypt; Frihy et al., 2010; Refaat and Eldeberk, 2016; Ismail, et al., 2012), Gulf of Guinea, West Africa (Figure S5.d, Lagos in Nigeria; Nwilo et al., 1997; Olaniyan and Afiesimama, 2003) and Majuro in Pacific Marshall Islands (Figure S5.e, Hoeke et al., 2013).

The primary goal is to assess whether our method is able to reproduce the extreme still water level (ESWL) by comparing our calculations with sea level tide-gauge timeseries from the Global Extreme Sea Level Analysis (GESLA) dataset (Woodworth et al., 2017), which has already been validated globally by Melet et al., (2018) (Figure S4 therein). The Extreme coastal water level is then computed by adding wave runoff (from using local coastal slope and offshore waves in Stockdon'06 parametrization) to the ESWL.

Our second objective is to determine the capabilities of the methodology to capture these observed historical overtopping/flooding major events from the comparison of computed extreme coastal water levels with these sea levels. Overtopping thresholds are computed from regional topography maxima and are aimed at describing regional characteristics, not local features. As stated in the main manuscript, we do not intend here to describe small scale topography, dune breaching (for instance during Xynthia) and overflow due to local

depressions in coastal elevation maxima (such as harbors, inlet), local shoreface singularities (such as coral reefs, convergence and sheltering, refraction/diffraction) but to detect regional characteristics of coastal topography and the general exposure to potential overtopping potential. Process-based numerical simulations perform better in capturing small-scale local behavior (in particular wave processes such as infragravity energy transfer – see Bertin et al., 2018). Here our focus is on regional and temporal patterns which fall within the scope of this study.”

Reviewer #3 (Remarks to the Author):

Review of Almar et al “How waves are accelerating global coastal overtopping”

Overall this is an interesting and innovative paper which advances the state-of-the-art of the global analysis of wave processes and especially overtopping at the coast. So I commend the authors on pulling it all this information and analysis together. It also demonstrates the complexity of the problem, something which is quite familiar to experts such as coastal scientists and engineers, but is not well understood beyond this small community due to the multiple and sometimes interacting factors that are involved.

While the analysis has great merits, there are also numerous details that raise questions and require development. Having read the material several times more questions and inconsistencies are apparent and these need to be addressed so the paper is understandable. Below I make suggestions on how the manuscript can be improved firstly via some general comments and then more specific comments referring to specific locations. In some cases the specific comments build on the general comments.

I would recommend publication if the authors can respond adequately to the points raised below.

We thank the reviewer for carefully assessing our manuscript and for his/her overall appreciation of our study.

General comments

Title: “How waves are accelerating global coastal overtopping” – this tends towards a journalist title -- the paper certainly shows evidence of an increase but not an acceleration of overtopping over 23 years of hindcast conditions (Figure 4). Further, the increase in overtopping is driven by mean sea level rise influencing wave transformation to the coast -- so again this makes the title misleading to me – a more neutral, as well as better and more accurate title would be “A global analysis of trends in coastal overtopping”. I note acceleration of overtopping is apparent in the simulation of the future but this reflects an accelerating RCP8.5 emission

trajectory – sea-level rise under an RCP1.9 or RCP2.6 emissions pathway would more likely show a steady rising, but non-accelerating trend in overtopping (see Oppenheimer et al., 2019 – SROOC Special Report by IPCC).

We thank the Reviewer for suggesting this neutral title which indeed better suits our findings. This was also requested by another reviewer and we have now changed the title as suggested by the Reviewer to: *“A global analysis of trends in coastal overtopping”*

Thank you for the reference to the IPCC SROOC report, which is now cited and was added to the reference list.

We have also added a more nuanced statement on the projected trends we see in coastal overtopping as follows (Lines 29-32): *“Our results also indicate that the projected acceleration in coastal overtopping should be starting about now and will be clearly discernible by 21st mid-century under a high-emission/low-mitigation scenario (RCP 8.5), while also being noticeable for lower-emissions/greater-mitigation scenarios (RCP 4.5 and RCP 2.6).”*

There is a lack of practical experience of flooding coming through in the manuscript and there seems to be an opinion that overtopping will impact all coastal flooding – for example the mention of deltas. Yes it will impact the seaward edge of deltas and may create new pathways, but the interior flooding in deltas will mainly be driven by river flow, surge, tide and sea level. In narrow flood plains or on low-lying small islands, overtopping the main source of floods. The paper does a bad job of communicating this important context to the research in a nuanced manner and this needs improvement.

We thank the reviewer for this insightful comment. We have now clarified that our study focuses on wave-exposed open coasts rather than inlets or estuaries. Throughout the manuscript, we have made small adjustments to ensure that it is clear that we only assess overtopping potential and not flooding. This is discussed in the new “Limitations” section we have now added (Lines 202-254):

“Limitations and way forward

Being a global scale assessment, inevitably there will be several limitations when interpreting our results at local scale. One of the main limitations is due to different impacts waves will have on different types of coasts (e.g deltas and sheltered coasts vs open coasts). Recent studies have shown that waves might have a complex influence on flooding at tidal inlets and estuaries, and particularly at large deltas, in combination with local hydrology and other sea level contributions derived from met-ocean forcing (Tazkia et al., 2017; Gallien et al., 2018; Lashley et al., 2019; Le Roy et al., 2015). Local precipitation or river discharge can lead to compound flood events when they occur concurrently to storm surge events and/or large wave runup events (Brammer, 2014; Ward et al., 2018; Moftakhari et al. 2017; Bevacqua et al. 2019; Paprotny et al., 2020). These additional factors could not be taken into account in our analysis due to the lack of suitable datasets at the global scale.

Global scale coastal flooding studies currently face a double observational bottleneck. On one hand, it is currently impossible to observe sea levels right at the coast, in particular wave contributions to extreme coastal water levels. On the other hand, accurate measurements of global coastal morphological evolution and subsidence trends is still to be done despite promising local to regional emerging satellite techniques (Benveniste et al., 2019; Melet et al., 2020b). In a near future, an advanced approach such as presented by Vos et al. (2020) to reconstruct foreshore slope from satellite-derived shoreline tracking and tide level, could be applied globally. The global scale of the analysis presented here necessitates some simplifications in estimating ECWL, particularly in calculating wave runup using Stockdon et al.'s (2006) empirical formulae. Stockdon et al.'s (2006) wave runup parametrization was developed for and is applicable for open wave-exposed sandy beaches. For these sandy coasts and beaches, as a rule of thumb, the wave setup is 20% of offshore wave height (Guza and Thornton, 1982). It is however commonly used for different environments such as gravel beaches (Poate et al., 2016). At rocky coasts with rocky platforms, wave runup is important but reduced by bottom friction over the rocky bottom (Dodet et al., 2018). At muddy coasts like the Amazon River to Guyana in South America, high suspended sediment concentrations tend to dampen wave action (and hence overtopping) (Winterwerp et al., 2007). The indiscriminate application of Stockdon et al.'s (2006) formulations at these latter two types of coasts may have therefore resulted in over-estimations in wave runup and hence overtopping in our analysis.

With respect to storm surge estimates used in this study, the relatively coarse resolution of the barotropic model (MOG-2D) used and the known inability of ERA-interim to capture extreme wind events, mean that our analysis would not account for hurricanes in our estimates of extreme storm surge events.

Regarding future globally aggregated projections of future coastal overtopping computed in this study, it must be noted that non-linear interactions between sea level rise and other contributing components (tides, waves, storm surge) have not been not accounted for. Furthermore, climate change driven variations in storm surge and waves have were not accounted for in this study.

The nearshore topography was considered constant in time here with no morphodynamic evolution, which means that possible sea level rise driven changes in the coastal slope and maximum coastal elevation are neglected herein. Even if detailed present-day bathymetry were available, past and future bathymetry would still remain unknown. As a result, this and other recent global studies, use a fixed coastal bathymetry over time periods spanning 50 – 100 years. However, coastal systems are among the most dynamic environments on Earth, continually evolving at various spatio-temporal scales, with, for e.g. a single large storm being able to reshape regional bathymetry which could significantly affect extreme coastal water levels in subsequent years. Thus, the consideration of passive coastal bathymetry over a 100-year period necessarily assumes that computed coastal flooding is only a function of changes in coastal water levels (Le Cozannet et al., 2019; Serafin et al., 2017; 2019). It should also be noted that local vertical land movement (e.g. land subsidence) can in places (e.g. Jakarta, New Orleans, Ho Chi Minh City) result in relative sea level rise rates that are far greater than the global mean sea level rise (Nicholls et al., 2014; Shirzaei and Bürgmann, 2018) considered in all future projections herein. Consideration of these regional and local contributions to relative sea level rise will affect

coastal flooding projections for certain locations, in particular at coastal cities and low-lying deltas (Hallegatte et al., 2013; Erkens et al., 2015; Brown and Nicholls, 2015; Kulp and Strauss, 2019; Becker et al., 2020).

Finally, coastlines have been modified in various ways by human activities, particularly in urbanized areas in which, for example ports have been constructed, land has been reclaimed from the ocean (Luijendijk et al., 2018), seawalls built to combat coastline recession, cliffs stabilized, and groins placed in an attempt to retain a beach fringe and maintain dunes (Serafin et al., 2019). For example, in the US alone, 14% of national coastline is estimated to be hardened with engineering structures, and this percentage is expected to increase to 33% by 2100 (see Tavares et al., 2020). Such human interventions to the natural system generally results in steepening of coastal slopes (e.g. seawalls, dikes), resulting in smaller wave dissipation zones compared to natural coasts, which is not accounted for in this study. “

The literature is very focused on post-2010 papers with only two citations from the 20th Century. This partly reflects the recent explosion in literature in this area, but the basic ideas being implemented here have been around and applied more locally for several decades – there is no link to that literature which would give qualitative expectations of the results found. For example, there has been a large amount of analysis on joint probability methods for wave overtopping and water levels to understand defence performance in more developed areas like Europe. Building on this, in places where sea-level rise is part of planning such as in much of north-west Europe, design of new defences often considers these effects i.e. adaptation to increased overtopping is already happening. The novelty of the analysis is the global aspect of the analysis. However, this sub-global experience – which in many ways this global analysis builds upon – should be acknowledged. Indeed, this was a choice we made to almost exclusively refer to more recent publications, partly to satisfy journal limit of 70 reference). And it is true that literature on coastal dynamics, and sea level rise and flooding in particular, has largely grown over the last past years. In response to this comment, we have adjusted now made reference to older seminal local to regional studies (e.g. Townend, 1994) and also to European efforts (EurOtop - Allsop et al., 2018) (Lines 35-43):

“Over the 21st century, sea level rise is projected to double the frequency of coastal flooding at most locations around the world (Vitousek et al., 2017; Lambert et al. 2020, Oppenheimer et al., 2019) potentially affecting an estimated global population of nearly 1 billion people without appropriate flood mitigation strategies (Nicholls and Small, 2002; Neumann et al., 2015; Tebaldi et al., 2012; Townend et al., 1994; Kulp et al., 2019). Regions with limited water-level variability at the coast (i.e., short tailed flood-level distributions), mainly located in the Tropics, are likely to be the most affected (Vitousek et al., 2017). An increase in flood occurrence in low-lying, vulnerable coastal zones could force significant population migration (Nicholls and Cazenave, 2010; Oppenheimer et al., 2019; Hauer et al., 2020). One process that could lead to coastal flooding is overtopping, which occurs when the extreme coastal water level (as defined by Gregory et al., 2019) exceeds the maximum coastal elevation (e.g. dunes, dykes, cliffs; see EurOtop - Allsop et al., 2018).”

The estimate of maximum coastal elevation is not especially clear and yet the outcome of this analysis is

absolutely critical to the results that emerge -- how far inland is maximum elevation evaluated? In the examples a few hundred metres, but what are the spatial limits on the algorithm that is applied? In Figure 2 the maximum elevation seem high in many locations such as in NW Europe where there is a large longshore variability in coastal elevation which is simply not expressed? Based on the numbers that I see here I would expect overtopping to be excluded in all cases. And yet in Figure M1, there is overtopping in these areas which surprises me. I would welcome the author's comments on this observation and query.

This is a comment shared by the two other reviewers. We have now extended the description of the topographic analyses (slope and maxima) and profiles computation in the new version of the manuscript. We chose our study to be representative of the regional scale: each regional 0.5° alongshore transect is made robust by averaging ten 0.05° transects. Cross-shore transects cover 100 points with a fixed 0.01° resolution (~1000 m), a distance which covers the local maximum elevation in most coastal environments. In this way, the main regional topography (e.g. typical low lying beach-dune, high cliff coastline) is accounted for but the influence of local features influence (small inlets) is not accounted for. This is now explained as follows (Lines 279- 285):

“Here, the Global Self-consistent, Hierarchical, High-resolution Geography Database (GSHHS - Wessel and Smith, 1996) coastline at the "h" highest resolution (~kilometric) was used. The coastal shoreline and topography are highly variable alongshore. In order to obtain reasonably robust estimates, cross-shore aerial topography profiles were extracted every 0.05 degrees alongshore (see **Figure M1**). From these, a regional profile every 0.5° alongshore was made robust by averaging ten 0.05° transects. This means that our analysis and conclusions are representative of main regional topographical features (e.g. typical low-lying beach-dune, high cliff coastline) but not of local features (e.g. estuaries).”

In Figure 2, for readability, values were regionally smoothed for the plot so patterns are clearly distinguishable while raw regional 0.5° data were used for the overtopping analyses. This explains the difference noticed by the Reviewer.

More details were added on the coastal topography extraction in the text and in Figure M1 as follows:

In the main text (Lines 83-87):

“The global distribution of the maximum sub-aerial coastal protection elevations (within 1 km landward of the shoreline) derived from the AW3D30 data base (see Data and Methods and **Figure M1** for an illustration of the transect extraction) is shown in **Figure 2.a**. These maximum coastal elevations also take into account coastal dunes and coastal structures if resolved by the 30 m resolution of AW3D30 data base.”

New Figure M1:

Figure M1. Regional AW3D30 topography in southwestern France compared to airborne LIDAR measurements that serves as a reference topography (from Diaz et al., 2019). Black and red segments represent cross shore profile examples at fine and coarse resolution, respectively. The insert shows LIDAR data and different cross-shore profiles (s1, s2 and s3 transects). Purple and blue lines in the insets represent the LIDAR and AW3D30 cross-shore transects, respectively. Markers are points used to compute the coastal slope (average of slopes between successive points from the shoreline to the maximum subaerial coastal elevation, indicated by a solid triangle).

On reading the paper several times, an additional figure defining a schematic profile and the different types of maximum coastal elevation would make the paper clearer – I suggest to develop such an additional figure.

We thank the reviewer for this suggestion, which was also suggested by another Reviewer.. We now provide the suggested information in Figure M1 focusing on how topography transects, slopes and elevation maxima are computed.

Figures 3 and 5 -- why is the overtopping data shown along the coast discontinuous compared to earlier Figure 2 where component data is continuous along the coast? As far as I can see, this change is not mentioned in the text and undermines confidence in the results.

We apologize for not explained this in the previous version of the manuscript. This has now been clarified in the caption of Figure 3 (see below). Figure 2 presents the coastal topographical slope and elevation at all 14140 computational points with an 0.5° resolution using AW3D30 data. Among these points, only 800 experienced overtopping in our study over the 1993-2015 period and these are the only points displayed in Figure 3 5.

Figure 3 was also re-edited to show greater details in the locations as shown below:

Figure 3. Global distribution of coastal overtopping over the period 1993 -2015: a) time-averaged annual number of overtopping hours (hrs/yr) and b) The 23 year trend in the annual number of overtopping hours ($N_{a,i}$) (computed using the complete hourly time series of ECWL, discretized into years) (see Methods for details on the computation approach adopted). Overtopping is assumed to occur when ECWL exceeds the maximum coastal elevation (derived from AW2D30). $ECWL = SLA + DAC + T + R$ is computed by combining hourly data of

all contributing components over the 1993-2015 period. For clarity, locations for which zero overtopping was computed are not shown in this figure (in contrast with Figure 2 where all computational transects are shown).

The Discussion needs to focus on insights on overtopping – more general comments on coastal flooding are not appropriate with this dataset.

We thank the Reviewer for this suggestion, which is shared also with Reviewer 2. In response we deleted whole paragraph that discussed general flooding considerations and human interventions The previous Discussion and looking forward section was completely revamped into new “Limitations and way forward”. We hope the reviewer finds the new section more straightforward and appropriate for the manuscript.

There is a lot of consideration of beaches. What about highly muddy coasts like the Amazon River to Guyana in South America – on these muddy coasts with high suspended sediment concentrations waves suspend material, but at high concentrations this tends to dampen wave action and hence overtopping as far as I understand. I would welcome the author’s comments on this observation and query.

We appreciate this point made by the Reviewer. Indeed, the interpretation of our overtopping estimates can get tricky at muddy coasts coastlines. Physically, high concentrations of suspended sediment will dampen wave energy and result in a reduction of overtopping. In any case, computation of overtopping in such areas accurately also faces the challenge of rapidly and continuously varying bathymetry and subaerial topography. In addition, Stereoscopy is also known to provide less accurate results for very low-lying mangrove environments. However, at first order and looking at trends, we believe our conclusions may still be valid as a first-pass estimate even in such areas.

We have however now added a cautionary statement for muddy coasts in the “Limitations” section (Lines 223-226) and an authoritative reference was added in the list:

“At muddy coasts like the Amazon River to Guyana in South America, high suspended sediment concentrations tend to dampen wave action (and hence overtopping) (Winterwerp et al., 2007). The indiscriminate application of Stockdon et al.’s (2006) formulations at these latter two types of coasts may have therefore resulted in over-estimations in wave runup and hence overtopping in our analysis.”

Reference added:

Winterwerp, J.C., de Graaff, R.F. Groeneweg, J., Luyendijk, A.P., Modelling of wave damping at Guyana mud coast, *Coastal Engineering*, 54 (3) (2007), 249-261

Specific comments

Line 20-22. “This study, for the first time, presents global scale coastal overtopping estimates, which account for not only the effects of sea level rise, storm surge and wave setup as traditionally done, but also that of wave runup and existing coastal protection measures.” I think the wave runup is improved much more than coastal protection. No discrimination of the importance and significance of the improvement.

We thank the Reviewer for pointing this out. The sentence has now been revised as follows (Lines 19-21) :

“This study, for the first time, presents global-scale coastal overtopping potential estimates, which account for not only the effects of sea level rise and storm surge as traditionally done, but also that of wave runup at exposed open coasts.”

Line 27 “under “business-as-usual” scenario RCP 8.5” – is this a business-as-usual scenario? (see latter comment and reference) – maybe more honestly described as a “high climate change” scenario?

We thank the reviewer for making this important distinction. We now refer to RCP 8.5 as a “high emission, low mitigation” scenario throughout the manuscript, also in line with the recent commentary by Hausfather and Peters (2020).

Line 27-28 “the globally integrated number of annual overtopping hours will increase at a rate faster than that of the global mean sea level rise itself.” Explain the physics why – linked to greater water depths and more wave energy reaching the coast – an indirect effect of sea-level rise.

In response to this comment we have now modified the relevant sentence as follows (Lines 177 – 181), however we have deleted this sentence from the abstract due to space limitations:

*“This is because the threshold elevation for overtopping is exceeded more often with accelerated sea level rise due to greater water depths and more wave energy reaching the coast, leading to a faster increase in the rate of change in $N_{a,g}$ compared to the rate of change of sea level itself. Not surprisingly, therefore, **Figure S1** shows that the number of regions around the world that exposed to overtopping increases non-linearly with increasing global mean sea level.”*

Line 30-31 “and will reach values more than 50 times larger by the end of the 21st century.” – this sounds like the RCP8.5 scenario is inevitable – this conclusion is conditional on a specific emissions pathway and this should be stated here in the Abstract

In response to this comment, we have now refined this phrase as follows (Lines 28 – 29):

“...reaching values that are as much as 50 times larger by the end of the 21st century under RCP 8.5, relative to present-day estimates.”

Line 34-35 “Coastal flooding is threatening human societies (Hinkel et al., 2014; Vousdoukas et al., 2018) and infrastructures (Koks et al., 2019) and sea level rise is expected to exacerbate the situation in the decades to come.” – a bit awkward as an opening – rephrase.

We have now modified this sentence as follows (Lines 35-38):

“Over the 21st century, sea level rise is projected to double the frequency of coastal flooding at most locations around the world (Vitousek et al., 2017; Lambert et al. 2020, Oppenheimer et al., 2019) potentially affecting an estimated global population of nearly 1 billion people without appropriate flood mitigation strategies (Nicholls and Small, 2002; Neumann et al., 2015; Tebaldi et al., 2012; Townend et al., 1994; Kulp et al., 2019).”

Line 39-41. “In particular, the low-lying coasts of Africa and Asia are thought to be the most vulnerable areas worldwide, at which an increase in flood occurrence could force population migration (Nicholls and Cazenave, 2010).” – forced migration could occur anywhere so better to say – “force significant population migration”.

We have now rephrased this sentence to be more generic as follows (Lines 40-42):

“An increase in flood occurrence in low-lying, vulnerable coastal zones could force significant population migration (Nicholls and Cazenave, 2010; Oppenheimer et al., 2019; Hauer et al., 2020).”

Line 52-53 “mainly due to the lack of global information on detailed coastal topography, knowledge of which is required to compute the wave contributions accurately.” Historically lack of computer power has been a big constraint. Earlier assessments often considered waves but very simply to avoid lots of computation.

We agree with the reviewer that computational power has been a major constraint in the past. However, coastal topography, that substantially affects wave contribution, remained unavailable (or/and inaccurate) until recently at global scale and, which we have exploited here. Because we show their importance, we hope that our findings constitute a call to refine wave coastal dynamics and contributions using numerical modeling at global scale.

Line 57 “highly simplified coastal topography/bathymetry” – is it better to say “highly simplified coastal topography/bathymetry assumptions”?

We completely agree with the Reviewer and have modified this phrase as follows (Lines 59-60):

“...contribution of waves to extreme sea levels are still based on highly simplified coastal topography/bathymetry assumptions (e.g. constant slope worldwide).”

Line 58-60 *“While many studies have acknowledged that local topography and foreshore slope can influence flood exposure and risk greatly (Vousdoukas et al., 2018; Luijendijk et al., 2018; Hauer et al., 2020; Minderhoud, 2019; Kulp et al., 2019), no concerted efforts have been taken to address this shortcoming to date.” I find this statement too vague – this paper attempts to address a specific aspect of this deficiency concerning waves and overtopping, rather than say improving flood plain topography and inundation. This text should be made more specific to the aims and objectives of this paper.*

In response to this comment, we have now added a sentence to differentiate our study from the numerous existing literature. New text reads as follows (Lines 60-66):

“While many studies have acknowledged that local topography can greatly influence wave runup, and consequently flood exposure and the associated risk (Vousdoukas et al., 2018; Luijendijk et al., 2018; Hauer et al., 2020; Minderhoud, 2019; Kulp et al., 2019), no concerted efforts have been taken yet to address this shortcoming at global scale. Here we address this long-felt need by combining a new state-of-the-art global digital surface elevation model (ALOS World 3D from JAXA at 30m spatial resolution, named hereafter AW3D30 – Tadono et al., 2016; Zhang et al., 2019) with ECWL derived from a combination of satellite altimetry, tide and surge models and wave reanalyzes, taking into account the key contribution of wave runup at open coasts.”

Line 64-65 *“global scale estimates of the acceleration of overtopping in recent decades and under one high-end sea level rise scenario.” Change ‘acceleration’ to ‘increase’ – and state the timescale of the scenario analysis.*

We thank the reviewer for making this important distinction. In response we have now changed the sentence as follows (Lines 66-69):

“We quantify, for the first time, global scale estimates of the increase of potential overtopping in recent decades and present first-pass, globally aggregated projections of future coastal overtopping in response to projected global mean sea level rise over the 21st century.”

Line 72 Change *“World” to “Global”.*

The title of the Section has now been changed (Line 77) :

“Global distribution of maximum coastal elevation and sub-aerial coastal slope”

Line 73 *“The length of the global coastline exceeds 1.6 million kilometers (Burke et al., 2001)” – is this an appropriate length to quote as the estimate includes estuary costs. My understanding is that the analysis in the paper is focussed on the open coast where wave action is important, so is the length of open coast used in the analysis the length that should be cited – I expect this is much shorter than the number from Burke et al (2001). Please review and revise.*

Line 74-76 -- *31% sandy beaches translates into more than 500,000 km of sandy beaches worldwide based on the global coastal length of more than 1.6 million km reported in the paper. Is this credible and consistent with*

the length assumed in Luijendijk et al. (2018) who focussed mainly on the open coast – this links to the point above and they might be addressed together.

We have considered the above two comments together. The global shoreline can be seen as a fractal object depending on the level of detail considered. To be consistent, we recomputed the length of the coasts used in our dataset (derived from GSHHS - Wessel and Smith, (1996)- 0.5° alongshore coastline resolution) that indeed focus on the open coasts where wave action is present. We now refer to this in the context of Luijendijk et al. (2018) and the result is slightly above 500,000 km.

New text reads as follows (Lines 78-81):

“Depending on the level of detail considered, the total length of global coastlines ranges from 0.5 million kilometers if only open coasts are considered to 1.5 million kilometers, when including bays, estuaries and rugged coasts (Hinkel et al., 2013; Burke et al., 2001). The topology of open coasts is highly variable, comprising open sandy coasts, barrier islands, cliffs, river deltas, and engineered coasts (Schwartz, 2003).”

Line 74-76 “Among these coastlines, sandy beaches (fine to coarse sand) represent ~31% of ice-free world coasts (Luijendijk et al., 2018, Vousdoukas et al., 2020). In general, sandy beach slopes range from 0.01 (for finer sediment) and 0.2 (for gravel beaches) (Poate et al., 2016).” This is confusing as gravel is by definition not sand – are you talking about all beaches regardless of grain size or just sand beaches? -- please revise and clarify.

We agree these sentences were rather confusing and were not adding anything to the paper and for clarity we deleted them. We hope the section is now more straightforward.

Revised paragraph now reads (Lines 78-94):

*“Depending on the level of detail considered, the total length of global coastlines ranges from 0.5 million kilometers if only open coasts are considered to 1.5 million kilometers, when including bays, estuaries and rugged coasts (Hinkel et al., 2013; Burke et al., 2001). The topology of open coasts is highly variable, comprising open sandy coasts, barrier islands, cliffs, river deltas, and engineered coasts (Schwartz, 2003). Two key coastal topographical parameters that are relevant for coastal overtopping are the foreshore slope, which influences wave runup and thus ECWL, and the maximum sub-aerial coastal elevation, which sets the threshold that is to be exceeded by ECWL for overtopping to occur. The global distribution of the maximum sub-aerial coastal protection elevations (within 1 km landward of the shoreline) derived from the AW3D30 data base (see Data and Methods and **Figure M1** for an illustration of the transect extraction) is shown in **Figure 2.a**. These maximum coastal elevations also take into account coastal dunes and coastal structures if resolved by the 30 m resolution of AW3D30 data base. The maximum sub-aerial coastal elevation appears to generally increase with latitude (**Figure 2.a**) and has a global average of 7 m. The coastal slope (**Figure 2.b**) relevant for wave runup calculations is computed from the shoreline to the maximum sub-aerial coastal elevation as derived in **Figure***

2.a (see Data and Methods and **Figure M1**). The global median value of the sub-aerial coastal slope this derived is 0.04. Regional patterns are clearly visible, such as the along-coast gradient in beach slope along the west coast of North America, from relatively low (0.04) in the tropics to rather steep (0.15) in high latitudes with rockier coastlines. Similar features are observed in the southern hemisphere. Africa, the continent with the largest length of sandy coasts (e.g. Luijendijk et al. 2018), generally has gentle coastal slopes. “

Line 79-81 “Coastal morphology has been modified in various ways by human activities, particularly in urbanized areas in which, for example ports have been constructed, seawalls built to combat coastline recession, cliffs stabilized, and groins placed in an attempt to retain a beach fringe and maintain dunes (Serafin et al., 2019).” There has also been significant coastal reclamation or land claim – a major process. Add as a process with a source.

The manuscript had to be shortened and for clarity we decided to delete this paragraph which we felt was not adding value and was not part of the core of the manuscript. We hope this section now reads better.

Line 84-85 “urbanization and certain land use practices (agriculture, deforestation of mangroves; see Luijendijk et al., 2018; Mentaschi et al., 2018),” – these are not really appropriate references for this point. Please revise with appropriate references.

As above. We decided to delete some unnecessary sentences to shorten the manuscript and and to improve readability.

Line 89-90 “The coastal elevations shown in Figure 2(a) are the maximum subaerial coastal elevations (including dunes and coastal structures).” – the definition of coastal elevation is unclear. Improve Figure M1 to define what this means?

This is a comment made by all three reviewers. In this regard, we improved Figure M1 and extended the related text in the manuscript. It now better details how the topographical information was extracted and how the slope and elevation are defined and computed.

The methodology is clarified in the main text:

In the “Global distribution of maximum coastal elevation and sub-aerial coastal slope” section (Lines 87-94):

“The maximum sub-aerial coastal elevation appears to generally increase with latitude (**Figure 2.a**) and has a global average of 7 m. The coastal slope (**Figure 2.b**) relevant for wave runup calculations is computed from the shoreline to the maximum sub-aerial coastal elevation as derived in **Figure 2.a** (see Data and Methods and **Figure M1**). The global median value of the sub-aerial coastal slope this derived is 0.04. Regional patterns are clearly visible, such as the along-coast gradient in beach slope along the west coast of North America, from relatively low (0.04) in the tropics to rather steep (0.15) in high latitudes with rockier coastlines. Similar

features are observed in the southern hemisphere. Africa, the continent with the largest length of sandy coasts (e.g. Luijendijk et al. 2018), generally has gentle coastal slopes.”

and in the “Coastal topography extraction” section, Lines 288-294:

“The maximum coastal elevation and coastal slope at each profile were calculated using an automated detection method. In this method, the first step is the identification of the local sea-land orientation of each profile, based on the average topography values on the two sides of the shoreline: the higher side is taken to be land and the lower side to be sea. Second, the highest coastal point of each transect (e.g. dune, cliff top, crest of structure) was taken as the local landward maximum that was closest to shoreline, within 1 km landward of the shoreline (see **Figure M1**). The slope used in the wave contribution calculations was then estimated as the average slope between the shoreline and maximum coastal elevation, following the method presented by Diaz et al. (2019) – see insert in **Figure M1**.”

New Figure M1 :

Figure M1. Regional AW3D30 topography in southwestern France compared to airborne LIDAR measurements that serves as a reference topography (from Diaz et al., 2019). Black and red segments represent cross shore profile examples at fine and coarse resolution, respectively. The insert shows LIDAR data and different cross-shore profiles (s1, s2 and s3 transects). Purple and blue lines in the insets represent the LIDAR and AW3D30 cross-shore transects, respectively. Markers are points used to compute the coastal slope (average of slopes between successive points from the shoreline to the maximum subaerial coastal elevation, indicated by a solid triangle).

Line 97 “Japan show high variability of slope/elevation within small distances” – I do not see this in either panel – is Figure 2 at the right scale to show this global information?

We agree, regional subpanels can show regional patterns in more detail. However, to better convey and strengthen our main message, we decided to delete this sentence from the manuscript following the Reviewer remark.

Line 103-104 “Overtopping occurs when the coastal water level exceeds that of the crest level of natural (e.g. dunes) or artificial (e.g. dykes) coastal defences.” This is an important definition as it is core to the paper and should be stated as the language and definitions vary across the literature. “Overtopping occurs when the coastal water level exceeds that of the crest level of natural (e.g. dunes) or artificial (e.g. dykes) coastal defences and landward flow can occur.”

This definition has now been rephrased and moved to the Data and Methods sections, due to the structural changes we had to make in the manuscript to accommodate Reviewer comments while satisfying journal manuscript length limits. New text reads as follows (Lines 337 – 339):

“Potential overtopping is defined to occur when the ECWL exceeds the maximum coastal elevation. To temporally-aggregate the event level information, the number of hours of potential overtopping occurrences is counted at each computational point for every year in the 1993 – 2015 period.”

Line 103-104 – how well are coastal defences represented in the dataset – this must be a large uncertainty in the data. Is it even useful to include coastal defences in the analysis with the available data?

Line 115 “small-scale coastal defences” – so what are we talking about here? Elsewhere a virtue of including defences is made – now defences cannot be resolved. What is small scale. More generally there is a lack of clarity of this aspect of the research.

Here we respond to both of the above comments together. This is an important point and we apologize for the lack of clarity on this issue in the previous version of the manuscript. AW3D30 has a 30 m resolution, which

makes it possible to resolve relatively wide coastal protections such as man-made dunes or concrete walls found at regional scale. However, city level infrastructure and harbor seawall cannot be resolved and distinguished and remain out of the scope of this study, which is firmly fixed on regional patterns.

We have now modified this sentence (Lines 83 – 87), together with the improved Figure M1 and associated text, we believe the spatial scales relevant for our assessment are now sufficiently clear:

“The global distribution of the maximum sub-aerial coastal protection elevations (within 1 km landward of the shoreline) derived from the AW3D30 data base (see Data and Methods and **Figure M1** for an illustration of the transect extraction) is shown in **Figure 2.a**. These maximum coastal elevations also take into account coastal dunes and coastal structures if resolved by the 30 m resolution of AW3D30 data base. “

Line 107 “In all, Eq. 1 was applied at 14,140 coastal profiles around the world.” So one profile covers roughly every 100 km based on the length stated in Burke et al (2001)? The sampling strategy for selecting these profiles needs to be explained and linked to data/methods.

We have now clarified this as follows (Lines 279-287):

“Here, the Global Self-consistent, Hierarchical, High-resolution Geography Database (GSHHS - Wessel and Smith, 1996) coastline at the "h" highest resolution (~kilometric) was used. The coastal shoreline and topography are highly variable alongshore. In order to obtain reasonably robust estimates, cross-shore aerial topography profiles were extracted every 0.05 degrees alongshore (see **Figure M1**). From these, a regional profile every 0.5° alongshore was made robust by averaging ten 0.05° transects. This means that our analysis and conclusions are representative of main regional topographical features (e.g. typical low-lying beach-dune, high cliff coastline) but not of local features (e.g. estuaries). Furthermore, islands with a circumference less than 0.5° were excluded from the analysis, as we deemed it sufficient at a global scale and representative of the regional values seen in the literature. This resulted in a total of 14,140 profiles for which the analysis was performed.”

Line 117-119 “Among these, the low-lying sedimentary plains such as deltas (e.g. Bengal, Nile and Mississippi Deltas for instance, see Nicholls et al., 2007 and Besset et al., 2019) emerge as the areas in the world that are most threatened by episodic coastal flooding.” Firstly how important is overtopping to flooding in these huge and wide delta plains? This research sheds little light on flooding in deltas and this statement demonstrates that the authors have not thought through the implications of their results sufficiently. Further I do not even see a high incidence of overtopping in the Bengal delta. Check and rephrase.

This comment is linked to previous comments regarding compound flooding. While we agree with the Reviewer that overtopping is not the only process at play in deltas (inlands), they clearly contribute to the hazard during episodic flooding, similarly to what is observed in other low lying lands, such as in Africa. This has been

documented for Katrina and for several other deltas, like the Niger and the Volta, or the Mekong, regarding coastal cities on the outer part of the delta barriers (sediment and alluvial barriers exposed to waves).

The reviewer is right in the sense the inland extent of overtopping can be limited in terms of overflow and volume compared with other components such as surge due to wind and atmospheric pressure. Nevertheless, in areas affected by compound flooding, an increase in overtopping could also increase the compound effect.

In response to this comment we have rephrased the relevant sentences as follows:

(Lines 115 – 119):

“Regional hot-spots of overtopping can be seen in southeast Asia, northern Europe, the southern Mediterranean coast, and eastern coast of the USA. Among these, the low-lying sedimentary plains such as deltas (e.g. Bengal, Nile and Mississippi for instance, see Nicholls et al., 2007 and Besset et al., 2019) emerge as areas that are most prone to overtopping. The limitations associated with the application of our approach at deltaic coasts are addressed in the Limitations and way forward section. “

(Lines 203 – 216):

“Being a global scale assessment, inevitably there will be several limitations when interpreting our results at local scale. One of the main limitations is due to different impacts waves will have on different types of coasts (e.g. deltas and sheltered coasts vs open coasts). Recent studies have shown that waves might have a complex influence on flooding at tidal inlets and estuaries, and particularly at large deltas, in combination with local hydrology and other sea level contributions derived from met-ocean forcing (Tazkia et al., 2017; Gallien et al., 2018; Lashley et al., 2019; Le Roy et al., 2015). Local precipitation or river discharge can lead to compound flood events when they occur concurrently to storm surge events and/or large wave runup events (Brammer, 2014; Ward et al., 2018; Moftakhari et al. 2017; Bevacqua et al. 2019; Paprotny et al., 2020). These additional factors could not be taken into account in our analysis due to the lack of suitable datasets at the global scale.”

(Lines 240 – 247):

“Thus, the consideration of passive coastal bathymetry over a 100-year period necessarily assumes that computed coastal flooding is only a function of changes in coastal water levels (Le Cozannet et al., 2019; Serafin et al., 2017; 2019). It should also be noted that local vertical land movement (e.g. land subsidence) can in places (e.g. Jakarta, New Orleans, Ho Chi Minh City) result in relative sea level rise rates that are far greater than the global mean sea level rise (Nicholls et al., 2014; Shirzaei and Bürgmann, 2018) considered in all future projections herein. Consideration of these regional and local contributions to relative sea level rise will affect coastal flooding projections for certain locations, in particular at coastal cities and low-lying deltas (Hallegatte et al., 2013; Erkens et al., 2015; Brown and Nicholls, 2015; Kulp and Strauss, 2019; Becker et al., 2020).”

Line 137-138 "Figure 3. Global map of coastal overtopping (number of hours per year): a) occurrence and b) 23 year trend of occurrences (see methods for details on computation approach adopted)." – is a 23 year trend meaningful – when conventionally 30 years of data is required to define a climate variable?

We appreciate the Reviewer's comment on the length of the data set, and we do agree that 7 more years of data would have resulted in a more robust trend. However, we would prefer to present this trend calculation here for several reasons. First, this study represents the best effort study to date in terms of coastal overtopping at global scale, and the availability of satellite altimetry observations (>1993) defined (like for many others analysis) our study period. Second, the computed globally aggregated overtopping trend is statistically significant at the 0.95 confidence level, although we acknowledge the trend could be lower at particular locations and even become not significant as this trend can be affected by regional long decadal interannual variability aliasing as induced by climate modes.

Line 145-146 "Global overtopping has increased almost by 1.5 from 1993." Not especially clear to a reader – try "Global overtopping has increased almost 1.5 times from 1993 to 2016."

We thank the Reviewer for pointing out this typo. This has now been corrected at follows:

Lines (25 – 26):

"In a globally aggregated sense, our results show that the annual overtopping hours have increased by almost 50% between 1993 and 2015. "

Lines (136 – 137):

"This positive trend has resulted in approximately an increase by 50% in $N_{a,g}$ from 1993 to 2015."

Lines (187 – 189):

"Rather, it is the combination of regional sea level, storm surge, wave runup and tide, with the increasing regional sea level being the main driver, that is responsible for the observed positive trend in overtopping, resulting in an increase of globally aggregated annual overtopping hours by almost 50% from 1993 to 2015."

Line 148 – Figure 4 -- overtopping without waves – this is sometimes called overflow as it reflects time averaged water levels.

While we agree with the reviewer, we think using a yet another different term might confuse the reader and therefore have not changed this. We have however completely re-modelled Figure 4 (as shown below) and we hope that it is now more straightforward and of a sufficiently high quality standard for NCOMMs:

Figure 4. a) The globally aggregated annual number of overtopping hours ($N_{a,g}$) (grey bars), contribution to overtopping from waves and tide only ($R + T$, blue bars), and the contribution to overtopping from regional sea level rise and storm surge only (e.g. $SLA + DAC$, orange bars). b) Future projections of $N_{a,g}$ when contribution to overtopping from waves and tide ($R + T$) are included (grey bars) or excluded (orange bars) considering projected median global mean sea level rise. The secondary x-axes at the bottom indicate the years at which the various median global mean sea level rise values from the main x-axis will be reached in RCP 8.5, RCP 4.5 and RCP 2.6 projections presented in IPCC, 2019 (Oppenheimer et al. 2019). Insert in b) compares different rates of changes of global mean sea level (in mm/yr) with the computed rate of change of $N_{a,g}$. Triangles represent computed $N_{a,g}$ values and the dashed line is an exponential regression ($R^2=0.8$) fitted to the triangles, indicating an exponential factor 2.7 between the rates of change global mean sea level and $N_{a,g}$.

I would make the insert in b) even bigger and the computed marks in bold.

This has been done in the re-modelled Figure 4 (see above)

Line 148 – Figure 4 – the scale on the x axis from 2000 to 2100 is wrong – correct.

This has now been corrected. Thank you.

Line 148 – Figure 4 – why does the (b) panel not include waves only as panel (a)?

This was a mistake in the previous Figure. What we intended to show was the differential contributions of R+T and SLA+DAC to ECWL. We apologize for this error, which has now been corrected in the new Figure 4.

Line 158 (“business-as-usual”) – correct based on earlier comments

We thank the reviewer for making this important distinction. We now refer to RCP 8.5 as a “high emission, low mitigation” scenario throughout the manuscript, also in line with the recent commentary by Hausfather and Peters (2020).

Line 159-160 “Figure 4.b shows that, in a globally aggregated sense, if wave runup were not to be considered in computations, the total annual overtopping hours by 2100 would be underestimated by over 40%.” This is a great global quantification – but presented like a completely new result. Qualitatively it is not a surprise to coastal scientists and engineers and it would be better to present it this way – we knew the term was missing and we knew it had a positive sign so the qualitative effect is self evident. You have quantified it, which is an important contribution – but this is not fundamentally new. See for example <https://doi.org/10.1016/j.oceaneng.2016.01.026> and earlier work which inspired it such as Townend (1994). The Townend paper in particular illustrates the relevant previous analysis which this paper fails to cite -- Townend IH (1994) Variation in design conditions in response to sea-level rise. Proc Inst Civil Eng 106(3):205–213.

We thank the reviewer for pointing out this reference. This important pioneering study has now been cited in introduction as follows (Lines 35 – 38):

“Over the 21st century, sea level rise is projected to double the frequency of coastal flooding at most locations around the world (Vitousek et al., 2017; Lambert et al. 2020, Oppenheimer et al., 2019) potentially affecting an estimated global population of nearly 1 billion people without appropriate flood mitigation strategies (Nicholls and Small, 2002; Neumann et al., 2015; Tebaldi et al., 2012; Townend et al., 1994; Kulp et al., 2019).”

Also following the Reviewer’s very clear and correct identification of one of the key novelties of our assessment, we have now refined our conclusion related to Figure 4(b) as follows (Lines 166 – 167):

“**Figure 4.b** shows that, in a globally aggregated sense, if wave and tide contributions were not considered in ECWL computations (orange bars), $N_{a,g}$ by 2100 would be underestimated by over 80%.”

Line 170-171 “We demonstrate that overtopping events are in fact mainly due to the combined effect of large wave runup events and high astronomical tides.” Is this new?

In response to this comment we have refined our text as follows (Lines 182 – 186):

“This study has, for the first time, quantitatively assessed the potential for coastal overtopping at global scale, both for the past few decades and over the 21st century, by combining high-resolution coastal topography from recently developed global satellite-based products with state-of-the-art computations of extreme coastal water level at the coast (including wave contributions). Our results confirm that overtopping events are mainly due to the combined effect of wave runup astronomical tides.”

Line 171-173 “However, these contributing processes by themselves do not induce a significant trend in the globally integrated number of annual overtopping hours, rather, it is the combination of regional sea level, wave runup and tide that results in an increase of this quantity.” – it is the combination but I would say sea level is the driver – so it is the combination and the fact that one component (SLR) has a strong trend.

In response to this comment, we have now modified the text as follows (Lines 185 – 189):

“Our results confirm that overtopping events are mainly due to the combined effect of wave runup astronomical tides. However, these processes alone do not explain the observed positive trend in the globally aggregated annual overtopping hours. Rather, it is the combination of regional sea level, storm surge, wave runup and tide, with the increasing regional sea level being the main driver, that is responsible for the observed positive trend in overtopping, resulting in an increase of globally aggregated annual overtopping hours by almost 50% from 1993 to 2015.”

Line 173-175 “Thus, our results reaffirm the previously reported (Prime et al., 2016; Serafin et al., 2017) finding that sea-level rise will have a greater impact on 21st century coastal flooding than future changes in wave climate).” – your results are only relevant to flooding due to overtopping and there are other flood mechanisms linked to sea-level rise.

We agree with the reviewer. However, this sentence is no longer present in the new Limitations and way forward section.

Lines 175-176 “The interaction of sea level and topography increases overtopping events at a rate faster than sea level rise itself with a found exponential factor of 2.7 with SLR.” On soft coasts topography would respond to sea-level rise – is this effect considered as it would offset increase in run-up?

We fully agree with this comment and have included the following text in the new Limitations and Way forward section (Lines 236 – 247):

“As a result, this and other recent global studies, use a fixed coastal bathymetry over time periods spanning 50 – 100 years. However, coastal systems are among the most dynamic environments on Earth, continually evolving at various spatio-temporal scales, with, for e.g. a single large storm being able to reshape regional bathymetry which could significantly affect extreme coastal water levels in subsequent years. Thus, the consideration of passive coastal bathymetry over a 100-year period necessarily assumes that computed coastal flooding is only a function of changes in coastal water levels (Le Cozannet et al., 2019; Serafin et al., 2017; 2019). It should also be noted that local vertical land movement (e.g. land subsidence) can in places (e.g. Jakarta, New Orleans, Ho Chi Minh City) result in relative sea level rise rates that are far greater than the global mean sea level rise (Nicholls et al., 2014; Shirzaei and Bürgmann, 2018) considered in all future projections herein. Consideration of these regional and local contributions to relative sea level rise will affect coastal flooding projections for certain locations, in particular at coastal cities and low-lying deltas (Hallegatte et al., 2013; Erkens et al., 2015; Brown and Nicholls, 2015; Kulp and Strauss, 2019; Becker et al., 2020).”

Lines 177-178 “Under the RCP 8.5 sea level rise trajectory, the projected acceleration in coastal overtopping should be starting about now and will be clearly discernible by about 2050.” This is a good and interesting result. However, as some authors have argued that the RCP8.5 emissions scenario is becoming less likely and lower emissions are much more likely (Hausfather and Peters, 2020, Emissions – the ‘business as usual’ story is misleading, Nature, 577, 618-620.) Some commentary on this aspect and the possible changes with lower emission scenarios is required.

We thank the Reviewer for this interesting comment. Please note that that our conclusions are based global mean sea level rise values and as such can be linked with any climate scenario (as can be seen in the multiple X-axis in the revised Figure 4 (b). In response to this comment we have modified this sentence to also encompass climate scenarios other than RCP 8.5 as follows (Lines 196 – 199):

“Under the high emission, low mitigation climate scenario RCP 8.5, the acceleration in coastal overtopping is projected to be starting about now (2020) and will be clearly discernible by 21st mid-century. This acceleration in overtopping is also projected, albeit at a lower pace, for the low-emission/high-mitigation and moderate-emission/moderate-mitigation scenarios RCP 2.6 and 4.5 respectively.”

Line 179-186 – this paragraph is not very clear – what are the relevant insights or limitations from the paper.

This comment on the lack of a good description of limitations is shared by the other reviewers, and in response we have now included an extensive discussion on Limitations and way forward as follows (Lines 202 – 254):

“Limitations and way forward

Being a global scale assessment, inevitably there will be several limitations when interpreting our results at local scale. One of the main limitations is due to different impacts waves will have on different types of coasts (e.g deltas and sheltered coasts vs open coasts). Recent studies have shown that waves might have a complex

influence on flooding at tidal inlets and estuaries, and particularly at large deltas, in combination with local hydrology and other sea level contributions derived from met-ocean forcing (Tazkia et al., 2017; Gallien et al., 2018; Lashley et al., 2019; Le Roy et al., 2015). Local precipitation or river discharge can lead to compound flood events when they occur concurrently to storm surge events and/or large wave runup events (Brammer, 2014; Ward et al., 2018; Moftakhari et al. 2017; Bevacqua et al. 2019; Paprotny et al., 2020). These additional factors could not be taken into account in our analysis due to the lack of suitable datasets at the global scale.

Global scale coastal flooding studies currently face a double observational bottleneck. On one hand, it is currently impossible to observe sea levels right at the coast, in particular wave contributions to extreme coastal water levels. On the other hand, accurate measurements of global coastal morphological evolution and subsidence trends is still to be done despite promising local to regional emerging satellite techniques (Benveniste et al., 2019; Melet et al., 2020b). In a near future, an advanced approach such as presented by Vos et al. (2020) to reconstruct foreshore slope from satellite-derived shoreline tracking and tide level, could be applied globally. The global scale of the analysis presented here necessitates some simplifications in estimating ECWL, particularly in calculating wave runup using Stockdon et al.'s (2006) empirical formulae Stockdon et al.'s (2006) wave runup parametrization was developed for and is applicable for open wave-exposed sandy beaches. For these sandy coasts and beaches, as a rule of thumb, the wave setup is 20% of offshore wave height (Guza and Thornton, 1982). It is however commonly used for different environments such as gravel beaches (Poate et al., 2016). At rocky coasts with rocky platforms, wave runup is important but reduced by bottom friction over the rocky bottom (Dodet et al., 2018). At muddy coasts like the Amazon River to Guyana in South America, high suspended sediment concentrations tend to dampen wave action (and hence overtopping) (Winterwerp et al., 2007). The indiscriminate application of Stockdon et al.'s (2006) formulations at these latter two types of coasts may have therefore resulted in over-estimations in wave runup and hence overtopping in our analysis.

With respect to storm surges estimates used in this study, the relatively coarse resolution of the barotropic model (MOG-2D) used and the known inability of ERA-interim to capture extreme wind events, mean that our analysis would not account for hurricanes in our estimates of extreme storm surge events.

Regarding future globally aggregated projections of future coastal overtopping computed in this study, it must be noted that non-linear interactions between sea level rise and other contributing components (tides, waves, storm surge) have not been not accounted for. Furthermore, climate change driven variations in storm surge and waves have were not accounted for in this study.

The nearshore topography was considered constant in time here with no morphodynamic evolution, which means that possible sea level rise driven changes in the coastal slope and maximum coastal elevation are neglected herein. Even if detailed present-day bathymetry were available, past and future bathymetry would still remain unknown. As a result, this and other recent global studies, use a fixed coastal bathymetry over time periods spanning 50 – 100 years. However, coastal systems are among the most dynamic environments on Earth, continually evolving at various spatio-temporal scales, with, for e.g. a single large storm being able to

reshape regional bathymetry which could significantly affect extreme coastal water levels in subsequent years. Thus, the consideration of passive coastal bathymetry over a 100-year period necessarily assumes that computed coastal flooding is only a function of changes in coastal water levels (Le Cozannet et al., 2019; Serafin et al., 2017; 2019). It should also be noted that local vertical land movement (e.g. land subsidence) can in places (e.g. Jakarta, New Orleans, Ho Chi Minh City) result in relative sea level rise rates that are far greater than the global mean sea level rise (Nicholls et al., 2014; Shirzaei and Bürgmann, 2018) considered in all future projections herein. Consideration of these regional and local contributions to relative sea level rise will affect coastal flooding projections for certain locations, in particular at coastal cities and low-lying deltas (Hallegatte et al., 2013; Erkens et al., 2015; Brown and Nicholls, 2015; Kulp and Strauss, 2019; Becker et al., 2020).

Finally, coastlines have been modified in various ways by human activities, particularly in urbanized areas in which, for example ports have been constructed, land has been reclaimed from the ocean (Luijendijk et al., 2018), seawalls built to combat coastline recession, cliffs stabilized, and groins placed in an attempt to retain a beach fringe and maintain dunes (Serafin et al., 2019). For example, in the US alone, 14% of national coastline is estimated to be hardened with engineering structures, and this percentage is expected to increase to 33% by 2100 (see Tavares et al., 2020). Such human interventions to the natural system generally results in steepening of coastal slopes (e.g. seawalls, dikes), resulting in smaller wave dissipation zones compared to natural coasts, which is not accounted for in this study.”

Line 197 “accurate modelling” – what does ‘accurate’ mean – it is meaningless without some standard as if you look at modelling in more and more detail you will always find inaccuracies. Review and improve to indicate your standards.

This sentence is no longer present in the new Limitations and way forward section.

Line 227-228 “To account for artificial coastal protection, the FLOod PROtection Standards FLOPROS (Scussolini et al., 2016) dataset was used as a third estimate of maximum subaerial coastal elevations (Vousdoukas et al., 2018).” How good is this dataset?

Arguably FLOPROS is very limited compared with satellite-based products (MERIT and AW3D30), as it is based on exceedance thresholds of ESWL, and so without accounting for wave effects. As a result, there is significant underestimation of computed overtopping all over the world. However, FLOPROS does have global coverage, so we chose to include it in our sensitivity analysis.

Line 229-250 “Coastal topography extraction” The method is not clear in terms of issues such as how far inland elevation is measured and other key factors. Please review and check that others could follow and reproduce your results. Also what is the estimated accuracy of the elevation.

In response to this and similar comments from other Reviewers, we have now enhanced this section with a

substantially enhanced Figure M1 (reproduced below). Furthermore, the text describing the extraction of coastal slopes and maximum coastal elevations has also been made more clear as follows (Lines 278 – 294):

“Maximum subaerial coastal elevation and coastal slopes were extracted from the above-mentioned MERIT, FLOPROS and AW3D30 datasets along the global coastline. Here, the Global Self-consistent, Hierarchical, High-resolution Geography Database (GSHHS - Wessel and Smith, 1996) coastline at the "h" highest resolution (~kilometric) was used. The coastal shoreline and topography are highly variable alongshore. In order to obtain reasonably robust estimates, cross-shore aerial topography profiles were extracted every 0.05 degrees alongshore (see **Figure M1**). From these, a regional profile every 0.5° alongshore was made robust by averaging ten 0.05° transects. This means that our analysis and conclusions are representative of main regional topographical features (e.g. typical low-lying beach-dune, high cliff coastline) but not of local features (e.g. estuaries). Furthermore, islands with a circumference less than 0.5° were excluded from the analysis, as we deemed it sufficient at a global scale and representative of the regional values seen in the literature. This resulted in a total of 14,140 profiles for which the analysis was performed.

The maximum coastal elevation and coastal slope at each profile were calculated using an automated detection method. In this method, the first step is the identification of the local sea-land orientation of each profile, based on the average topography values on the two sides of the shoreline: the higher side is taken to be land and the lower side to be sea. Second, the highest coastal point of each transect (e.g. dune, cliff top, crest of structure) was taken as the local landward maximum that was closest to shoreline, within 1 km landward of the shoreline (see **Figure M1**). The slope used in the wave contribution calculations was then estimated as the average slope between the shoreline and maximum coastal elevation, following the method presented by Diaz et al. (2019) – see insert in **Figure M1**.”

Figure M1. Regional AW3D30 topography in southwestern France compared to airborne LIDAR measurements that serves as a reference topography (from Diaz et al., 2019). Black and red segments represent cross shore profile examples at fine and coarse resolution, respectively. The insert shows LIDAR data and different cross-shore profiles (s1, s2 and s3 transects). Purple and blue lines in the insets represent the LIDAR and AW3D30 cross-shore transects, respectively. Markers are points used to compute the coastal slope (average of slopes between successive points from the shoreline to the maximum subaerial coastal elevation, indicated by a solid triangle).

Line 229-250 "Coastal topography extraction" The elevation is assumed to be constant with time – a quite reasonable assumption but this needs to be stated explicitly.

In response to this comment and similar comments from other Reviewers, we have now added an exhaustive *Limitations and way forward* section where this assumption (and others) are now explicitly stated and discussed. The new text related to this specific comment is as follows (Lines 234 – 247):

“The nearshore topography was considered constant in time here with no morphodynamic evolution, which means that possible sea level rise driven changes in the coastal slope and maximum coastal elevation are neglected herein. Even if detailed present-day bathymetry were available, past and future bathymetry would still remain unknown. As a result, this and other recent global studies, use a fixed coastal bathymetry over time periods spanning 50 – 100 years. However, coastal systems are among the most dynamic environments on Earth, continually evolving at various spatio-temporal scales, with, for e.g. a single large storm being able to reshape regional bathymetry which could significantly affect extreme coastal water levels in subsequent years. Thus, the consideration of passive coastal bathymetry over a 100-year period necessarily assumes that computed coastal flooding is only a function of changes in coastal water levels (Le Cozannet et al., 2019; Serafin et al., 2017; 2019). It should also be noted that local vertical land movement (e.g. land subsidence) can in places (e.g. Jakarta, New Orleans, Ho Chi Minh City) result in relative sea level rise rates that are far greater than the global mean sea level rise (Nicholls et al., 2014; Shirzaei and Bürgmann, 2018) considered in all future projections herein. Consideration of these regional and local contributions to relative sea level rise will affect coastal flooding projections for certain locations, in particular at coastal cities and low-lying deltas (Hallegatte et al., 2013; Erkens et al., 2015; Brown and Nicholls, 2015; Kulp and Strauss, 2019; Becker et al., 2020).”

Figure S6. These validations are hard to understand – where are they. For example, Katrina affected much of the coast of Louisiana and Mississippi so where is this – surges and waves varied substantially in space and there is hardly surge in this validation? Expand this part.

In response to this comment and similar comments by other Reviewers we have now reinforced and expanded this supplementary section S3 as follows (Lines 94-141):

“S3. Validation of the global overtopping computations for historical coastal flooding events

Our methodology with a 0.5° alongshore resolution (i.e. the resolution of the global satellite-based product adopted to derive coastal topography and the resolution of the global scale ocean forcing used in this analysis) precludes any conclusions at the local scale and hence our analysis should be considered as the first-pass, best-effort assessment.

Here, the methodology adopted to compute overtopping at the global scale is tested for four documented major coastal flooding events (**Figure S5**) along the Atlantic coast of Europe (**Figure S5.a**, Xynthia storm in France, Bertin et al., 2012), Gulf of Mexico (**Figure S5.b**, Hurricane Katrina hurricane in USA, Fritz et al., 2007), Mediterranean South-East coast (**Figure S5.c**, Nile delta in Egypt; Frihy et al., 2010; Refaat and Eldeberk, 2016; Ismail, et al., 2012), Gulf of Guinea, West Africa (**Figure S5.d**, Lagos in Nigeria; Nwilo et al., 1997; Olaniyan and Afiesimama, 2003) and Majuro in Pacific Marshall Islands (**Figure S5.e**, Hoeke et al., 2013).

The primary goal is to assess whether our method is able to reproduce the extreme still water level (ESWL) by comparing our calculations with sea level tide-gauge timeseries from the Global Extreme Sea Level Analysis

(GESLA) dataset (Woodworth et al., 2017), which has already been validated globally by Melet et al., (2018) (Figure S4 therein). The Extreme coastal water level is then computed by adding wave runup (from using local coastal slope and offshore waves in Stockdon'06 parametrization) to the ESWL.

Our second objective is to determine the capabilities of the methodology to capture these observed historical overtopping/flooding major events from the comparison of computed extreme coastal water levels with these sea levels. Overtopping thresholds are computed from regional topography maxima and are aimed at describing regional characteristics, not local features. As stated in the main manuscript, we do not intend here to describe small scale topography, dune breaching (for instance during Xynthia) and overflow due to local depressions in coastal elevation maxima (such as harbors, inlet), local shoreface singularities (such as coral reefs, convergence and sheltering, refraction/diffraction) but to detect regional characteristics of coastal topography and the general exposure to potential overtopping potential. Process-based numerical simulations perform better in capturing small-scale local behavior (in particular wave processes such as infragravity energy transfer – see Bertin et al., 2018). Here our focus is on regional and temporal patterns which fall within the scope of this study. The computed ESWL (ECWL minus wave runup) estimate (dashed black) shows a good agreement with GESLA tidal gauge derived ESWL (thin black). It should be noted that that ESWL alone cannot explain the overtopping and flooding observed in these case studies. It is only when wave runup is added that the water level exceeds the coastal elevation maxima for these 5 events. This is evident for the Gulf of Guinea case (**Figure S5.d**) where large tidal (spring) amplitude alone does not induce overtopping, but overtopping does occur even at lower tidal amplitude with concurrent large waves. This is particularly the case for Lagos and Pacific Islands events where the flooding is due to distant swell (Hoeke et al., 2013; Ford et al., 2018) by contrast to local storms associated with strong winds and surge (i.e. Xynthia storm in France and Katrina in US). This comparison with historical events together with the global sensitivity analysis of overtopping computations to different topography datasets provide sufficient confidence in the ability of the methodology implemented in this study to capture the salient characteristics of coastal overtopping at regional to global scale.

Figure S5. Validation of overtopping event detection with reported coastal flooding events: a) along the Atlantic coast of Europe (Xynthia storm in France, Bertin et al., 2012), b) Gulf of Mexico (Hurricane Katrina in USA, Fritz et al., 2007), c) Mediterranean South-East coast (Nile delta in Egypt; Frihy et al., 2010; Refaat and Eldeberk, 2016; Ismail, et al., 2012), d) Gulf of Guinea, West Africa (Lagos in Nigeria; Nwilo et al., 1997; Olaniyan and Afiesimama, 2003) and e) Majuro in Marshall Pacific Islands, USA (Hoeke et al., 2013). Our Extreme Still Water Level ESWL (ECWL minus wave runoff) estimate (dashed black line) is compared with that obtained from GESLA tidal gauges (thin black line). Total Extreme Coastal Water Level (ECWL), including wave runoff (thick black line), is compared to regional maximum coastal elevation (red continuous lines) from AW3D30. The MERIT derived maximum coastal elevation is also given (red dash) for comparison. Overtopping occurs when extreme coastal water level exceeds the maximum coastal elevation considered as the threshold for overtopping.“

References: There are errors or inconsistencies in the references which need a complete check – what is Kulp et al (2019) and Minderhoud (2019) in the text which are not in the references – although there are other similar references. Luijendijk et al., 2018 is missing in the references. Beetham, E. P. & Kench, P. reference incomplete.

These are just examples – other errors were noted.

We have re-checked all references in the revised manuscript and in the list of references, following NComm guidelines and corrected all errors.

English: A native speaker should review the revised document before it is resubmitted to smooth the language where needed.

We agree. Typos were present in the first submitted manuscript, for which we apologize. In the new version, the text has been largely restructured, improved and reviewed by native speakers to help convey our message in a coherent and cogent way and to strengthen the manuscript overall.

REVIEWER COMMENTS

Reviewer #1 (Remarks to the Author):

General Comments on re-review of Almar et al.

The reviewer would like to apologize for the time the re-review has taken.

The manuscript has been substantially improved. Overall the manuscript is interesting and the reviewer commends the authors on this undertaking. However, a significant concern remains regarding the backshore elevations and their potential impact on overtopping estimates and how that may affect the study conclusions. Going back to the original review concern that AW3D30 vertical elevation errors are approximately 5 m. The lack of consideration for overtopping uncertainty given the large topographic errors are concerning and could impact the conclusions of the paper. Comparing it to other datasets with large vertical errors is unsatisfying (e.g., section S2). There are many regions that have high quality LiDAR data available for much of the time period the authors investigate. If this were used in a few regions to validate the general trends in their analysis, the results would be much more convincing.

The Zhang et al., 2019 paper that is referenced regarding the AW3D30 (ALOS) compared to GPS and LiDAR. The difference between ALOS and LiDAR data is over 4 m (below). It feels uninformed to simply cite a study that shows such large differences without any additional vetting of the results.

Figures from Zhang et al., 2019 showing an elevation difference of nearly 5 m.

Reprinted from Remote Sensing of Environment, Volume 225, Zhang et al., Accuracy assessment of ASTER, SRTM, ALOS, and TDX DEMs for Hispaniola and implications for mapping vulnerability to coastal flooding, 290-306, Copyright 2019, with permission from Elsevier.

The author's state in L340 "A sensitivity analysis of the overtopping projections to the choice of the topography dataset (i.e. AW3D30, MERIT, FLOPROS) was conducted and the results are shown in Supplementary Figure S4 and Supplementary Section S2. Figure S4 shows that using FLOPROS or MERIT in the computations leads to higher overtopping rates, which is a direct result of the lower estimates of maximum subaerial coastal elevation compared in these two datasets, compared to AW3D30 (Figure S2)." But the figure M1 clearly shows the (more accurate) LiDAR data being ~2m higher, this would result in less overtopping. Notably these differences are larger than the SLR signal they are testing for. It is obvious overtopping will go up with SLR, but the quantification is valuable and also called into question with the potentially large topographic errors.

The misleading title has not been revised as the response suggests. Please update the title to the one the authors propose in the response "A global analysis of trends in potential coastal overtopping" or a more accurate title would be "A global analysis of extreme coastal water levels" since overtopping is not explicitly considered and ECWL is used as a proxy.

The authors have addressed the datum issue.

Figures have been updated and are more consistent with typical Nature standards.

This study does not explicitly consider overtopping. It considers ECWL. It is key that the authors explicitly state that they are using a Stockdon runup derived ECWL as a proxy for overtopping. Although this is a reasonable proxy, strictly speaking it is not overtopping and as such the title in the current manuscript remains misleading.

Specific items:

L122-23: "Our results show that the combination of tides and wave runup is the main contributor to episodic coastal overtopping." This assertion is misleading. Compound coastal flooding is not explicitly considered and the fact the concomitant high tide and waves cause coastal overtopping is very well known in the literature. I recommend the authors remove this particular sentence.

L160, L177, L294, L298 – I believe insert should be inset

L186 "Our results confirm that overtopping events are mainly due to the combined effect of wave runup astronomical tides" – This has been confirmed many times in the literature. Please see L122-123 above. Please remove.

L317-321 – The equations do not look typeset, they appear to be inserted. The pictures should be removed and the equations added properly.

L213 The following statement is untrue. "it is currently impossible to observe sea levels right at the coast". It is possible to do so, it is not possible to do so globally.

L219 – I believe you are missing a period after "formulae"

Reviewer #2 (Remarks to the Author):

The authors performed a very thorough and thoughtful job of responding to the reviews. The section on limitations was a particularly necessary addition to round out the paper and discuss uncertainties. I have no further requests for manuscript modifications.

Reviewer #3 (Remarks to the Author):

The authors have responded in detail to my and the other reviewer's review comments. They have clarified most of the review points that I raised. However, I still have a concern that the paper is unclear concerning the interpretation and implications of the results. Most importantly, it still implies that coastal overtopping could impact the entire low elevation coastal zone – this is not correct and I think it is important that the length scale of the flood impacts should be communicated – while it is hard to generalise in a global scale analysis – in most cases the flood effects of waves can be thought of a line process and will be fairly localised – swash effects are rarely limited to the shore immediately adjacent to breaking waves—wave set-up may have larger effects but it is worth noting that Kirezci et al (2020) found the effects of wave set-up to be relatively small. The authors need to make sure that the implications of the results in this paper are expressed – rather than the possible general consequences of flooding.

I makes some suggestions below.

ABSTRACT – first two sentences (Line 17 to 19) should be deleted as they are a misleading distraction that overstate the importance of overtopping

Line 35-43 – whole paragraph overstates the potential impact of coastal flooding -- the estimate of 1 billion people is the low elevation coastal zone population according to Kulp and Strauss (2019) – which is just one paper – and only some fraction of the low elevation coastal zone people will be impacted by sea-level rise.

Line 63 – delete "long-felt" – I would disagree with this phrase so keep it simple and uncontroversial

Line 78-80 – waves and wave run-up are only relevant to the 500,000 km open coast shoreline – on the other more sheltered coasts wave action is less on the important. The "14,140 coastal profiles around the world" will be situated on the 500,000 km of open coast shoreline. I think this could be explained more clearly.

Figure 2 a – hard to see the light yellow – how do you make this clearer.

Line 181-201 – here some sense of the scale of the impact would be useful – replacing the earlier text I suggested be deleted.

Line 203 "when interpreting our results at local scale." – should global scale studies be interpreted at the local scale? – Here I am interested in the global scale and regional implications.

Line 203-211 – could be talking about the scale of wave effects here

Line 227-229 – should have a reference here for hurricanes – e.g., Bloemendaal et al (2020)

Response to Reviewer comments

We express our gratitude again to the editor and the reviewers for carefully reading our revised manuscript and for providing constructive comments through this really interactive discussion. We have given full attention to all individual comments and revised the manuscript accordingly. The reviewers' feedback and suggestions have resulted in an improved manuscript, which we hope is now suitable for publication. Our detailed responses to each comment are provided below. For clarity, we use the following color code:

reviewer comments

our response

our modifications in the manuscript

GENERAL RESPONSES TO REVIEWER COMMENTS

We thank the editor for inviting a revision of our work for further consideration. Our revision addresses all points raised by our reviewers. In particular, the main points which are a continuation of the discussion that commenced in the first round of reviews:

#1: Reviewer #1 raises again the issue of the accuracy of maximum coastal elevations derived from global satellite DEMs.

Following this comment, we have added a second region with airborne LIDAR data to assess the accuracy of especially DEM derived maximum coastal elevations. This second region is in The Netherlands, and contains a large low-lying area fronted by a beach-dune system. Also, to comply with the Reviewer recommendations, we have added some other commonly used global satellite DEMs that are freely available (CoastalDEM, MERIT, SRTM, ASTER and TandemX) in our augmented assessment of the accuracy of DEM derived maximum coastal elevations. The new Figure M1 shows that ALOS-AW3D30 is indeed the satellite DEM with the lowest error with respect to maximum coastal elevation estimations, which is the critical parameter where coastal overtopping is concerned. The main reason for the superior performance of AW3D30 can be explained by the elevation baseline in AW3D30 using the ICESat1 mission, which is more accurate in terms of altitudes. The residual discrepancies between the AWD3D and Lidar comparisons of coastal maximum elevations shown in our analysis (average error ~ 1.5 m -2 m for AWD3D) are likely to be at least partly due to the fact that satellite DEMs are by necessity constructed by combining data acquired over a number of year (in the case of AWD3D, this period spanned 2006 – 2011), while

Lidar data for a region is collected within a few days, meaning that there will always be a mismatch in the timing of the two data sets being compared. This type of discrepancies will be impossible to avoid unless satellites achieve higher sampling rates.

Furthermore, as we have highlighted in the manuscript (Lines 216-217), the scale our study is global, and studies of this scale cannot capture very local scale details. However, we believe that global first-pass estimates such as ours will enable the determination of current trends and the identification of regional hotspots, which may then be investigated further via higher resolution, local studies at vulnerable locations.

Finally, although our study is the first to specifically focus on coastal overtopping (i.e. including the effect of wave runup) at global scale, we also wish to point out that ours is not the first study to use satellite DEMs for coastal flooding studies. There are many published global/regional studies that have used satellite DEMs to assess coastal flooding due to extreme total water levels (i.e. combination of tide, surge and wave setup) or extreme still water level (i.e. combination of tide and surge). These include, but are not limited to Hinkel et al., 2014; Brown et al., 2016; Muis et al., 2017; Vousdoukas et al., 2018b; Lincke and Hinkel., 2018; Kirezci et al., 2020; Vousdoukas et al., 2020a; Tiggeloven et al., 2020 etc.

We have addressed this Reviewer comment in the *Limitations and way forward* Section as follows (Lines 208-219):

“Although satellite DEMs are increasingly used in global/regional coastal flooding studies (e.g. Hinkel et al., 2014; Brown et al., 2016; Muis et al., 2017; Vousdoukas et al., 2018b; Lincke and Hinkel, 2018; Kirezci et al., 2020; Vousdoukas et al., 2020a; Tiggeloven et al., 2020), the quantitative accuracy of such assessments would necessarily be a function of the noise and accuracy associated with the DEM used; a limitation also applicable to our study. However, with more and more advanced technologies used in successive missions, the accuracy of satellite DEMs is continually improving, enabling more reliable impact assessments based on satellite DEMs. While a global validation of AW3D30 is beyond the scope of the present study, the regional DEM/Lidar comparisons shown Figure M1 for two different sites in France and The Netherlands show that, among the 6 satellite DEMs tested here (CoastalDEM, MERIT, SRTM, ASTER, TandemX, and AW3D30), AW3D30 has the lowest error (average error ~ 1.5 m -2 m) with respect to maximum coastal elevation estimations (i.e. the critical parameter in terms of overtopping). It should however be noted that the results of global scale studies cannot be expected capture intricate local scale details. Nevertheless, first-pass global scale studies, such as that presented here, will enable the determination of current trends

and the identification of regional hotspots, which may then be investigated further via higher resolution, local studies at vulnerable locations.”

and in the *Data and Methods* Section (Lines 297-313):

“Figure M1 shows the performance of six different satellite DEMs (CoastDEM, MERIT, SRTM, ASTER, TandemX and AW3D30) relative to airborne LIDAR data at two low-lying coastal regions: (a) the open coast of the Netherlands, which covers the largest part of the North and South Holland provinces (acquired for the entire country over the 2014-2019 period) and (b) the South West coast of France (acquired in October 2017). Despite the fact that the AW3D30 DEM is based on composite imagery acquired over the period 2006 – 2011, different to the LIDAR dates, it is still the DEM with lowest error (average error ~ 1.5 m -2 m) among the six DEMs considered, in terms of maximum coastal elevation estimations - the critical parameter where coastal overtopping is concerned.”

New Figure M1:

“Figure M1. Error quantification of coastal maximum elevations derived from six global DEMs (CoastDEM, MERIT, SRTM, ASTER, TandemX and AW3D30), relative to LIDAR data for two regions in France and the Netherlands: Panels (a) and (b) respectively show the maps of the local elevation data and the transects (5 km spacing) that were used to calculate the differences, in France (10 transects) and the Netherlands (29 transects). Panels (c) and (d) show the elevation profiles derived for example profiles (red lines in the maps) in France and the Netherlands. Identified coastal maxima for each case are shown by a circle marker. Panel (e) presents boxplots of the coastal maximum elevation differences separately for France and the Netherlands for each global DEM considered. Boxes indicate the 25th-75th percentile range, with a horizontal line and dot showing the median and the mean respectively. Whiskers indicate the 5th-95th range and circles points that are out of this range.”

#2: Reviewer #3 highlights that implications of the results to flood hazard should be indicated more precisely.

The Reviewer already mentioned this point in the previous round of reviews and we thank him/her for raising this issue again. We now realize that in our revised submission we had only partly addressed this comment by specifying the limitations of our methodology and the areas at which it may be less applicable (e.g. sheltered estuaries). But the fundamental difference between overtopping and flooding was still missing in our response to this comment, and we apologize for overlooking this. In response to this comment, we have now added text in the *Introduction* and *Limitations and way forward* Sections, and better nuanced our previous references to overtopping vs coastal flooding throughout the manuscript. Apart from the many changes we have made in the text in response to this comment (see below the point-by point response), we have now highlighted the variability of inland and alongshore length scales associated with overtopping in the *Introduction* (Lines 37-44) as follows:

“One process that could lead to coastal flooding is overtopping, which occurs when the extreme coastal water level (ECWL, as defined by Gregory et al., 2019) exceeds the maximum coastal elevation (e.g. dunes, dykes, cliffs; Allsop et al., 2018). However, the occurrence of overtopping does not necessarily imply that the entire low elevation coastal zone is flooded, rather this phenomenon drives localized coastal flooding, immediately adjacent to areas of overtopping. The flooding that may occur due to overtopping is likely to be both temporally and spatially variable due to the combined effects of temporal and alongshore gradients in breaking wave heights, and alongshore variations in coastal elevation maxima. In addition, overtopping can result in protection failure

(Schmocker and Hager, 2012) which can result in broader, more catastrophic flooding (Bertin et al., 2014).”

Moreover, a new paragraph has been added in the *Limitations and way forward* Section (Lines 202-207):

“Furthermore, when interpreting the consequences of overtopping on coastal flooding, it should be noted that the occurrence of overtopping does not necessarily imply that the entire low elevation coastal zone is flooded. Rather, overtopping drives localized coastal flooding, immediately adjacent to areas of overtopping, which would likely be both temporally and spatially variable due to the combined effects of temporal and alongshore gradients in breaking wave heights, and alongshore variations in coastal elevation maxima.”

#3: Following Reviewer #3’s suggestion, we have toned down our introductory statement on the potential impact of coastal flooding. The modified text now reads (Lines 33-35):

“Over the 21st century, sea level rise is projected to at least double the frequency of coastal flooding at most locations around the world (Vitousek et al., 2017; Vousdoukas et al., 2018a; Oppenheimer et al., 2019; Lambert et al. 2020,) potentially affecting millions of people living in low lying coastal zones, unless effective flood mitigation strategies are implemented in the years ahead (Nicholls and Small, 2002; Neumann et al., 2015; Tebaldi et al., 2012; Townend et al., 1994; Kulp et al., 2019).”

#4: Also in response to a comment by Reviewer #3, we have now clarified that waves and wave run-up are only relevant to ~ 500,000 km of open coast shoreline. The modified text now reads (Lines 76-77):

“Of the global coastline spanning approximately 1.5 million kilometers, only about 1/3^d is exposed to waves, with direct wave action being less relevant on sheltered coasts, including bays, estuaries and rugged coasts (Hinkel et al., 2013; Burke et al., 2001).”

#5: Based on another comment by Reviewer #3, while introducing the “14,140 coastal profiles around the world”, we have now better nuanced this as (Lines 98-99):

“14,140 coastal profiles situated along the open coasts of the world”

REVIEWER COMMENTS

Reviewer #1:

R1.1: The manuscript has been substantially improved. Overall the manuscript is interesting and the reviewer commends the authors on this undertaking. However, a significant concern remains regarding the backshore elevations and their potential impact on overtopping estimates and how that may affect the study conclusions. Going back to the original review concern that AW3D30 vertical elevation errors are approximately 5 m. The lack of consideration for overtopping uncertainty given the large topographic errors are concerning and could impact the conclusions of the paper. Comparing it to other datasets with large vertical errors is unsatisfying (e.g., section S2). There are many regions that have high quality LiDAR data available for much of the time period the authors investigate. If this were used in a few regions to validate the general trends in their analysis, the results would be much more convincing.

Please see our general response #1 above. Following this comment, we have added a second region with airborne LIDAR data to assess the accuracy of especially DEM derived maximum coastal elevations. This second region is in the The Netherlands, and contains a large low lying area fronted by a beach-dune system. Also, to comply with the Reviewer recommendations, we have added some other commonly used global satellite DEMs that are freely available (CoastalDEM, MERIT, SRTM, ASTER and TandemX) in our augmented assessment of the accuracy of DEM derived maximum coastal elevations. . The new Figure M1 shows that ALOS-AW3D30 is indeed the satellite DEM with the lowest error with respect to maximum coastal elevation estimations, which is the critical parameter where coastal overtopping is concerned. Specifically, the manuscript has been modified in two places (mainly) in response to this comment.

Lines 208-219 in the *Limitations and way forward* Section:

“Although satellite DEMs are increasingly used in global/regional coastal flooding studies (e.g. Hinkel et al., 2014; Brown et al., 2016; Muis et al., 2017; Vousdoukas et al., 2018b; Lincke and Hinkel., 2018; Kirezci et al., 2020; Vousdoukas et al., 2020a; Tiggeloven et al., 2020), the quantitative accuracy of such assessments would necessarily be a function of the noise and accuracy associated with the DEM used; a limitation also applicable to our study. However, with more and more advanced technologies used in successive missions, the accuracy of satellite DEMs is continually improving, enabling more reliable impact assessments based on satellite DEMs. While a global validation of AW3D30 is beyond the scope of the present study, the regional DEM/Lidar comparisons shown Figure M1 for two different sites in France and The Netherlands show that, among the 6 satellite DEMs tested here (CoastalDEM, MERIT, SRTM, ASTER, TandemX, and AW3D30), AW3D30

has the lowest error (average error ~ 1.5 m -2 m) with respect to maximum coastal elevation estimations (i.e. the critical parameter in terms of overtopping). It should however be noted that the results of global scale studies cannot be expected capture intricate local scale details. However, first-pass global scale studies, such as that presented here, will enable the determination of current trends and the identification of regional hotspots, which may then be investigated further via higher resolution, local studies at vulnerable locations.”

Lines 297-313 in the *Data and Methods* Section:

“Figure M1 shows the performance of six different satellite DEMs (CoastDEM, MERIT, SRTM, ASTER, TandemX and AW3D30) relative to airborne LIDAR data at two low-lying coastal regions: (a) the open coast of the Netherlands, which covers the largest part of the North and South Holland provinces (acquired for the entire country over the 2014-2019 period) and (b) the South West coast of France (acquired in October 2017). Despite the fact that the AW3D30 DEM is based on composite imagery acquired over the period 2006 – 2011, different to the LIDAR dates, it is still the DEM with lowest error (average error ~ 1.5 m -2 m) among the six DEMs considered, in terms of maximum coastal elevation estimations - the critical parameter where coastal overtopping is concerned.”

New Figure M1:

“Figure M1. Error quantification of coastal maximum elevations derived from six global DEMs (CoastDEM, MERIT, SRTM, ASTER, TandemX and AW3D30), relative to LIDAR data for two regions in France and the Netherlands: Panels (a) and (b) respectively show the maps of the local elevation data and the transects (5 km spacing) that were used to calculate the differences, in France (10 transects) and the Netherlands (29 transects). Panels (c) and (d) show the elevation profiles derived for example profiles (red lines in the maps) in France and the Netherlands. Identified coastal maxima for each case are shown by a circle marker. Panel (e) presents boxplots of the coastal maximum elevation differences separately for France and the Netherlands for each global DEM considered. Boxes indicate the 25th-75th percentile range, with a horizontal line and dot showing the median and the mean respectively. Whiskers indicate the 5th-95th range and circles points that are out of this range.”

R1.2: The Zhang et al., 2019 paper that is referenced regarding the AW3D30 (ALOS) compared to GPS and LiDAR. The difference between ALOS and LiDAR data is over 4 m (below). It feels uninformed to simply cite a study that shows such large differences without any additional vetting of the results. Figures from Zhang et al., 2019 showing an elevation difference of nearly 5 m (see pdf).

Indeed, we apologize for mentioning this study without providing further information. As suggested by the Reviewer, we chose to add another comparison site with LIDAR in our own study, while extending our comparison to 6 different global satellite DEMs. We have therefore removed this reference to Zhang et al. (2019) here.

R1.3: The author's state in L340 "A sensitivity analysis of the overtopping projections to the choice of the topography dataset (i.e. AW3D30, MERIT, FLOPROS) was conducted and the results are shown in Supplementary Figure S4 and Supplementary Section S2. Figure S4 shows that using FLOPROS or MERIT in the computations leads to higher overtopping rates, which is a direct result of the lower estimates of maximum subaerial coastal elevation compared in these two datasets, compared to AW3D30 (Figure S2)." But the figure M1 clearly shows the (more accurate) LiDAR data being ~2m higher, this would result in less overtopping. Notably these differences are larger than the SLR signal they are testing for. It is obvious overtopping will go up with SLR, but the quantification is valuable and also called into question with the potentially large topographic errors.

Figure M1 has now been completely revised using LIDAR data from two different regions, and 6 different global satellite DEMs. This revised Figure M1 shows that, of the six DEMs compared, AW3D30 has the lowest error (average error ~ 1.5 m -2 m) with respect to maximum coastal elevation estimations (i.e. the critical parameter in terms of overtopping), which is of the same order of magnitude of the maximum sea level rise (1 m) considered in our study. This, in our view, provides

sufficient confidence in the overtopping results we have produced. Apart from this, the sensitivity test results shown in Supplementary Figure S4 provides further confidence in our results. Here, it can be seen that globally aggregated annual number of overtopping hours computed using different DEMs result in the same qualitative behaviour, with the estimates computed with AW3D30 lying in the middle of the range of estimates. This supports one of the key messages of this study that the globally aggregated annual number of overtopping hours are increasing at a rate faster than that of the global mean sea level rise itself, following an exponential relationship between rates of overtopping and sea level rise. Note also that this conclusions considers “rate of sea level rise”, and not the “absolute” amount of sea level rise.

Here we would like to also re-iterate that there are many published global/regional studies that have used satellite DEMs to assess coastal flooding due to extreme total water levels (i.e. combination of tide, surge and wave setup) or extreme still water level (i.e combination of tide and surge). These include, but are not limited to Hinkel et al., 2014; Brown et al., 2016; Muis et al.,2017; Vousdoukas et al., 2018b; Lincke and Hinkel., 2018; Kirezci et al., 2020; Vousdoukas et al., 2020a; Tiggeloven et al., 2020 etc). We have now mentioned this in the manuscript as follows in the *Limitations and way forward* Section (Lines 208-211):

“Although satellite DEMs are increasingly used in global/regional coastal flooding studies (e.g. Hinkel et al., 2014; Brown et al., 2016; Muis et al.,2017; Vousdoukas et al., 2018b; Lincke and Hinkel., 2018; Kirezci et al., 2020; Vousdoukas et al., 2020a; Tiggeloven et al., 2020), the quantitative accuracy of such assessments would necessarily be a function of the noise and accuracy associated with the DEM used; a limitation also applicable to our study. However, with more and more advanced technologies used in successive missions, the accuracy of satellite DEMs is continually improving, enabling reliable assessments based on satellite DEMs.”

The comparison shown in Figure M1 indicates that some of the DEMs used in these previous studies, which are published in reputed journals, may contain higher errors than the AW3D30 DEM we have adopted in our study.

R1.4: The misleading title has not been revised as the response suggests. Please update the title to the one the authors propose in the response “A global analysis of trends in potential coastal overtopping” or a more accurate title would be “A global analysis of extreme coastal water levels” since overtopping is not explicitly considered and ECWL is used as a proxy.

Based on this suggestion and our own deliberations on how best to convey the main outcome of our study we have now changed the title to “A global analysis of extreme coastal water levels with implications for potential coastal overtopping”.

R1.5: The authors have addressed the datum issue.

No action needed

R1.6: Figures have been updated and are more consistent with typical Nature standards.

No action needed

R1.7: This study does not explicitly consider overtopping. It considers ECWL. It is key that the authors explicitly state that they are using a Stockdon runup derived ECWL as a proxy for overtopping.

We completely agree, and have now made this clear in text as follows:

Lines 64-67 in the *Introduction*:

“Using the occurrence of ECWLs above the maximum coastal elevation as a proxy for coastal overtopping, here we quantify, for the first time, the global scale increase of potential coastal overtopping in recent decades, and present first-pass, globally aggregated projections of future coastal overtopping in response to projected global mean sea level rise over the 21st century.”

R1.8: Although this is a reasonable proxy, strictly speaking it is not overtopping and as such the title in the current manuscript remains misleading.

We hope that the textual changes we have made in response to Reviewer comments and detailed in our above responses address this concern. We have also modified the title and have now clearly stated that we are using ECWL as a proxy in the Introduction. However, after much deliberation, we kept the reference to overtopping in the title, as the wide readership of *Nature Communications* may include many non-specialists who would be unaware of the exact definitions of the different types of extreme sea levels, such as ECWL. In contrast, the word overtopping is more easily understood by all and sundry.

Specific items:

R1.9: L122-23: “Our results show that the combination of tides and wave runup is the main contributor to episodic coastal overtopping.” This assertion is misleading. Compound coastal flooding is not explicitly considered and the fact the concomitant high tide and waves cause coastal overtopping is very well known in the literature. I recommend the authors remove this particular sentence.

We agree. This sentence has now been removed.

R1.10: L160, L177, L294, L298 – I believe insert should be inset

Thank you. We have corrected this now.

R1.11: L186 “Our results confirm that overtopping events are mainly due to the combined effect of wave runup astronomical tides” – This has been confirmed many times in the literature. Please see L122-123 above. Please remove.

Following the Reviewer’s previous recommendation, we removed a sentence (L122-123 in previous manuscript). However here, we feel that is important to highlight that it is phenomena other than waves and tides alone do not explain the observed trend in overtopping. Therefore, we rephrased this sentence and opted to retain it here.

The revised sentences now read (Lines 180-183):

“While overtopping events are mainly due to the combined effect of wave runup astronomical tides, these processes alone do not explain the observed increasing trend in the globally aggregated annual overtopping hours. Rather, it is the combination of regional sea level, storm surge, wave runup and tide that is responsible for the observed increasing trend in overtopping, with the increasing trend in regional sea level being the main driver.”

R1.12: L317-321 – The equations do not look typeset, they appear to be inserted. The pictures should be removed and the equations added properly.

We thank the reviewer for pointing this out. This was due to the old versions of Microsoft Word replacing equations with images during auto-saving. This has now been fixed now using the latest version of MS Word.

R1.13: L213 The following statement is untrue. “it is currently impossible to observe sea levels right at the coast”. It is possible to do so, it is not possible to do so globally.

We thank the Reviewer for this sharp comment. We have now refined our statement as (Lines 220-221):

“it is currently impossible to observe sea levels right at the coast all along the global coastline,....”

R1.14:L219 – I believe you are missing a period after “formulae”

This has been corrected.

Reviewer #2 (Remarks to the Author):

R2.1: The authors performed a very thorough and thoughtful job of responding to the reviews. The section on limitations was a particularly necessary addition to round out the paper and discuss uncertainties. I have no further requests for manuscript modifications.

We thank Reviewer 2 again for his constructive comments that helped us to improve the manuscript and we are glad to see that he/she is satisfied with the revised version.

Reviewer #3 (Remarks to the Author):

R3.1: The authors have responded in detail to my and the other reviewer's review comments. They have clarified most of the review points that I raised. However, I still have a concern that the paper is unclear concerning the interpretation and implications of the results. Most importantly, it still implies that coastal overtopping could impact the entire low elevation coastal zone – this is not correct and I think it is important that the length scale of the flood impacts should be communicated – while it is hard to generalise in a global scale analysis – in most cases the flood effects of waves can be thought of a line process and will be fairly localised – swash effects are rarely limited to the shore immediately adjacent to breaking waves—wave set-up may have larger effects but it is worth noting that Kirezci et al (2020) found the effects of wave set-up to be relatively small. The authors need to make sure that the implications of the results in this paper are expressed— rather than the possible general consequences of flooding.

The Reviewer already mentioned this point in the previous round of reviews and we thank him/her for raising this issue again. We now realize that in our revised submission we had only partly addressed this comment by specifying the limitations of our methodology and the areas at which it may be less applicable (e.g. sheltered estuaries). But the fundamental difference between overtopping and flooding was still missing in our response to this comment, and we apologize for overlooking this. In response to this comment, we have now made the following changes in the text:

In the *Abstract* (Lines 21-22):

“While wave runup is widely recognized as an important determinant of episodic coastal flooding overtopping, it has hitherto been ignored in global-scale assessments.”

In the *Introduction* (Lines 37-44):

“One process that could lead to coastal flooding is overtopping, which occurs when the extreme coastal water level (ECWL, as defined by Gregory et al., 2019) exceeds the maximum coastal

elevation (e.g. dunes, dykes, cliffs; Allsop et al., 2018). However, the occurrence of overtopping does not necessarily imply that the entire low elevation coastal zone is flooded, rather, this phenomenon drives localized coastal flooding, immediately adjacent to areas of overtopping. The flooding that may occur due to overtopping is likely to be both temporally and spatially variable due to the combined effects of temporal and alongshore gradients in breaking wave heights, and alongshore variations in coastal elevation maxima. In addition, overtopping can result in protection failure (Schmocker and Hager, 2012) which can result in broader, more catastrophic flooding (Bertin et al., 2014)”

In the *Limitations and way forward* Section (Lines 202-207):

“Furthermore, when interpreting the consequences of overtopping on coastal flooding, it should be noted that the occurrence of overtopping does not necessarily imply that the entire low elevation coastal zone is flooded. Rather, overtopping drives localized coastal flooding, immediately adjacent to areas of overtopping, which would likely be both temporally and spatially variable due to the combined effects of temporal and alongshore gradients in breaking wave heights, and alongshore variations in coastal elevation maxima.”

I makes some suggestions below.

R3.2: ABSTRACT – first two sentences (Line 17 to 19) should be deleted as they are a misleading distraction that overstate the importance of overtopping

We followed the Reviewer advice and decided to remove the first sentence but we do believe that the second sentence provides a broad introduction to our manuscript and have opted to maintain it as it provides an important message. This sentence (Lines 18-19) has now been modified as:

“Climate change and anthropogenic pressures are widely expected to exacerbate coastal hazards such as episodic coastal flooding, the magnitudes of which remain highly uncertain to date.”

R3.3: Line 35-43 – whole paragraph overstates the potential impact of coastal flooding -- the estimate of 1 billion people is the low elevation coastal zone population according to Kulp and Strauss (2019) – which is just one paper – and only some fraction of the low elevation coastal zone people will be impacted by sea-level rise.

We have now rephrased this paragraph to convey the more general message that there are millions of people living in low lying coastal zone. The modified text now reads as (Lines 33-35):

“Over the 21st century, sea level rise is projected to at least double the frequency of coastal flooding at most locations around the world (Vitousek et al., 2017; Vousdoukas et al., 2018a; Oppenheimer et

al., 2019; Lambert et al. 2020,) potentially affecting millions of people living in low lying coastal zones, unless effective flood mitigation strategies are implemented in the years ahead (Nicholls and Small, 2002; Neumann et al., 2015; Tebaldi et al., 2012; Townend et al., 1994; Kulp et al., 2019).”

R3.4: Line 63 – delete “long-felt” – I would disagree with this phrase so keep it simple and uncontroversial

Modified as suggested

R3.5: Line 78-80 – waves and wave run-up are only relevant to the 500,000 km open coast shoreline – on the other more sheltered coasts wave action is less on the important. The “14,140 coastal profiles around the world” will be situated on the 500,000 km of open coast shoreline. I think this could be explained more clearly.

We thank the Reviewer for this very good comment and have now changed the text to better reflect this. The modified text reads as (Lines 76-77):

“Of the global coastline spanning approximately 1.5 million kilometers, only about 1/3rd is exposed to waves, with direct wave action being less relevant on sheltered coasts, including bays, estuaries and rugged coasts (Hinkel et al., 2013; Burke et al., 2001).”

and in Lines 98-99:

“14,140 coastal profiles situated along the open coasts of the world”

R3.6: Figure 2 a – hard to see the light yellow – how do you make this clearer.

This colormap was deliberately chosen to highlight the low lying coasts in red, which are most relevant for the present study, while the highest coastal elevations, where overtopping does not occur, appear in light yellow (> 20m).

R3.7: Line 181-201 – here some sense of the scale of the impact would be useful – replacing the earlier text I suggested be deleted.

We are not sure that we understand this comment well and to which earlier text it refers to, in this review or the previous review.

If the comment refers to the scales of overtopping versus flooding, this has been addressed in the previous responses to the comments and the text of the manuscript has been clarified accordingly, as follows:

In the *Introduction*, the length scales of overtopping has been clarified as follows (Lines 37-44):

“One process that could lead to coastal flooding is overtopping, which occurs when the extreme coastal water level (ECWL, as defined by Gregory et al., 2019) exceeds the maximum coastal elevation (e.g. dunes, dykes, cliffs; Allsop et al., 2018). However, the occurrence of overtopping does not necessarily imply that the entire low elevation coastal zone is flooded, rather, this phenomenon drives localized coastal flooding, immediately adjacent to areas of overtopping. The flooding that may occur due to overtopping is likely to be both temporally and spatially variable due to the combined effects of temporal and alongshore gradients in breaking wave heights, and alongshore variations in coastal elevation maxima. In addition, overtopping can result in protection failure (Schmocker and Hager, 2012) which can result in broader, more catastrophic flooding (Bertin et al., 2014”

and in Lines 202-207:

“Furthermore, when interpreting the consequences of overtopping on coastal flooding, it should be noted that the occurrence of overtopping does not necessarily imply that the entire low elevation coastal zone is flooded. Rather, overtopping drives localized coastal flooding, immediately adjacent to areas of overtopping, which would likely be both temporally and spatially variable due to the combined effects of temporal and alongshore gradients in breaking wave heights, and alongshore variations in coastal elevation maxima.”

R3.8: Line 203 “when interpreting our results at local scale.” – should global scale studies be interpreted at the local scale? – Here I am interested in the global scale and regional implications

We agree with the Reviewer, and of course they should not, but people do. So this is our attempt to caution people in case they do look to our results for local estimates. We have now modified this sentence as follows (Lines 196-197):

“Being a global scale assessment, inevitably there will be several limitations if attempts are made to interpret our results at local scale”

R3.9: Line 203-211 – could be talking about the scale of wave effects here

In response to the comment we have now added the following sentences (Lines 202-207):

“Furthermore, when interpreting the consequences of overtopping on coastal flooding, it should be noted that the occurrence of overtopping does not necessarily imply that the entire low elevation coastal zone is flooded. Rather, overtopping drives localized coastal flooding, immediately adjacent to areas of overtopping, which would likely be both temporally and spatially variable due to the combined effects of temporal and alongshore gradients in breaking wave heights, and alongshore variations in coastal elevation maxima.”

R3.10: Line 227-229 – should have a reference here for hurricanes – e.g., Bloemendaal et al (2020)

Thank you for this interesting reference. We have now cited it and added it to the reference list.

New reference added in Line 236 “for tropical cyclones (Bloemendaal et al., 2020)” and in the reference list:

“Bloemendaal, N., Haigh, I.D., de Moel, H. et al. Generation of a global synthetic tropical cyclone hazard dataset using STORM. *Sci Data* 7, 40 (2020)”

References cited

Bloemendaal, N. et al. Generation of a global synthetic tropical cyclone hazard dataset using STORM. *Sci. Data* 7, (2020).

Brown, S., Nicholls, R. J., Lowe, J. A. & Hinkel, J. Spatial variations of sea-level rise and impacts: An application of DIVA. *Clim. Change* 134, (2016).

Hinkel, J. et al. Coastal flood damage and adaptation costs under 21st century sea-level rise. *Proc. Natl. Acad. Sci.* 111, 3292–3297 (2014).

Kirezci, E. et al. Projections of global-scale extreme sea levels and resulting episodic coastal flooding over the 21st Century. *Sci. Rep.* 10, 1–12 (2020).

Lincke, D. & Hinkel, J. Economically robust protection against 21st century sea-level rise. *Glob. Environ. Chang.* 51, 67–73 (2018).

Muis, S. et al. A comparison of two global datasets of extreme sea levels and resulting flood exposure. *Earth’s Futur.* 5, 379–392 (2017).

Tiggeloven, T. et al. Global-scale benefit-cost analysis of coastal flood adaptation to different flood risk drivers using structural measures. *Nat. Hazards Earth Syst. Sci.* 20, (2020).

Vousdoukas, M. I. et al. Climatic and socioeconomic controls of future coastal flood risk in Europe. *Nat. Clim. Chang.* (2018). doi:10.1038/s41558-018-0260-4

Vousdoukas, M. I. et al. Economic motivation for raising coastal flood defenses in Europe. *Nat. Commun.* 11, (2020).

Zhang, K. et al. Accuracy assessment of ASTER, SRTM, ALOS, and TDX DEMs for Hispaniola and implications for mapping vulnerability to coastal flooding. *Remote Sens. Environ.* 225, (2019).

REVIEWERS' COMMENTS

Reviewer #3 (Remarks to the Author):

The authors have done a good job of making the paper more clear and I am happy for the paper to be published except for one point that does not stand up to scrutiny to me eyes is lines 162 and 163

"a noticeable sea level rise driven increase in $N_{a,g}$ is estimated to have already started (around 2020), as indicated by the upward inflection around 2020."

I have looked and looked and 2020 does not strike at all when I look at Figure 4 -- rather I see a smoothly accelerating trend

I see

"a noticeable sea level rise driven increase in $N_{a,g}$ is occurring with an accelerating trend through the 21st Century, especially with high emissions."

If you want a date where I would be confident it is non-linear I would say 2040 for high emissions -- it is definitely increasing beforehand but the acceleration is more subtle.

And there may be an element of the baseline -- if used global sea-level rise from 1900 you would see acceleration through the 20th century with some wobbles -- I am not asking the authors to do this -- but rather encouraging them to think carefully about.

With the text change I suggest and a review by the authors to make sure it does not affect anything else I would be happy for this to be published

Response to Reviewer comments

We express our gratitude to the editorial team and the reviewer #3 comment for carefully reading our revised manuscript and for providing constructive comments through this highly interactive discussion. We have given our full attention to the last comments made and revised the manuscript accordingly. For clarity, we use the following color code:

reviewer comments

our response

our modifications in the manuscript

REVIEWER #3 COMMENTS

The authors have done a good job of making the paper more clear and I am happy for the paper to be published except for one point that does not stand up to scrutiny to me eyes is lines 162 and 163 "a noticeable sea level rise driven increase in Na,g is estimated to have already started (around 2020), as indicated by the upward inflection around 2020." I have looked and looked and 2020 does not strike at all when I look at Figure 4 -- rather I see a smoothly accelerating trend I see "a noticeable sea level rise driven increase in Na,g is occurring with an accelerating trend through the 21st Century, especially with high emissions." If you want a date where I would be confident it is non-linear I would say 2040 for high emissions -- it is definitely increasing beforehand but the acceleration is more subtle. And there may be an element of the baseline -- if used global sea-level rise from 1900 you would see acceleration through the 20th century with some wobbles -- I am not asking the authors to do this -- but rather encouraging them to think carefully about.

With the text change I suggest and a review by the authors to make sure it does not affect anything else I would be happy for this to be published

We thank the reviewer for this comment. In response, we have made changes to the manuscript to reflect the fact that the accelerating trend will be present throughout the 21st century, especially with high emissions.

Modifications affected in the text include:

Abstract (was also shortened to comply with the 150 word limit) and Short summary (310 character limit):

"Abstract

Climate change and anthropogenic pressures are widely expected to exacerbate coastal hazards such as episodic coastal flooding. This study, for the first time, presents global-scale potential coastal overtopping estimates, which account for not only the effects of sea level rise and storm surge, but also for wave runup at exposed open coasts. We find that the globally aggregated annual overtopping hours have increased by almost 50 % over the last two decades. A first-pass future assessment indicates that globally aggregated annual overtopping hours will accelerate faster than the global mean sea-level rise itself, with a clearly discernible increase occurring around mid-century regardless of climate scenario. Under RCP 8.5, the globally aggregated annual overtopping hours by the end of the 21st-century is projected to be up to 50 times larger compared to present-day. As sea level continues to rise, more regions around the world are projected to become exposed to coastal overtopping.

Short summary

Global annual coastal overtopping has increased by almost 50 % over the last two decades. Climate change will lead to a clearly discernible increase in global overtopping hours by mid-century regardless of climate scenario, and may reach values that are up to 50 times larger by end-century compared to present day, under a low mitigation climate scenario.”

In the Results and Discussion, section Overtopping and projected sea level rise, text lines 158-163 now reads:

“Figure 4(b) also shows that, when the wave and tide contributions are included in the computation (grey bars), a clearly discernible increase in $N_{a,g}$ is projected to occur around mid-century regardless of climate scenario, as indicated by the upward inflection around mid-century. In contrast, if wave and tide contributions to overtopping were ignored, a noticeable increase in overtopping hours is only expected by around 2080, and only under RCP 8.5. Figure 4(b) shows that, relative to its present-day value, $N_{a,g}$ could be as much as 50 times larger by the end of the 21st century under RCP 8.5.”

And Results and Discussion, in the new Results summary section (Lines 248-249):

“The acceleration in coastal overtopping is expected to continue throughout the 21st century and will be clearly discernible by mid-century under any climate scenario.”

Again, we thank the Reviewer #3 and others for their thorough reading of the manuscript and providing such constructive comments.